# Adversarially Pretrained Transformers May Be Universally Robust In-Context Learners

**Soichiro Kumano**
The University of Tokyo
kumano@cvm.t.u-tokyo.ac.jp

**Hiroshi Kera**
Chiba University, National Institute of Informatics
kera@chiba-u.jp

**Toshihiko Yamasaki**
The University of Tokyo
yamasaki@cvm.t.u-tokyo.ac.jp

## Abstract

Adversarial training is one of the most effective defenses against adversarial attacks, but it incurs a high computational cost. In this study, we present the first theoretical analysis suggesting that adversarially pretrained transformers can serve as *universally robust foundation models*—models that can adapt robustly to diverse downstream tasks with only lightweight tuning. Specifically, we demonstrate that single-layer linear transformers, after adversarial pretraining across a variety of classification tasks, can generalize robustly to unseen classification tasks through in-context learning from clean demonstrations (i.e., without requiring additional adversarial training or examples). This universal robustness stems from the model's ability to adaptively focus on robust features within given tasks. We also identify two open challenges for attaining robustness: the accuracy–robustness trade-off and sample-hungry training. This study initiates the discussion on the utility of universally robust foundation models. While their training is expensive, the investment would prove worthwhile as downstream tasks can obtain adversarial robustness for free. The code is available at https://github.com/s-kumano/universally-robust-in-context-learner.

## 1 Introduction

Adversarial examples—subtle and often imperceptible perturbations to inputs that lead machine learning models to make incorrect predictions—reveal a fundamental vulnerability in modern deep learning systems (Szegedy et al., 2014). Adversarial training is one of the most effective defenses against such attacks (Goodfellow et al., 2015; Madry et al., 2018), where classification loss is minimized under worst-case (i.e., adversarial) perturbations. This min–max optimization significantly increases the computational cost compared to standard training. Despite extensive efforts to develop alternative defenses, most such approaches have subsequently been shown to offer only spurious robustness (Athalye et al., 2018; Croce & Hein, 2020; Tramer et al., 2020). Consequently, adversarial training remains the de facto standard, and practitioners must incur this cost to obtain adversarially robust models.

Recently, it has become standard practice to leverage foundation models for target tasks. Thanks to large-scale pretraining, these models can be adapted to diverse downstream tasks through lightweight tuning. This raises a natural question: *Can adversarially trained foundation models enable efficient and robust adaptation to a wide range of downstream tasks?* Although training such models is expensive, the investment would be justified if numerous downstream tasks could inherit adversarial robustness for free, without requiring costly adversarial training themselves. While this is a promising research direction, the utility of such *universally robust foundation models* remains largely unexplored, as the computational and financial costs make empirical evaluation across multiple runs impractical.

In this study, we present the first theoretical analysis suggesting that adversarially pretrained transformers can serve as universally robust foundation models. Specifically, we show that single-layer linear transformers, after adversarial pretraining across a variety of classification tasks, can generalize

robustly to previously unseen classification tasks through in-context learning (Brown et al., 2020) from clean demonstrations. Namely, these transformers achieve robust adaptation without requiring additional adversarial examples or training. In-context learning is a transformer capability that enables efficient adaptation to new tasks from a few input–output demonstrations in the prompt, without any parameter updates.

Our analysis builds upon the conceptual framework of robust features (class-discriminative and human-interpretable) and non-robust features (human-imperceptible yet predictive) (Ilyas et al., 2019; Tsipras et al., 2019). Based on this framework, we show that adversarially pretrained single-layer linear transformers can adaptively focus on robust features within each downstream task, rather than non-robust or non-predictive features, thereby achieving universal robustness. This framework also reveals that universal robustness holds under mild conditions, except in an unrealistic scenario where non-robust features overwhelmingly outnumber robust ones.

We also show that two open challenges in robust machine learning (Schmidt et al., 2018; Tsipras et al., 2019) persist in our setting. First, adversarially pretrained single-layer linear transformers exhibit lower clean accuracy than their standardly pretrained counterparts. Second, to achieve clean accuracy comparable to standard models, these transformers require more in-context demonstrations.

Our contributions are summarized as follows:

- We provide the first theoretical evidence for universally robust foundation models: under mild conditions, adversarially pretrained transformers with a single linear self-attention layer can adapt robustly to unseen classification tasks through in-context learning.
- Based on the framework of robust and non-robust features, we characterize the condition for successful robust adaptation. Moreover, we show that universal robustness arises from the models' adaptive focus on robust features within each downstream task.
- We identify two open problems for these transformers: the accuracy–robustness trade-off and sample-hungry in-context learning.

This study explores the potential of universally robust foundation models, which can provide diverse downstream tasks with adversarial robustness without additional adversarial training. A key challenge is the cost of adversarial pretraining. We assume that, as with standard foundation models, such efforts would be undertaken by large organizations, which could offset development costs through licensing or API fees. The growing demand for safe and reliable AI strengthens this incentive. Encouragingly, advances in acceleration techniques for adversarial training, such as fast adversarial training (Wong et al., 2020) and adversarial finetuning (Jeddi et al., 2020), suggest that the cost of adversarial training could approach the cost of standard training. We view our theoretical analysis as an important first step toward fostering the practical development of universally robust foundation models.

## 2  RELATED WORK

Additional related work can be found in Appendix A.

**Adversarial Training.** Adversarial training (Goodfellow et al., 2015; Madry et al., 2018), which augments training data with adversarial examples (Szegedy et al., 2014), is one of the most effective adversarial defenses. Its major limitation is the high computational cost. To address this, several methods have focused on efficient generation of adversarial examples (Andriushchenko & Flammarion, 2020; Kim et al., 2021; Park & Lee, 2021; Shafahi et al., 2019; Wong et al., 2020; Zhang et al., 2019a) and adversarial finetuning (Jeddi et al., 2020; Mao et al., 2023; Suzuki et al., 2023; Wang et al., 2024a). However, these methods require task-specific adversarial training. In this study, we introduce the concept of universally robust foundation models, which can adapt robustly to a wide range of downstream tasks without requiring any adversarial training or examples.

**Robust and Non-Robust Features.** It is widely hypothesized that adversarial vulnerability arises from models' reliance on non-robust features (Ilyas et al., 2019; Tsipras et al., 2019). While robust features are class-discriminative, human-interpretable, and semantically meaningful, non-robust features are subtle, often imperceptible to humans, yet statistically correlated with labels and therefore predictive. Humans can rely only on robust features, whereas models can leverage both types of features to maximize accuracy. Tsipras et al. (2019) showed that standard classifiers

depend heavily on non-robust features, making them vulnerable to adversarial perturbations that can manipulate these subtle features. They also showed that adversarial training encourages models to rely primarily on robust features, which enhances robustness but often reduces clean accuracy—a phenomenon known as the accuracy–robustness trade-off (Dobriban et al., 2023; Mehrabi et al., 2021; Raghunathan et al., 2019; 2020; Su et al., 2018; Tsipras et al., 2019; Yang et al., 2020; Zhang et al., 2019b). Subsequent studies have confirmed that adversarially trained neural networks exhibit a greater reliance on robust features (Augustin et al., 2020; Chalasani et al., 2020; Engstrom et al., 2019; Etmann et al., 2019; Kaur et al., 2019; Santurkar et al., 2019; Srinivas et al., 2023; Tsipras et al., 2019; Zhang & Zhu, 2019). In this study, we incorporate the concept of robust and non-robust features into the data assumptions for our theoretical analysis. Based on this framework, we find that adversarially pretrained single-layer linear transformers prioritize robust features over non-robust features, and exhibit the accuracy–robustness trade-off.

## 3 THEORETICAL RESULTS

**Notation.** For $n \in \mathbb{N}$, let $[n] := \{1, \ldots, n\}$. Denote the $i$-th element of a vector $\boldsymbol{a}$ by $a_i$, and the element in the $i$-th row and $j$-th column of a matrix $\boldsymbol{A}$ by $A_{i,j}$. Let $U(\mathcal{S})$ be the uniform distribution over a set $\mathcal{S}$. The sign function is denoted as $\mathrm{sgn}(\cdot)$. For $d_1, d_2 \in \mathbb{N}$, let $\boldsymbol{1}_{d_1}$ and $\boldsymbol{1}_{d_1,d_2}$ be the $d_1$-dimensional all-ones vector and $d_1 \times d_2$ all-ones matrix, respectively. The $d_1 \times d_1$ identity matrix is denoted as $\boldsymbol{I}_{d_1}$. Similarly, we write the all-zeros vector and matrix as $\boldsymbol{0}_{d_1}$ and $\boldsymbol{0}_{d_1,d_2}$, respectively. We use $\gtrsim, \lesssim$, and $\approx$ only to hide constant factors in informal statements.

### 3.1 PROBLEM SETUP

**Overview.** We adversarially train a single-layer linear transformer on $d \in \mathbb{N}$ distinct datasets. The $c$-th training data distribution is denoted by $\mathcal{D}_c^{\mathrm{tr}}$ for $c \in [d]$. The $c$-th dataset consists of $N+1$ samples, $\{(\boldsymbol{x}_n^{(c)}, y_n^{(c)})\}_{n=1}^{N+1} \overset{\text{i.i.d.}}{\sim} \mathcal{D}_c^{\mathrm{tr}}$. The transformer is encouraged to adaptively learn data structures from $N$ clean in-context demonstrations $\{(\boldsymbol{x}_n, y_n)\}_{n=1}^{N}$ and to generalize to the $(N+1)$-th perturbed sample $\boldsymbol{x}_{N+1} + \boldsymbol{\Delta}$, where $\boldsymbol{\Delta}$ represents an adversarial perturbation. We then evaluate the adversarial robustness of the trained transformer on a test dataset $\{(\boldsymbol{x}_n^{\mathrm{te}}, y_n^{\mathrm{te}})\}_{n=1}^{N+1} \overset{\text{i.i.d.}}{\sim} \mathcal{D}^{\mathrm{te}}$, which may exhibit different structures from any training distributions.

**Transformer.** We first define the input sequence for a transformer as

$$\boldsymbol{Z}_{\boldsymbol{\Delta}} := \begin{bmatrix} \boldsymbol{x}_1 & \boldsymbol{x}_2 & \cdots & \boldsymbol{x}_N & \boldsymbol{x}_{N+1} + \boldsymbol{\Delta} \\ y_1 & y_2 & \cdots & y_N & 0 \end{bmatrix} \in \mathbb{R}^{(d+1)\times(N+1)}, \tag{1}$$

where $\boldsymbol{x}_1, \ldots, \boldsymbol{x}_N \in \mathbb{R}^d$ are training data, $y_1, \ldots, y_N \in \{\pm 1\}$ are their binary labels, $\boldsymbol{x}_{N+1} \in \mathbb{R}^d$ is a test sample (query), and $\boldsymbol{\Delta} \in \mathbb{R}^d$ is an adversarial perturbation (see below). The transformer is expected to learn data structures adaptively from $N$ demonstrations $\{(\boldsymbol{x}_n, y_n)\}_{n=1}^{N}$ and to predict the label of $\boldsymbol{x}_{N+1}$. The $(d+1, N+1)$-th element of $\boldsymbol{Z}_{\boldsymbol{\Delta}}$ serves as a placeholder for the prediction of $\boldsymbol{x}_{N+1} + \boldsymbol{\Delta}$. We define a single-layer linear transformer $\boldsymbol{f} : \mathbb{R}^{(d+1)\times(N+1)} \to \mathbb{R}^{(d+1)\times(N+1)}$, which is commonly employed in theoretical studies of in-context learning (Ahn et al., 2023; Cheng et al., 2024; Gatmiry et al., 2024; Mahankali et al., 2024; Zhang et al., 2024b), as follows:

$$\boldsymbol{f}(\boldsymbol{Z}_{\boldsymbol{\Delta}}; \boldsymbol{P}, \boldsymbol{Q}) := \frac{1}{N} \boldsymbol{P} \boldsymbol{Z}_{\boldsymbol{\Delta}} \boldsymbol{M} \boldsymbol{Z}_{\boldsymbol{\Delta}}^{\top} \boldsymbol{Q} \boldsymbol{Z}_{\boldsymbol{\Delta}}, \qquad \boldsymbol{M} := \begin{bmatrix} \boldsymbol{I}_n & 0 \\ 0 & 0 \end{bmatrix} \in \mathbb{R}^{(N+1)\times(N+1)}, \tag{2}$$

where $\boldsymbol{P} \in \mathbb{R}^{(d+1)\times(d+1)}$ serves as the value weight matrix and $\boldsymbol{Q} \in \mathbb{R}^{(d+1)\times(d+1)}$ serves as the product of the key and query weight matrices. Following prior work on in-context learning (Ahn et al., 2023; Cheng et al., 2024; Gatmiry et al., 2024; Li et al., 2025), we adopt a mask matrix $\boldsymbol{M}$ to prevent the tokens from attending to the query token.

**Training Data Distribution.** The transformer is pretrained on $d$ distinct datasets. Inspired by Tsipras et al. (2019), we consider the following data structure that explicitly separates robust and non-robust features (cf. Section 2) according to their dimensional indices:

**Assumption 3.1** (Individual training data distribution). Let $c \in [d]$ be the index of a training data distribution and $\mathcal{D}_c^{\mathrm{tr}}$ be the $c$-th distribution. A sample $(\boldsymbol{x}, y) \sim \mathcal{D}_c^{\mathrm{tr}}$ satisfies the following:

$$y \sim U(\{\pm 1\}), \qquad x_c = y, \qquad \forall i \in [d], i \neq c: \ x_i \sim \begin{cases} U([0, y\lambda]) & (y = 1) \\ U([y\lambda, 0]) & (y = -1) \end{cases}, \qquad (3)$$

where $0 < \lambda < 1$. For any $i \neq j$, $x_i$ and $x_j$ are independent, given $y$.

In this distribution, each sample has a feature that is strongly correlated with its label (i.e., a robust feature) in the $c$-th dimension and weakly correlated features (i.e., non-robust features) in the remaining dimensions. The correlation between the non-robust features and the label is bounded by $\lambda$. The robust features mimic human-interpretable, semantically meaningful attributes in natural objects (e.g., shape). The non-robust features mimic human-imperceptible yet predictive attributes (e.g., texture).

**Test Data Distribution.** The test data distribution may exhibit more diverse structures than the training data distributions, and may include non-predictive features in addition to robust and non-robust features.

**Assumption 3.2** (Test data distribution). Let the index sets of robust, non-robust, and irrelevant features be $\mathcal{S}_{\mathrm{rob}}, \mathcal{S}_{\mathrm{vul}}, \mathcal{S}_{\mathrm{irr}} \subset [d]$, respectively. Suppose that these sets are disjoint, i.e., $\mathcal{S}_{\mathrm{rob}} \cap \mathcal{S}_{\mathrm{vul}} = \mathcal{S}_{\mathrm{vul}} \cap \mathcal{S}_{\mathrm{irr}} = \mathcal{S}_{\mathrm{irr}} \cap \mathcal{S}_{\mathrm{rob}} = \emptyset$ and that $\mathcal{S}_{\mathrm{rob}} \cup \mathcal{S}_{\mathrm{vul}} \cup \mathcal{S}_{\mathrm{irr}} = [d]$. Let the number of robust, non-robust, and irrelevant features be $d_{\mathrm{rob}} := |\mathcal{S}_{\mathrm{rob}}|$, $d_{\mathrm{vul}} := |\mathcal{S}_{\mathrm{vul}}|$, and $d_{\mathrm{irr}} := |\mathcal{S}_{\mathrm{irr}}|$, respectively. Let the scales of the robust, non-robust, and irrelevant features be $\alpha > 0$, $\beta > 0$, and $\gamma \geq 0$, respectively. Let $\mathcal{D}^{\mathrm{te}}$ be a test data distribution. A sample $(\boldsymbol{x}, y) \sim \mathcal{D}^{\mathrm{te}}$ satisfies the following:

(1. Label) The label $y$ follows the uniform distribution $U(\{\pm 1\})$.

(2. Expectation and Moments) For every $i \in \mathcal{S}_{\mathrm{irr}}$, $\mathbb{E}[x_i] = 0$. For every $i \in [d]$ and $n \in \{2, 3, 4\}$, there exist constants $C_i > 0$ and $C_{i,n} \geq 0$ such that

$$\mathbb{E}[yx_i] = \begin{cases} C_i\alpha & (i \in \mathcal{S}_{\mathrm{rob}}) \\ C_i\beta & (i \in \mathcal{S}_{\mathrm{vul}}) \ , \\ 0 & (i \in \mathcal{S}_{\mathrm{irr}}) \end{cases} \qquad |\mathbb{E}[(yx_i - \mathbb{E}[yx_i])^n]| \leq \begin{cases} C_{i,n}\alpha^n & (i \in \mathcal{S}_{\mathrm{rob}}) \\ C_{i,n}\beta^n & (i \in \mathcal{S}_{\mathrm{vul}}) \ . \\ C_{i,n}\gamma^n & (i \in \mathcal{S}_{\mathrm{irr}}) \end{cases} \qquad (4)$$

(3. Covariance) There exist constants $0 \leq q_{\mathrm{rob}}, q_{\mathrm{vul}} < 1$ such that

$$\left| \{ \ i \in \mathcal{S}_{\mathrm{rob}} \ \mid \ \textstyle\sum_{j \in \mathcal{S}_{\mathrm{rob}} \cup \mathcal{S}_{\mathrm{vul}}} \mathbb{E}[(yx_i - \mathbb{E}[yx_i])(yx_j - \mathbb{E}[yx_j])] < 0 \ \} \right| \leq q_{\mathrm{rob}} d_{\mathrm{rob}}, \quad (5)$$

$$\left| \{ \ i \in \mathcal{S}_{\mathrm{vul}} \ \mid \ \textstyle\sum_{j \in \mathcal{S}_{\mathrm{rob}} \cup \mathcal{S}_{\mathrm{vul}}} \mathbb{E}[(yx_i - \mathbb{E}[yx_i])(yx_j - \mathbb{E}[yx_j])] < 0 \ \} \right| \leq q_{\mathrm{vul}} d_{\mathrm{vul}}. \quad (6)$$

(4. Independence) For every $i \in \mathcal{S}_{\mathrm{irr}}$, $x_i$ is independent of $y$ and all $x_j$ for $j \neq i$.

In contrast to the training distributions, the test distribution may contain $d_{\mathrm{rob}}$ robust features and $d_{\mathrm{irr}}$ irrelevant features. These irrelevant features simulate natural noise or redundant dimensions commonly found in real-world data. For example, in MNIST (Deng, 2012), the top-left pixel is always zero and thus not predictive. Assumption 4 requires each irrelevant feature to be independent of both the label and all other features. The robust and non-robust features are not necessarily mutually independent.

Assumption 2 (Expectation) ensures that the robust and non-robust features are positively correlated with the label. Given sufficient data, it is always possible to preprocess features to positively align with their binary labels. For example, with a large $N$, this can be achieved by multiplying each feature $x_i$ by $\mathrm{sgn}(\mathbb{E}[yx_i]) \approx \mathrm{sgn}(\sum_{n=1}^{N} y_n x_{n,i})$, ensuring $\mathbb{E}[y(\mathrm{sgn}(\mathbb{E}[yx_i])x_i)] = |\mathbb{E}[yx_i]| \geq 0$.

Assumption 2 (Moments) bounds the $n$-th central moment of each feature by a constant multiple of the $n$-th power of its scale for $n \in \{2, 3, 4\}$. This condition ensures that the feature distribution does not exhibit excessively large fluctuations relative to its scale ($n = 2$), extreme asymmetry ($n = 3$), or heavy tails ($n = 4$). For example, with appropriate constant factors, exponential distributions satisfy this condition. Moreover, empirical studies suggest that pixel values (or contrasts) after typical filtering (e.g., Gabor filtering) approximately follow exponential distributions (Ruderman, 1994;

Srivastava et al., 2003). This observation suggests that filtered pixel values (and contrasts) are broadly consistent with this assumption.

Assumption 3 bounds the number of features whose total covariance with other informative features (i.e., robust and non-robust features) is negative. As stated in Theorem 3.6, we typically assume that $q_{\text{rob}}$ and $q_{\text{vul}}$ are small (but not necessarily infinitesimal). This assumption prevents unrealistic cases where useful features are overly anti-correlated with others, which could hinder learning. When all the predictive features are conditionally independent given the label, $q_{\text{rob}} = 0$ and $q_{\text{vul}} = 0$ satisfy this assumption. Empirically, $q_{\text{rob}}$ and $q_{\text{vul}}$ appear to be small in real-world datasets (cf. Fig. A2).

These conditions encompass a wide class of realistic data distributions.

- **Example 1: Training data distribution.** Each training distribution $\mathcal{D}_c^{\text{tr}}$ is a special case of the test distribution $\mathcal{D}^{\text{te}}$. Specifically, it contains $d_{\text{rob}} = 1$ robust feature with scale $\alpha \approx 1$ and $d_{\text{vul}} = d - 1$ non-robust features with scale $\beta \approx \lambda$. There are no irrelevant features, i.e., $d_{\text{irr}} = 0$. By construction and due to the properties of uniform distribution, this distribution satisfies all the conditions in Assumption 3.2.

- **Example 2: Standard distributions.** With appropriate constant factors, the test distribution class includes standard distributions, such as uniform, normal, exponential, beta, gamma, Bernoulli, binomial, etc. For example, consider the normal distribution. For $i \in \mathcal{S}_{\text{rob}}$, Assumption 2 is satisfied by setting $yx_i \sim \mathcal{N}(\alpha, \alpha^2)$ with $C_i = 1$, $C_{i,2} = 1$, $C_{i,3} = 0$, and $C_{i,4} = 3$. Assumptions 3 and 4 are satisfied when all the features are mutually independent.

- **Example 3: MNIST/Fashion-MNIST/CIFAR-10.** Empirical evidence suggests that preprocessed MNIST (Deng, 2012), Fashion-MNIST (Xiao et al., 2017), and CIFAR-10 (Krizhevsky, 2009) approximately satisfy our assumptions. Consider MNIST. Let $\{\boldsymbol{x}_n^{(0)}\}_{n=1}^N, \{\boldsymbol{x}_n^{(1)}\}_{n=1}^N \in [0,1]^{784}$ denote the samples of digits zero and one, respectively. We assign $y = 1$ to digit zero and $y = -1$ to digit one. We center the data via $\boldsymbol{x}' \leftarrow \boldsymbol{x} - \bar{\boldsymbol{x}}$ with $\bar{\boldsymbol{x}} := (1/2N) \sum_{n=1}^N (\boldsymbol{x}_n^{(0)} + \boldsymbol{x}_n^{(1)})$ and align features with the label using $\boldsymbol{x}'' \leftarrow \text{sgn}(\sum_{n=1}^N (\boldsymbol{x}_n^{(0)} - \boldsymbol{x}_n^{(1)})) \odot \boldsymbol{x}'$. In this representation, common background pixels have near-zero expectations (i.e., $\gamma \approx 0$), while discriminative pixels—such as the left and right arcs of zero or the vertical stroke of one—correlate strongly with the label (i.e., $\alpha \approx 0.2$) (cf. Fig. A2). Additionally, some pixels that are occasionally activated by atypical samples (e.g., corners activated by slanted digits) exhibit weak correlation with the label (i.e., $\beta \approx 0.01$), reflecting non-robust but predictive attributes. Empirical analysis reveals that most pixels exhibit positive total covariance with others, consistent with Assumption 3 (cf. Fig. A2). The main departure from Assumption 3.2 is that real-world datasets exhibit a gradual transition in feature robustness rather than an explicit binary separation between robust and non-robust features.

- **Example 4: Linear combination of orthonormal bases.** Under mild conditions, any distribution in which robust and non-robust directions form an orthonormal basis can be transformed into our setting via principal component analysis (cf. Appendix C).

**Adversarial Attack.** We assume that the query (test sample) $\boldsymbol{x}_{N+1}$ is subject to the adversarial perturbation $\boldsymbol{\Delta}$ constrained by the $\ell_\infty$ norm, i.e., $\|\boldsymbol{\Delta}\|_\infty \leq \epsilon$, where $\epsilon \geq 0$ denotes the perturbation budget. In practice, $\epsilon$ is chosen to be consistent with the scale of non-robust features (e.g., $\epsilon \approx \lambda$ for the training distributions and $\epsilon \approx \beta$ for the test distribution). This ensures that perturbations can manipulate non-robust features while leaving robust features intact and remaining imperceptible to humans.

**Pretraining With In-Context Loss.** For pretraining, we consider the following problem based on the in-context loss (Ahn et al., 2023; Bai et al., 2023; Mahankali et al., 2024; Zhang et al., 2024b):

$$\min_{\boldsymbol{P},\boldsymbol{Q} \in [0,1]^{(d+1) \times (d+1)}} \mathbb{E}_{c \sim U([d]), \{(\boldsymbol{x}_n, y_n)\}_{n=1}^{N+1} \overset{\text{i.i.d.}}{\sim} \mathcal{D}_c^{\text{tr}}} \left[ \max_{\|\boldsymbol{\Delta}\|_\infty \leq \epsilon} -y_{N+1} [f(\boldsymbol{Z}_{\boldsymbol{\Delta}}; \boldsymbol{P}, \boldsymbol{Q})]_{d+1, N+1} \right]. \quad (7)$$

This formulation encourages the transformer to extract robust, generalizable representations from $N$ clean in-context demonstrations and to accurately classify the adversarially perturbed query. We impose the constraint on the transformer parameters to prevent the problem from becoming ill-posed. We choose $[0,1]^d$ instead of $[-1,1]^d$ to simplify the theoretical derivation.

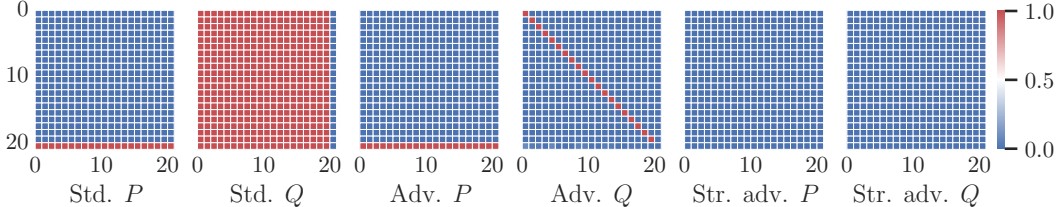

Figure 1: Parameter heatmaps learned via adversarial training (7) with $d = 20$ and $\lambda = 0.1$. For the standard, adversarial, and strong adversarial regimes, we used $\epsilon = 0$, $\frac{1+(d-1)(\lambda/2)}{d} = 0.098$, and $\frac{\lambda}{2} + \frac{3}{2}\frac{2-\lambda}{(d-1)\lambda^2+3} = 0.95$, respectively. We optimized (7) by stochastic gradient descent. Detailed experimental settings can be found in Appendix D.

## 3.2 LINEAR CLASSIFIERS AND ORACLE

**Standard Linear Classifiers Extract All Features and Are Therefore Vulnerable**. As a warm-up, consider standard training of a linear classifier parameterized by $w \in \mathbb{R}^d$ on the $c$-th training distribution $\mathcal{D}_c^{\mathrm{tr}}$. Standard training yields $w^{\mathrm{std}} := \arg\min_{w \in [0,1]^d} \mathbb{E}_{(x,y)\sim\mathcal{D}_c^{\mathrm{tr}}}[-y w^\top x] = \mathbf{1}_d$. This classifier utilizes all the features, i.e., the robust feature at the $c$-th dimension and the other non-robust features. Although $w^{\mathrm{std}}$ achieves correct predictions on clean samples ($\mathbb{E}[y w^{\mathrm{std}\top} x] > 0$), it is vulnerable to adversarial perturbations: $\mathbb{E}[\min_{\|\Delta\|_\infty \leq \epsilon} y w^{\mathrm{std}\top}(x + \Delta)] \leq 0$ for $\epsilon \geq \frac{1+(d-1)(\lambda/2)}{d}$.[1] This implies that when $d$ is small, the perturbation must satisfy $\epsilon \gtrsim 1$ to flip the prediction. In this regime, the perturbation alters the robust feature and is no longer human-imperceptible. However, as $d$ increases, the threshold decreases to $\epsilon \gtrsim \lambda$, which matches the scale of the non-robust features. Such perturbations are human-imperceptible yet sufficient to cause misclassification.

**Linear Classifiers Can Be Specifically Robust but Not Universally Robust.** Consider adversarial training: $\min_{w \in [0,1]^d} \mathbb{E}[\max_{\|\Delta\|_\infty \leq \epsilon} -y w^\top(x + \Delta)]$. For $\frac{\lambda}{2} \leq \epsilon < 1$, the optimal solution $w^{\mathrm{adv}}$ equals one at the $c$-th dimension and zero otherwise. The classifier relies solely on the robust feature at the $c$-th dimension and ignores the other non-robust features. Unlike the standard model, this model can correctly classify both clean and adversarial samples for $0 \leq \epsilon < 1$; thus, linear classifiers can be robust to a specific training distribution. However, $w^{\mathrm{adv}}$ tailored to $\mathcal{D}_c^{\mathrm{tr}}$ is vulnerable on the other distributions $\mathcal{D}_{c'}^{\mathrm{tr}}$ indexed by $c' \neq c$; thus, linear classifiers cannot be universally robust.

**Universally Robust Classifiers Exist.** Although linear classifiers cannot exhibit universal robustness across all $c$, universally robust classifiers do exist. For example, the classifier $h(x) := \mathrm{sgn}(x_i)$ with $i := \arg\max_{j \in [d]} |x_j|$ always produces correct predictions for clean data $x \sim \mathcal{D}_c^{\mathrm{tr}}$ for any $c$ and perturbed data $x + \Delta$ with $\|\Delta\|_\infty \leq \frac{1}{2}$.

## 3.3 ADVERSARIAL PRETRAINING

In this section, we analyze a global minimizer of the minimization problem (7).

**Optimization Challenges.** Although the training distributions are simple, the minimization problem (7) remains nontrivial due to the non-linearity and non-convexity of the model with respect to the trainable parameters $P$ and $Q$. The non-linearity of the self-attention and the inner-maximization are also obstacles. Indeed, the minimization problem (7) can be reformulated as the following non-linear maximization problem:

**Lemma 3.3** (Transformation of original optimization problem)**.** The minimization problem (7) can be transformed into the maximization problem $\max_{b \in \{0,1\}^{d+1}} \sum_{i=1}^{d(d+1)} \max(0, \sum_{j=1}^{d+1} b_j h_{i,j})$, where $h_{i,j} \in \mathbb{R}$ is a constant depending on $(i, j)$, and there exists a mapping from $b$ to $P$ and $Q$.

The proof can be found in Appendix E. This lemma highlights the inherent difficulty of optimizing (7), as it requires selecting a binary vector $b$ that balances $d(d + 1)$ interdependent non-linear terms.

---

[1] $\mathbb{E}[\min_{\|\Delta\|_\infty \leq \epsilon} y w^{\mathrm{std}\top}(x + \Delta)] = w^{\mathrm{std}\top}(\mathbb{E}[yx] - \epsilon \mathbf{1}_d) = \{1 + (d - 1)(\lambda/2)\} - d\epsilon \leq 0$.

**Global Solution.** By exploiting the symmetric property of $\boldsymbol{b}$ and further transformation of the problem in Lemma 3.3, we identify the global solutions to (7) for certain perturbation regimes.

**Theorem 3.4** (Parameters learned via adversarial pretraining). The global minimizers of (7) are

$$\left(1.\ \text{Standard};\ \epsilon = 0\right) \quad \boldsymbol{P} = \boldsymbol{P}^{\text{std}} := \begin{bmatrix} \boldsymbol{0}_{d,d+1} \\ \boldsymbol{1}_{d+1}^{\top} \end{bmatrix} \quad \text{and} \quad \boldsymbol{Q} = \boldsymbol{Q}^{\text{std}} := [\boldsymbol{1}_{d+1,d} \quad \boldsymbol{0}_{d+1}].$$

$$\left(2.\ \text{Adversarial};\ \epsilon = \tfrac{1+(d-1)(\lambda/2)}{d}\right) \quad \boldsymbol{P} = \boldsymbol{P}^{\text{adv}} := \begin{bmatrix} \boldsymbol{0}_{d,d+1} \\ \boldsymbol{1}_{d+1}^{\top} \end{bmatrix} \quad \text{and} \quad \boldsymbol{Q} = \boldsymbol{Q}^{\text{adv}} := \begin{bmatrix} \boldsymbol{I}_d & \boldsymbol{0}_d \\ \boldsymbol{0}_d^{\top} & 0 \end{bmatrix}.$$

$$\left(3.\ \text{Strongly adversarial};\ \epsilon \geq \tfrac{\lambda}{2} + \tfrac{3}{2}\tfrac{2-\lambda}{(d-1)\lambda^2+3}\right) \quad \boldsymbol{P} = \boldsymbol{0}_{d+1,d+1} \quad \text{and} \quad \boldsymbol{Q} = \boldsymbol{0}_{d+1,d+1}.$$

The proof and optimal parameters for different $\epsilon$ can be found in Appendix E. Importantly, the optimal $\boldsymbol{P}$ and $\boldsymbol{Q}$ are independent of any specific training distribution (i.e., index $c$), reflecting that the transformer acquires learning capability from demonstrations rather than memorizing individual tasks. Experimental results obtained using gradient descent align with our theoretical predictions (Fig. 1).

**Failure Case.** In the strongly adversarial regime, the global optimum becomes $\boldsymbol{P} = \boldsymbol{Q} = \boldsymbol{0}$, causing the transformer to always output zero regardless of the input. Namely, no universally robust single-layer linear transformers exist, despite the existence of universally robust classifiers (cf. Section 3.2). The perturbation scale $\epsilon \geq \tfrac{\lambda}{2} + \tfrac{3}{2}\tfrac{2-\lambda}{(d-1)\lambda^2+3}$ decreases in $d$: it transitions from $\epsilon = 1$ when $d = 1$ to $\epsilon \to \tfrac{\lambda}{2}$ as $d \to \infty$. In moderate dimensions ($d \approx \tfrac{1}{\lambda}$), adversarial perturbations must satisfy $\epsilon \gtrsim 1$ to break the robustness. They are comparable to the scale of the robust features and thus perceptible to humans, contradicting the concept of adversarial perturbations. However, in extremely high dimensions ($d \gtrsim \tfrac{1}{\lambda^2}$), it suffices to perturb by only $\epsilon \gtrsim \lambda$, which is on the same scale as the non-robust features and is typically imperceptible, thus preserving the concept of adversarial perturbations. This can be rephrased as follows: under our training distributions, single-layer linear transformers cannot achieve universal robustness when the non-robust dimensions (i.e., $d - 1$) substantially outnumber the single robust dimension.

### 3.4   UNIVERSAL ROBUSTNESS

In this section, we show that adversarial pretraining, combined with in-context learning from clean demonstrations, can yield universal robustness on both seen and unseen distributions.

**Standard Pretraining Leads to Vulnerability.** We begin by showing that the standard model fails to classify adversarially perturbed inputs.

**Theorem 3.5** (Standard pretraining case). There exist a constant $C > 0$ and a strictly positive function $g(d_{\text{rob}}, d_{\text{vul}}, d_{\text{irr}}, \alpha, \beta, \gamma)$ such that

$$\mathbb{E}_{\{(\boldsymbol{x}_n, y_n)\}_{n=1}^{N+1} \overset{\text{i.i.d.}}{\sim} \mathcal{D}^{\text{te}}} \left[ \min_{\|\boldsymbol{\Delta}\|_\infty \leq \epsilon} y_{N+1}[f(\boldsymbol{Z}_{\boldsymbol{\Delta}}; \boldsymbol{P}^{\text{std}}, \boldsymbol{Q}^{\text{std}})]_{d+1,N+1} \right]$$

$$\leq\ g(d_{\text{rob}}, d_{\text{vul}}, d_{\text{irr}}, \alpha, \beta, \gamma)\ \Big\{ \underbrace{C(d_{\text{rob}}\alpha + d_{\text{vul}}\beta)}_{\text{Prediction for original data}} - \underbrace{(d_{\text{rob}} + d_{\text{vul}} + d_{\text{irr}})\epsilon}_{\text{Adversarial effect}} \Big\}. \quad (8)$$

The proof can be found in Appendix F. This result analyzes the expectation of the product of the true label and the model prediction for the query. A positive value indicates correct classification, whereas a nonpositive value indicates misclassification. Since $g(d_{\text{rob}}, d_{\text{vul}}, d_{\text{irr}}, \alpha, \beta, \gamma)$ is always positive, when $C(d_{\text{rob}}\alpha + d_{\text{vul}}\beta) - (d_{\text{rob}} + d_{\text{vul}} + d_{\text{irr}})\epsilon$ is nonpositive, this implies incorrect classification.

*Standard models extract both robust and non-robust features and thus are vulnerable.* Assume $d_{\text{irr}} = 0$. Like standard linear classifiers, the standard model leverages both robust features $d_{\text{rob}}\alpha$ and non-robust features $d_{\text{vul}}\beta$. This also leads to vulnerability to adversarial perturbations, $-(d_{\text{rob}} + d_{\text{vul}})\epsilon$. The prediction becomes incorrect when $C(d_{\text{rob}}\alpha + d_{\text{vul}}\beta) - (d_{\text{rob}} + d_{\text{vul}})\epsilon \leq 0$, i.e., when $\epsilon \gtrsim \tfrac{d_{\text{rob}}\alpha + d_{\text{vul}}\beta}{d_{\text{rob}} + d_{\text{vul}}}$. When the perturbation size $\epsilon$ is on the same scale as the non-robust features ($\epsilon \lesssim \beta$),

the inequality can be rearranged as $d_{\text{vul}} \gtrsim \frac{\alpha-\beta}{\beta} d_{\text{rob}}$. In typical cases where the scale of the robust features is much larger than that of the non-robust features ($\alpha \gg \beta$), we can informally conclude:

(**Informal restatement of Theorem 3.5**) Assume that the scale of the robust features is much larger than that of the non-robust features ($\alpha \gg \beta$), the perturbation size is on the same scale as the non-robust features ($\epsilon \lesssim \beta$), and there are no non-predictive features ($d_{\text{irr}} = 0$). If $d_{\text{vul}} \gtrsim \frac{\alpha}{\beta} d_{\text{rob}}$, then the standardly pretrained single-layer linear transformer is vulnerable to adversarial attacks.

*Non-predictive features accelerate vulnerability.* Redundant dimensions $d_{\text{irr}}$ do not contribute to the first term (i.e., accuracy) but increase the second term (i.e., vulnerability). Therefore, they degrade robustness without improving predictive performance. In addition, $d_{\text{irr}}$ amplifies the adversarial effect at a rate of $d_{\text{irr}}\epsilon$, which is comparable to the effect of the useful dimensions, $d_{\text{rob}}\epsilon$ and $d_{\text{vul}}\epsilon$.

**Adversarial Pretraining Leads to Universal Robustness.** We now establish universal robustness of the adversarially pretrained model.

**Theorem 3.6** (Adversarial pretraining case). Suppose that $q_{\text{rob}}$ and $q_{\text{vul}}$ defined in Assumption 3.2 are sufficiently small. There exist constants $C_1, C_2 > 0$ such that

$$\mathbb{E}_{\{(\boldsymbol{x}_n,y_n)\}_{n=1}^{N+1} \overset{\text{i.i.d.}}{\sim} \mathcal{D}^{\text{te}}} \left[ \min_{\|\boldsymbol{\Delta}\|_\infty \leq \epsilon} y_{N+1}[f(\boldsymbol{Z}_{\boldsymbol{\Delta}}; \boldsymbol{P}^{\text{adv}}, \boldsymbol{Q}^{\text{adv}})]_{d+1,N+1} \right]$$

$$\geq \underbrace{C_1(d_{\text{rob}}\alpha + d_{\text{vul}}\beta + 1)(d_{\text{rob}}\alpha^2 + d_{\text{vul}}\beta^2)}_{\text{Prediction for original data}}$$

$$- \underbrace{C_2 \left\{ (d_{\text{rob}}\alpha + d_{\text{vul}}\beta + 1)\left(d_{\text{rob}}\alpha + d_{\text{vul}}\beta + \frac{d_{\text{irr}}\gamma}{\sqrt{N}}\right) + d_{\text{irr}}\left(\sqrt{\frac{d_{\text{irr}}}{N}} + 1\right)\gamma^2 \right\}\epsilon}_{\text{Adversarial effect}}. \quad (9)$$

The proof and a more general statement can be found in Appendix F and Theorem F.1, respectively. For notational simplicity, we assume small $q_{\text{rob}}$ and $q_{\text{vul}}$. However, we do not require infinitesimal $q_{\text{rob}}$ and $q_{\text{vul}}$. See Theorem F.1 and Appendix C. In contrast to Theorem 3.5, this theorem provides a lower bound. A positive right-hand side implies correct classification under adversarial perturbations.

*Adversarially trained models prioritize robust features.* Assume $d_{\text{irr}} = 0$. Up to constant factors, the lower bound reduces to $(d_{\text{rob}}\alpha + d_{\text{vul}}\beta + 1)\{d_{\text{rob}}\alpha^2 + d_{\text{vul}}\beta^2 - (d_{\text{rob}}\alpha + d_{\text{vul}}\beta)\epsilon\}$. The important factor is $d_{\text{rob}}\alpha^2 + d_{\text{vul}}\beta^2 - (d_{\text{rob}}\alpha + d_{\text{vul}}\beta)\epsilon$, which determines the sign. As shown in Theorem 3.5, the standard model extracts the robust and non-robust features at scales $d_{\text{rob}}\alpha$ and $d_{\text{vul}}\beta$, respectively. In contrast, the adversarially trained model extracts them at quadratic scales $d_{\text{rob}}\alpha^2$ and $d_{\text{vul}}\beta^2$. Since the robust features typically have larger magnitude ($\alpha^2 \gg \beta^2$), the adversarially trained model places greater emphasis on the robust features and mitigates the influence of the non-robust features, compared to its standard counterpart.

*Adversarially trained models are universally robust.* As shown above, up to constant factors, the prediction remains correct as long as $d_{\text{rob}}\alpha^2 + d_{\text{vul}}\beta^2 - (d_{\text{rob}}\alpha + d_{\text{vul}}\beta)\epsilon \geq 0$. This condition fails when $\epsilon \gtrsim \frac{d_{\text{rob}}\alpha^2 + d_{\text{vul}}\beta^2}{d_{\text{rob}}\alpha + d_{\text{vul}}\beta}$. When the perturbation size $\epsilon$ is on the same scale as the non-robust features ($\epsilon \lesssim \beta$), the inequality can be rearranged as $d_{\text{vul}} \gtrsim \frac{d_{\text{rob}}\alpha(\alpha-\beta)}{\beta^2}$. In typical cases where $\alpha \gg \beta$, we can informally conclude:

(**Informal restatement of Theorem 3.6**) Assume that the scale of the robust features is much larger than that of the non-robust features ($\alpha \gg \beta$), the perturbation size is on the same scale as the non-robust features ($\epsilon \lesssim \beta$), and there are no non-predictive features ($d_{\text{irr}} = 0$). If $d_{\text{vul}} \lesssim (\frac{\alpha}{\beta})^2 d_{\text{rob}}$, then the adversarially pretrained single-layer linear transformer is robust to adversarial attacks.

This threshold substantially improves on the standard model's robustness condition. For example, when $\alpha = 160/255$ and $\beta = 8/255$, the standard model is potentially robust up to $d_{\text{vul}} \lesssim 20d_{\text{rob}}$, whereas the adversarially pretrained model remains robust up to $d_{\text{vul}} \lesssim 400d_{\text{rob}}$. This result also suggests that the adversarially pretrained model is potentially vulnerable when the non-robust dimensions significantly outnumber the robust ones, consistent with the failure case in Section 3.3.

*Adversarially trained models are more robust to attacks that exploit non-predictive features.* Theorem 3.6 shows that even when the adversary exploits redundant dimensions, their effect is significantly attenuated. For simplicity, assume $N \to \infty$. The adversarial effect from the irrelevant features scales as $d_{\text{irr}}\gamma^2\epsilon$, which is linear in $d_{\text{irr}}$. In contrast, the clean term scales as $d_{\text{rob}}^2\alpha^3$ and $d_{\text{vul}}^2\beta^3$, i.e., quadratically in the number of the informative features. Thus, as long as the informative features dominate in magnitude and number, the influence of the non-predictive features on the model's robustness remains limited.

## 3.5 OPEN CHALLENGES

In this section, we show that two open challenges in robust classification (Schmidt et al., 2018; Tsipras et al., 2019) persist in our setting.

**Accuracy–Robustness Trade-Off.** Inspired by Tsipras et al. (2019), we consider a situation where robust features correlate with their labels with some probability, whereas non-robust features always correlate.

**Theorem 3.7** (Accuracy–robustness trade-off). Assume $d_{\text{rob}} = 1$, $d_{\text{vul}} = d - 1$, and $d_{\text{irr}} = 0$. In addition to Assumption 3.2, for $(\boldsymbol{x}, y) \sim \mathcal{D}^{\text{te}}$, suppose that $yx_i$ takes $\alpha$ with probability $p > 0.5$ and $-\alpha$ with probability $1 - p$ for $i \in \mathcal{S}_{\text{rob}}$. Moreover, $yx_i$ takes $\beta$ with probability one for $i \in \mathcal{S}_{\text{vul}}$. Define $\tilde{f}(\boldsymbol{P}, \boldsymbol{Q}) := \mathbb{E}_{\{(\boldsymbol{x}_n, y_n)\}_{n=1}^N \overset{\text{i.i.d.}}{\sim} \mathcal{D}^{\text{te}}}[y_{N+1}[f(\boldsymbol{Z_0}; \boldsymbol{P}, \boldsymbol{Q})]_{d+1, N+1}]$. Then, there exist strictly positive functions $g_1(d, \alpha, \beta)$ and $g_2(d, \alpha, \beta)$ such that

$$\tilde{f}(\boldsymbol{P}^{\text{std}}, \boldsymbol{Q}^{\text{std}}) = \begin{cases} g_1(d, \alpha, \beta)(\alpha + (d-1)\beta) & (\text{w.p.} \quad p) \\ g_1(d, \alpha, \beta)(-\alpha + (d-1)\beta) & (\text{w.p.} \quad 1 - p) \end{cases}, \tag{10}$$

$$\tilde{f}(\boldsymbol{P}^{\text{adv}}, \boldsymbol{Q}^{\text{adv}}) \leq g_2(d, \alpha, \beta)\{-(2p-1)\alpha^2 + (d-1)\beta^2\} \quad (\text{w.p.} \quad 1 - p). \tag{11}$$

The proof can be found in Appendix G. Unlike Theorems 3.5 and 3.6, this theorem considers the expectation over $\{(\boldsymbol{x}_n, y_n)\}_{n=1}^{N+1}$, instead of $\{(\boldsymbol{x}_n, y_n)\}_{n=1}^N$. The query $(\boldsymbol{x}_{N+1}, y_{N+1})$ is stochastic. If $d \gtrsim \frac{\alpha}{\beta}$, the standard model consistently produces correct predictions. However, if $d \lesssim (2p-1)(\frac{\alpha}{\beta})^2$, the adversarially trained model produces incorrect predictions with probability $1 - p$. This trade-off arises because the adversarially trained model discards the non-robust but predictive features.

**Need for Larger In-Context Sample Sizes.** We informally summarize Theorem H.1 as follows (omitting constant factors for clarity):

**(Informal summary of Theorem H.1)** Assume the same conditions as in Theorem 3.7. Consider $\mathbb{E}_{\boldsymbol{x}_{N+1}, y_{N+1}}[y_{N+1}[f(\boldsymbol{Z_0}; \boldsymbol{P}, \boldsymbol{Q})]_{d+1, N+1}]$. Assume $d \lesssim \frac{\alpha}{\beta}$, $p \to 0.5$, and a small $N$ regime. With probability at least $1 - \exp(-N)$, the standard model makes correct predictions. With probability at most $1 - \frac{1}{\sqrt{N}}$, the adversarially trained model makes correct predictions.

This result indicates that in low-sample regimes, the adversarially pretrained model requires substantially more in-context demonstrations to achieve clean accuracy comparable to that of the standard model. This stems from the model's reliance on the robust features, which are statistically underrepresented in small-sample regimes.

## 4 EXPERIMENTAL RESULTS

Additional results and detailed experimental settings are provided in Appendix D.

**Verification of Theorem 3.4.** We trained single-layer linear transformers (2) using stochastic gradient descent over $[0, 1]^d$ with the in-context loss (7). The training distribution was configured with $d = 20$ and $\lambda = 0.1$. We used $\epsilon = 0$, $\frac{1+(d-1)(\lambda/2)}{d} = 0.098$, and $\frac{\lambda}{2} + \frac{3}{2}\frac{2-\lambda}{(d-1)\lambda^2+3} = 0.95$ for the standard, adversarial, and strongly adversarial regimes, respectively. The heatmaps of the learned parameters are shown in Fig. 1. These results align with the theoretical predictions of Theorem 3.4.

**Verification of Theorems 3.5 to 3.7.** We evaluated the standardly and adversarially pretrained single-layer linear transformers with the theoretically predicted parameters (i.e., the parameters in

Table 1: Accuracy (%) of standardly and adversarially pretrained single-layer linear transformers. Left values represent clean accuracy. Right values represent robust accuracy. For $\mathcal{D}^{\mathrm{tr}}$ (cf. Assumption 3.1), we used $d = 100$ and $\lambda = 0.1$. For $\mathcal{D}^{\mathrm{te}}$ (cf. Assumption 3.2), we constructed the test distribution from multivariate normal distributions with $d_{\mathrm{rob}} = 10$, $d_{\mathrm{vul}} = 90$, $d_{\mathrm{irr}} = 0$, $\alpha = 1.0$, and $\beta = 0.1$. For the real-world datasets, the values were averaged across all 45 binary classification pairs from the 10 classes. The perturbation budgets were set as follows: $\epsilon = 0.15$ for $\mathcal{D}^{\mathrm{tr}}$, 0.2 for $\mathcal{D}^{\mathrm{te}}$, 0.1 for MNIST and CIFAR-10, and 0.15 for Fashion-MNIST. See Appendix D for details.

| | $\mathcal{D}^{\mathrm{tr}}$ | $\mathcal{D}^{\mathrm{te}}$ | MNIST | FMNIST | CIFAR10 |
|---|---|---|---|---|---|
| Standardly pretrained model | **100** / 0 | **100** / 0 | **94** / 4 | **91** / 20 | **68** / 21 |
| Adversarially pretrained model | **100** / **100** | 99 / **95** | 93 / **72** | 89 / **62** | 64 / **34** |

the standard regime and adversarial regime in Theorem 3.4) on $\mathcal{D}^{\mathrm{tr}}$, $\mathcal{D}^{\mathrm{te}}$, MNIST (Deng, 2012), Fashion-MNIST (Xiao et al., 2017), and CIFAR-10 (Krizhevsky, 2009). These results are provided in Tab. 1. The results suggest that the standard models achieve high clean accuracy but suffer severe degradation under adversarial attacks, consistent with Theorem 3.5. In contrast, the adversarially pretrained models maintain high robustness, supporting Theorem 3.6, while their clean accuracy is lower, aligning with the accuracy–robustness trade-off described in Theorem 3.7.

## 5 Conclusion and Limitations

We theoretically demonstrated that single-layer linear transformers, after adversarial pretraining across classification tasks, can robustly adapt to previously unseen classification tasks through in-context learning, without any additional training. These results pave the way for universally robust foundation models. We also showed that these transformers can adaptively focus on robust features, exhibit the accuracy–robustness trade-off, and require a larger number of in-context demonstrations.

Our limitations include assumptions about data distributions and architectures. While we assume that the data distributions consist of clearly separated robust and non-robust features, real-world datasets typically exhibit a more gradual transition (cf. Section 3.1, especially Example 3). Single-layer linear transformers lack the practical characteristics of multi-layer models and softmax attention. Although these theoretical assumptions are standard and comparable in strength to those in prior work (cf. studies on in-context learning in Appendix A), they limit the applicability of our results.

Universally robust foundation models are conceptually expected to adapt to any task and any form of perturbations. However, our theoretical results assume classification tasks and $\ell_\infty$ perturbations. Extending these results to other tasks and perturbation models is left for future work.

The cost of adversarial pretraining is also a limitation of universally robust foundation models. We expect that such efforts will be undertaken by large organizations, which could offset development costs through API fees. In addition, acceleration techniques for adversarial training, which have been extensively studied in the literature, can reduce this cost to a level comparable to standard training. Our theoretical analysis is an important first step toward fostering the practical development of universally robust foundation models. See also the last paragraph in Section 1.

### Reproducibility statement

All experimental procedures are described in Section 4 and Appendix D. The source code to reproduce our experimental results can be found in `https://github.com/s-kumano/universally-robust-in-context-learner`. Proofs of the theorems are provided in Appendices E to H.

### The Use of Large Language Models (LLMs)

We used LLMs to improve our writing. No essential contributions were made by the LLMs.

ACKNOWLEDGMENTS AND DISCLOSURE OF FUNDING

S. Kumano was supported by JSPS KAKENHI Grant Number JP23KJ0789 and by JST, ACT-X Grant Number JPMJAX23C7, JAPAN. H. Kera was supported by JST PRESTO Grant Number JPMJPR24K, JST BOOST Program Grant Number JPMJBY24C6, and JSPS Program for Forming Japan's Peak Research Universities (J-PEAKS) Grant Number JPJS00420230002.

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

## A   ADDITIONAL RELATED WORK

**In-Context Learning.**    In-context learning has emerged as a remarkable property of large language models, enabling them to adapt to a new task from a few input–output demonstrations without any parameter updates (Brown et al., 2020). Recent work has shown that in-context learning can implement various algorithms (Bai et al., 2023; Garg et al., 2022). One research direction has linked in-context learning with preconditioned gradient descent through empirical (Akyürek et al., 2023; Dai et al., 2023; Garg et al., 2022; Von Oswald et al., 2023; 2024) and theoretical analyses (Ahn et al., 2023; Bai et al., 2023; Cheng et al., 2024; Gatmiry et al., 2024; Mahankali et al., 2024; Zhang et al., 2024b). Additional results have indicated that in-context learning can implement ridge regression (Akyürek et al., 2023; Bai et al., 2023), second-order optimization (Fu et al., 2024; Giannou et al., 2024), reinforcement learning (Lee et al., 2023; Lin et al., 2024), and Bayesian model averaging (Zhang et al., 2023). In terms of robustness, some studies have shown that in-context learning can act as a nearly optimal predictor under noisy linear data (Bai et al., 2023) and noisy labels (Frei & Vardi, 2025). Moreover, it has been demonstrated that in-context learning is robust to shifts in the query distribution (Wies et al., 2023; Zhang et al., 2024b), but not necessarily to shifts in the context (Shi et al., 2023; 2024; Wei et al., 2023b; Zhang et al., 2024b). In this study, we focus on the adversarial robustness of in-context learning, rather than the underlying algorithms or its robustness to random noise and distribution shifts. Specifically, we examine whether a single adversarially pretrained transformer can robustly adapt to a broad range of tasks through in-context learning.

**Norm- and Token-Bounded Adversarial Examples.**    Adversarial examples were originally introduced as subtle perturbations to natural data, designed to induce misclassifications in models (Croce & Hein, 2020; Goodfellow et al., 2015; Madry et al., 2018; Szegedy et al., 2014). These perturba-

tions are typically constrained by a norm-based distance from the original inputs. The robustness of transformers to such norm-bounded adversarial examples has been studied primarily in vision transformers (Dosovitskiy et al., 2021). Several studies have shown that standard vision transformers are as vulnerable to these attacks as conventional vision models (Bai et al., 2021; Mahmood et al., 2021), though some have reported marginal differences (Aldahdooh et al., 2021; Benz et al., 2021; Bhojanapalli et al., 2021; Naseer et al., 2021; Paul & Chen, 2022; Shao et al., 2022; Tang et al., 2021). In contrast, adversarial attacks on language models are often neither norm-constrained nor imperceptible to humans. They involve substantial token modifications (Garg & Ramakrishnan, 2020; Jin et al., 2020; Li et al., 2020; Zang et al., 2020), the insertion of adversarial tokens (Liu et al., 2024; Shen et al., 2024; Wallace et al., 2019; Wei et al., 2023a; Zou et al., 2023), and the construction of entirely new adversarial prompts (Carlini et al., 2021; 2022; Nasr et al., 2023; Perez & Ribeiro, 2022; Wei et al., 2023a). These attacks aim not only to induce misclassification (Garg & Ramakrishnan, 2020; Jin et al., 2020; Li et al., 2020; Wallace et al., 2019; Zang et al., 2020), but also to provoke objectionable outputs (Liu et al., 2024; Perez & Ribeiro, 2022; Shen et al., 2024; Wei et al., 2023a; Zou et al., 2023) or to extract private information from training data (Carlini et al., 2021; 2022; Nasr et al., 2023). They are generally bounded by token-level metrics (e.g., the number of modified tokens). In this study, we focus exclusively on norm-bounded adversarial examples. Token-bounded ones are out of scope.

**Adversarial Training.**   Adversarial training, which augments training data with adversarial examples, is one of the most effective adversarial defenses (Goodfellow et al., 2015; Madry et al., 2018). Although originally developed for conventional neural architectures, adversarial training has also proven effective for transformers (Debenedetti et al., 2023; Liu et al., 2025; Shao et al., 2022; Tang et al., 2021; Wu et al., 2022). A major limitation of adversarial training is its high computational cost. To address this, several methods have focused on more efficient generation of adversarial examples (Andriushchenko & Flammarion, 2020; Kim et al., 2021; Park & Lee, 2021; Shafahi et al., 2019; Wong et al., 2020; Zhang et al., 2019a) and adversarial finetuning of standard pretrained models (Jeddi et al., 2020; Mao et al., 2023; Suzuki et al., 2023; Wang et al., 2024a). More recently, researchers have introduced adversarial prompt tuning, which trains visual (Mao et al., 2023; Wang et al., 2024b), textual (Fan et al., 2024; Li et al., 2024; Zhang et al., 2024a), or bimodal prompts (Jia et al., 2025; Luo et al., 2024; Yang et al., 2024; Zhou et al., 2024) in an adversarial manner. However, these methods require retraining for each task. In this study, we explore the potential of adversarially pretrained transformers for robust task adaptation via in-context learning, thereby eliminating the task-specific retraining and associated computational overhead.

**Adversarial Meta-Learning.**   Adversarial meta-learning seeks to develop a universally robust meta-learner that can swiftly and reliably adapt to new tasks under adversarial conditions. Existing approaches adversarially train a neural network on multiple tasks, and then finetune it on a target task using clean (Goldblum et al., 2020; Hou et al., 2021; Liu et al., 2021; Wang et al., 2021; Yin et al., 2018) or adversarial samples (Yin et al., 2018). In this study, we similarly aim to train such a meta-learner. However, rather than relying on neural networks and finetuning, we employ a transformer as the meta-learner and leverage its in-context learning ability for task adaptation.

**Related but Distinct Work.**   We here review theoretical work on the adversarial robustness of in-context learning. Assuming token-bounded adversarial examples, prior studies have shown that even a single token modification in the context can significantly alter the output of a standardly trained model on a clean query (Anwar et al., 2024), and deeper layers can mitigate this (Li et al., 2025). Assuming norm- and token-bounded examples, Fu et al. have shown that adversarial training with short adversarial contexts can provide robustness against longer ones (Fu et al., 2025). They considered a clean query and adversarial tokens appended to the original context. In this study, we explore how adversarially trained models handle norm-bounded perturbations to a query in a clean context. As a result, we reveal their universal robustness that can be generalized to a new task from a few demonstrations.

## B  CLARIFICATION ON SINGLE-TASK PRETRAINING AND TASK-SPECIFIC ADVERSARIAL TRAINING

**Single-Task Adversarial Training and In-Context Learning.**  We should clarify that a model trained on a single task, unlike one pretrained on multiple tasks (i.e., our main setting), lacks in-context learning capability. Namely, an approach that combines adversarial training (not adversarial pretraining) and in-context learning is not feasible. Specifically, the parameters of a model trained on a single task differ significantly from those on multiple tasks (shown in Theorem 3.4). Such models cannot provide correct answers for new tasks via in-context learning even in standard settings (without adversaries). The interesting property of transformers is that when trained on multiple tasks rather than a single one—as with large language models—they develop distinctly different parameters that enable in-context learning capability.

**Performance Comparison with Task-Specific Adversarial Training.**  As the no-free-lunch theorem indicates, task-specific approaches achieve higher performance than the approach that combines adversarial pretraining and in-context learning. Specifically, the following holds in terms of robust accuracy: (1) task-specific adversarially trained models $\geq$ (2) adversarially pretrained models with in-context learning $\gg$ (3) task-specific standardly trained models $\approx$ (4) standardly pretrained models with in-context learning $\approx 0\%$. More precisely, Models (2) have the limitation of robustness as predicted in Theorems 3.4 and 3.6. However, if training and test distributions match, Models (1) do not. However, we emphasize that Models (1) require users to perform adversarial training for each individual task and cannot generalize to test distributions other than the one it was trained on. This makes it unsuitable for our research focus on universally robust foundation models that can generalize across a wide range of tasks without task-specific adversarial training.

## C  ADDITIONAL THEORETICAL SUPPORT AND INSIGHTS

### C.1  LINEAR COMBINATION OF ORTHONORMAL BASES CAN BE TRANSFORMED INTO OUR TEST DISTRIBUTION.

Our test data distribution, Assumption 3.2, can implicitly represent data distributions comprising robust and non-robust directions forming an orthonormal basis. Consider $d$ orthonormal bases, $\{e_i\}_{i=1}^d$. We set $d_{\mathrm{irr}} = 0$, namely $d = d_{\mathrm{rob}} + d_{\mathrm{vul}}$. Each data point is represented as $x = c_1 e_1 + c_2 e_2 + \cdots + c_d e_d$, where coefficients $c_i$ are sampled probabilistically. These coefficients satisfy $\mathbb{E}[yc_i] = C_i\alpha$ for $i \in \mathcal{S}_{\mathrm{rob}}$ and $\beta$ for $i \in \mathcal{S}_{\mathrm{vul}}$. In addition, $|\mathbb{E}[(yc_i - \mathbb{E}[yc_i])^n]| \leq C_{i,n}\alpha^n$ for $i \in \mathcal{S}_{\mathrm{rob}}$ and $C_{i,n}\beta^n$ for $i \in \mathcal{S}_{\mathrm{vul}}$. Given a dataset of $N$ i.i.d. samples $\{(x_n, y_n)\}_{n=1}^N$, if $c_{n,i}$ is independent of $c_{n,j}$ for $i \neq j$ conditional on $y$, and $N$ is sufficiently large, then the covariance of $yx$ can be approximated as:

$$\frac{1}{N}\sum_{n=1}^N \left(y_n x_n - \sum_{k=1}^N y_k x_k\right)\left(y_n x_n - \sum_{k=1}^N y_k x_k\right)^\top$$

$$\approx \mathbb{E}[(yx - \mathbb{E}[yx])(yx - \mathbb{E}[yx])^\top] \tag{A12}$$

$$= \mathbb{E}\left[\left(\sum_{i=1}^d (y_i c_i - \mathbb{E}[yc_i])e_i\right)\left(\sum_{i=1}^d (y_i c_i - \mathbb{E}[yc_i])e_i\right)^\top\right] \tag{A13}$$

$$= \sum_{i,j=1}^d \mathbb{E}[(yc_i - \mathbb{E}[yc_i])(yc_j - \mathbb{E}[yc_j])]e_i e_j^\top \tag{A14}$$

$$= \sum_{i\in\mathcal{S}_{\mathrm{rob}}}^d C_{i,2}\alpha^2 e_i e_i^\top + \sum_{i\in\mathcal{S}_{\mathrm{vul}}}^d C_{i,2}\beta^2 e_i e_i^\top. \tag{A15}$$

This implies that through principal component analysis for $y_n x_n$, we can obtain $d$ orthonormal bases, $\{e_i\}_{i=1}^d$. By projecting a sample $x_n$ onto these bases, we obtain a transformed sample $x'_n := \{c_{n,1}, c_{n,2}, \ldots, c_{n,d}\}$. This demonstrates that when data is sampled from a distribution comprising robust and non-robust directions forming an orthonormal basis, if the coefficients are

mutually independent and the sample size is sufficiently large, we can preprocess the data to satisfy Assumption 3.2. Importantly, this preprocessing relies solely on statistics derivable from training samples.

## C.2 SUFFICIENT NUMBER OF DATASETS TO PROVIDE UNIVERSAL ROBUSTNESS

What determines the sufficient number of datasets needed to provide universal robustness to transformers? We conjecture that this may be determined by the number of robust bases. In this paper, we trained transformers using $d$ datasets. This stems from training with datasets where only one dimension is robust (in other words, datasets with a single robust basis), the number of dimensions $d$, and the assumption that all dimensions might contain robust features. If we assume that robust features never appear in the latter $d'$ dimensions, following the procedure in Appendix E, we can train robust transformers using only $d - d'$ datasets that describe the first $d - d'$ robust features. From this observation, we conjecture that the sufficient number of datasets required to provide universal robustness to transformers depends on the number of robust bases in the assumed data structure.

## C.3 EFFECTS OF $q_{\mathrm{rob}}$ AND $q_{\mathrm{vul}}$

We here analyze how $q_{\mathrm{rob}}$ and $q_{\mathrm{vul}}$ affect the robustness of adversarially trained transformer. As defined in Assumption 3.2, these parameters control the proportion of features whose total covariance with other features is negative. Theorem F.1 suggests that the transformer prediction for unperturbed data can be expressed as

$$C(d_{\mathrm{rob}}\alpha + d_{\mathrm{vul}}\beta)\big\{(1 - cq_{\mathrm{rob}})d_{\mathrm{rob}}\alpha^2 + (1 - cq_{\mathrm{vul}})d_{\mathrm{vul}}\beta^2\big\} + C'(d_{\mathrm{rob}}\alpha^2 + d_{\mathrm{vul}}\beta^2), \quad \text{(A16)}$$

where

$$c := \frac{(\max_{i \in \mathcal{S}_{\mathrm{rob}} \cup \mathcal{S}_{\mathrm{vul}}} C_i)(\max_{i \in \mathcal{S}_{\mathrm{rob}} \cup \mathcal{S}_{\mathrm{vul}}} C_{i,2})}{\min_{i \in \mathcal{S}_{\mathrm{rob}} \cup \mathcal{S}_{\mathrm{vul}}} C_i^3}. \quad \text{(A17)}$$

Examining the term $(1 - cq_{\mathrm{rob}})d_{\mathrm{rob}}\alpha^2 + (1 - cq_{\mathrm{vul}})d_{\mathrm{vul}}\beta^2$, we observe that larger values of $q_{\mathrm{rob}}$ and $q_{\mathrm{vul}}$ generally diminish the magnitude of transformer predictions. This indicates that negative correlations between features degrade the robustness of adversarially trained transformers. Additionally, the coefficient $c$ is characterized by $\max_{i \in \mathcal{S}_{\mathrm{rob}} \cup \mathcal{S}_{\mathrm{vul}}} C_{i,2}$, which represents a variance coefficient. This suggests that smaller feature variances enhance the robustness of adversarially trained transformers. For example, if each feature variance $C_{i,2}$ is sufficiently small, even $q_{\mathrm{rob}} = 1$ and $q_{\mathrm{vul}} = 1$ may be tolerated without significantly compromising robustness.

## C.4 DISADVANTAGE OF STANDARD FINETUNING: PARAMETER SELECTION PERSPECTIVE

In this study, we investigate task adaptation through in-context learning. As an alternative lightweight approach, standard finetuning—where all or part of the model parameters are updated—can also be employed. However, a key drawback of standard finetuning is that it requires parameter updates, whereas in-context learning does not. Moreover, finetuning necessitates careful selection of which parameters to update. Our analysis shows that improper parameter selection during finetuning can compromise the robustness initially established by adversarial pretraining. Consider adversarially pretrained parameters, $\boldsymbol{P}^{\mathrm{adv}}$ and $\boldsymbol{Q}^{\mathrm{adv}}$, and $\mathcal{D}_c^{\mathrm{tr}}$ as a downstream data distribution.

First, we examine the scenario where only $\boldsymbol{P}$ is updated while keeping $\boldsymbol{Q}^{\mathrm{adv}}$ fixed, formulated as:

$$\min_{\boldsymbol{P} \in [0,1]^{(d+1) \times (d+1)}} \mathbb{E}_{\{(\boldsymbol{x}_n, y_n)\}_{n=1}^{N+1} \overset{\text{i.i.d.}}{\sim} \mathcal{D}_c^{\mathrm{tr}}} \big[ -y_{N+1}[f(\boldsymbol{Z_0}; \boldsymbol{P}, \boldsymbol{Q}^{\mathrm{adv}})]_{d+1, N+1} \big]. \quad \text{(A18)}$$

In this case, as shown in the proof in Appendix E, $\boldsymbol{P} = \boldsymbol{P}^{\mathrm{std}}(= \boldsymbol{P}^{\mathrm{adv}})$ is the global solution. Consequently, as demonstrated in Theorem 3.6, the model's robustness is preserved.

Conversely, consider training $\boldsymbol{Q}$ while keeping $\boldsymbol{P}^{\mathrm{adv}}$ fixed, formulated as:

$$\min_{\boldsymbol{Q} \in [0,1]^{(d+1) \times (d+1)}} \mathbb{E}_{\{(\boldsymbol{x}_n, y_n)\}_{n=1}^{N+1} \overset{\text{i.i.d.}}{\sim} \mathcal{D}_c^{\mathrm{tr}}} \big[ -y_{N+1}[f(\boldsymbol{Z_0}; \boldsymbol{P}^{\mathrm{adv}}, \boldsymbol{Q})]_{d+1, N+1} \big]. \quad \text{(A19)}$$

In this scenario, $\boldsymbol{Q} = \boldsymbol{Q}^{\mathrm{std}}$ is the global solution. As established in Theorems 3.5, 3.7 and H.1, while this configuration enables the transformer to perform well on unperturbed queries, it fails to maintain robustness against perturbed inputs.

These findings highlight a critical insight: achieving robust task adaptation through standard finetuning requires careful parameter selection; otherwise, the pretrained model's adversarial robustness may be compromised. This parameter sensitivity represents a disadvantage compared to in-context learning, which preserves robustness without requiring parameter updates.

### C.5 NAIVE ADVERSARIAL CONTEXT MAY NOT IMPROVE ROBUSTNESS

One approach to enhancing the robustness of a standardly trained transformer is to incorporate adversarial examples into the context. In this section, we show that this is not the case in our setting. Consider the following transformer input:

$$\boldsymbol{Z}' := \begin{bmatrix} \boldsymbol{x}_1 + \boldsymbol{\Delta}_1 & \boldsymbol{x}_2 + \boldsymbol{\Delta}_2 & \cdots & \boldsymbol{x}_N + \boldsymbol{\Delta}_N & \boldsymbol{x}_{N+1} + \boldsymbol{\Delta}_{N+1} \\ y_1 & y_2 & \cdots & y_N & 0 \end{bmatrix}. \tag{A20}$$

The adversarial perturbations for the context, $\boldsymbol{\Delta}_1, \ldots, \boldsymbol{\Delta}_N$, are defined as $\boldsymbol{\Delta}_n := -\epsilon y_n \mathbf{1}_d$. In this setting, for $\epsilon \geq \frac{1+(d-1)(\lambda/2)}{d}$, the standard transformer prediction is given by:

$$\mathbb{E}_{\{(\boldsymbol{x}_n, y_n)\}_{n=1}^{N+1} \overset{\text{i.i.d.}}{\sim} \mathcal{D}_c^{\text{tr}}} \left[ \min_{\|\boldsymbol{\Delta}_{N+1}\|_\infty \leq \epsilon} y_{N+1}[f(\boldsymbol{Z}'; \boldsymbol{P}^{\text{std}}, \boldsymbol{Q}^{\text{std}})]_{d+1,N+1} \right] \leq 0. \tag{A21}$$

This result suggests that, in our setting, naive adversarial demonstrations do not improve the performance of the standard transformer. Intuitively, because adversarial training generates new adversarial examples at each step of gradient descent, fixed adversarial demonstrations may fail to counter newly generated adversarial perturbations to the query.

## D ADDITIONAL EXPERIMENTAL RESULTS

All experiments were conducted on Ubuntu 20.04.6 LTS, Intel Xeon Gold 6226R CPUs, and NVIDIA RTX 6000 Ada GPUs.

### D.1 SUPPORT FOR ASSUMPTION 3.2.

The statistics of preprocessed MNIST, Fashion-MNIST, and CIFAR-10 are provided in Fig. A2. Preprocessing was conducted as follows: (i) selection of two different classes from the ten available classes and assignment of binary labels to every sample from the training dataset, creating $\{(\boldsymbol{x}_n, y_n)\}_{n=1}^{N}$; (ii) centering the data via $\boldsymbol{x}' \leftarrow \boldsymbol{x} - \bar{\boldsymbol{x}}$ with $\bar{\boldsymbol{x}} := (1/N) \sum_{n=1}^{N} \boldsymbol{x}_n$; and (iii) aligning features with the label using $\boldsymbol{x}'' \leftarrow \text{sgn}(\sum_{n=1}^{N} y_n \boldsymbol{x}_n) \odot \boldsymbol{x}'$. These preprocessed datasets exhibit that each dimension has a positive correlation with the label and that few dimensions have negative total covariance. The main distinction from Assumption 3.2 is that their features are not clearly separated as robust or non-robust. Instead, they gradually transition from robust to non-robust characteristics.

### D.2 VERIFICATION OF THEOREM 3.4.

We trained a single-layer transformer (2) with the in-context loss (7). The training distribution was configured with $d = 20$ and $\lambda = 0.1$ in Fig. 1 and with $d = 100$ and $\lambda = 0.1$ in Fig. A3. For standard, adversarial, and strong adversarial regimes, we used $\epsilon = 0$, $\frac{1+(d-1)(\lambda/2)}{d} = 0.098$, and $\frac{\lambda}{2} + \frac{3}{2} \frac{2-\lambda}{(d-1)\lambda^2+3} = 0.95$ in Fig. 1 and $\epsilon = 0$, $\frac{1+(d-1)(\lambda/2)}{d} = 0.06$, and $\frac{\lambda}{2} + \frac{3}{2} \frac{2-\lambda}{(d-1)\lambda^2+3} = 0.77$ in Fig. A3. Optimization was conducted using stochastic gradient descent with momentum 0.9. Learning rates were set to 0.1 for all regimes in Fig. 1, and to 1.0 for standard and strong adversarial regimes and 0.2 for the adversarial regime in Fig. A3. Training ran for 100 epochs with a learning rate scheduler that multiplied the rate by 0.1 when the loss did not improve within 10 epochs. In each iteration of stochastic gradient descent, we sampled 1,000 datasets $\{(\boldsymbol{x}_n^{(c)}, y_n^{(c)})\}_{n=1}^{N+1}$ with $N = 1,000$. The distribution index $c$ was randomly sampled from $U([d])$, meaning that in each iteration, each of the 1,000 datasets may have different $c$ values. After each parameter update, we projected the parameters to $[0,1]^d$. Adversarial perturbation was calculated as $\boldsymbol{\Delta} := -\epsilon y_n \text{sgn}(\boldsymbol{P}_{d+1,\cdot} \boldsymbol{Z}_0 \boldsymbol{M} \boldsymbol{Z}_0^\top \boldsymbol{Q}_{\cdot,:d})$, which represents the optimal attack. The heatmaps of the learned parameters in Figs. 1 and A3 completely align with the theoretical predictions of Theorem 3.4.

## D.3 VERIFICATION OF THEOREMS 3.5 TO 3.7 AND H.1

We evaluated standardly and adversarially pretrained single-layer transformers on $\mathcal{D}^{\text{tr}}$, $\mathcal{D}^{\text{te}}$, the preprocessed MNIST, Fashion-MNIST, and CIFAR-10 datasets. For network parameters, we used the theoretically predicted $\boldsymbol{P}^{\text{std}}$ and $\boldsymbol{Q}^{\text{std}}$ as standard model parameters and $\boldsymbol{P}^{\text{adv}}$ and $\boldsymbol{Q}^{\text{adv}}$ as adversarially trained model parameters. This approach allowed us to circumvent the computationally expensive adversarial pretraining for every distinct $d$ setting. As described previously, our empirical results completely align with the theoretically predicted parameter configurations.

**Configuration in Figs. A4 and A5.** In Fig. A4, the basic settings were $d = 100$, $\lambda = 0.1$, $N = 1,000$, and $\epsilon = 0.15$. In Fig. A5, they were $d_{\text{rob}} = 10$, $d_{\text{vul}} = 90$, $d_{\text{irr}} = 0$, $\alpha = 1.0$, $\beta = 0.1$, $\gamma = 0.1$, and $\epsilon = 0.2$. The basic perturbation budget was set to 0.1. We considered 1,000 batches where each batch contained 1,000 in-context demonstrations (i.e., $N = 1000$), and 1,000 queries. The test distribution $\mathcal{D}^{\text{te}}$ was constructed based on normal distribution. During sampling, $yx_i$ was sampled from $\mathcal{N}(\alpha, \alpha^2)$ for $i \in \mathcal{S}_{\text{rob}}$, $\mathcal{N}(\beta, \beta^2)$ for $i \in \mathcal{S}_{\text{vul}}$, and $\mathcal{N}(0, \gamma^2)$ for $i \in \mathcal{S}_{\text{irr}}$. Each dimension is independent, given $y$.

**Configuration in Fig. A6.** The preprocessing procedure is described in Appendix D.1. As batches, we considered 45 binary class pairs from ten classes. The basic perturbation budget was set to 0.1. In the first row of Fig. A6, we used all training samples in the training dataset. As queries, we used all test samples in the test dataset.

**Analysis.** In Figs. A4 to A6, standard transformers consistently demonstrate vulnerability to adversarial attacks, whereas adversarially trained transformers maintain a certain level of robustness, validating Theorems 3.5 and 3.6. However, adversarially pretrained transformers exhibit lower clean accuracy, supporting Theorem 3.7.

In Figs. A4 and A5, we observe that a larger number of vulnerable dimensions increases model vulnerability. Conversely, Fig. A5 shows that a larger number of robust dimensions enhances model robustness. Robust models are less susceptible to increasing vulnerable dimensions and benefit more from increasing robust dimensions.

Additionally, as predicted in Theorems 3.5 and 3.6, standard training exhibits vulnerability to increasing redundant dimensions, which is more detrimental than the harmful effect from increasing vulnerable dimensions, since redundant dimensions do not benefit predictions and are only harmful for robustness. In contrast, adversarially trained transformers exhibit significant resistance to increases in these dimensions.

The second row of Fig. A6 indicates that standard transformers still achieve high classification accuracy in small demonstration regimes, whereas adversarially trained transformers show degraded performance. These results align with our theoretical predictions, Theorem H.1.

## E  PROOF OF LEMMA 3.3 AND THEOREM 3.4 (PRETRAINING)

**Lemma 3.3** (Transformation of original optimization problem)**.** The minimization problem (7) can be transformed into the maximization problem $\max_{\boldsymbol{b} \in \{0,1\}^{d+1}} \sum_{i=1}^{d(d+1)} \max(0, \sum_{j=1}^{d+1} b_j h_{i,j})$, where $h_{i,j} \in \mathbb{R}$ is a constant depending on $(i, j)$, and there exists a mapping from $\boldsymbol{b}$ to $\boldsymbol{P}$ and $\boldsymbol{Q}$.

*Proof.* See "Overview" below in detail. The proof sketch is as follows:

Let us simplify the transformer definition (2) as $f(x; \theta_1, \theta_2) := \theta_1 x^2 \theta_2 (x + \Delta)$, where $\theta_1, \theta_2 \in \{0, 1\}$. Then the optimization problem (7) becomes

$$\min_{\theta_1, \theta_2 \in \{0,1\}} \max_{\Delta} -yf(x + \Delta; \theta_1, \theta_2) = \min_{\theta_1, \theta_2 \in \{0,1\}} \max_{\Delta} -y(\theta_1 x^2 \theta_2 (x + \Delta)). \tag{A22}$$

This can be transformed to:

$$\min_{\theta_1, \theta_2 \in \{0,1\}} \max_{\Delta} -y(\theta_1 x^2 \theta_2 (x + \Delta)) = \max_{\theta_1, \theta_2 \in \{0,1\}} \min_{\Delta} \theta_1 (yx^2 \theta_2 (x + \Delta)). \tag{A23}$$

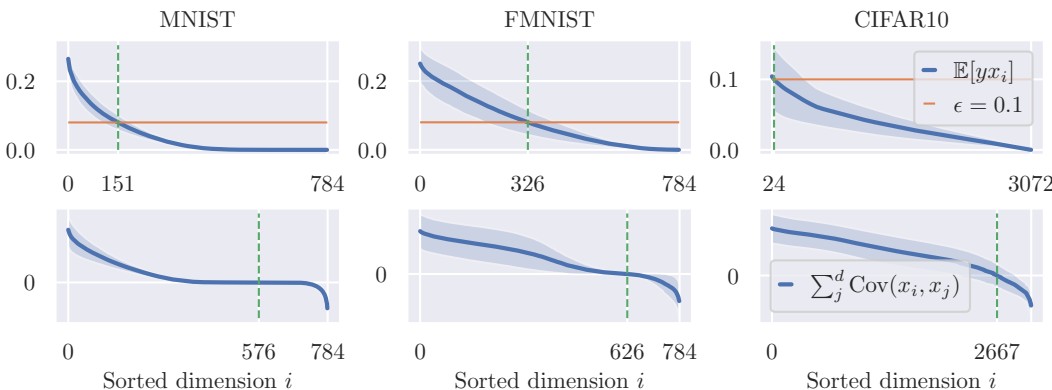

Figure A2: Statistical properties of preprocessed MNIST, Fashion-MNIST, and CIFAR-10 datasets. **First row:** Blue lines represent the mean of $(1/N)\sum_{n=1}^{N} y_n \boldsymbol{x}_n$ across 45 binary class pairs and shaded regions represent the sample standard deviation. Orange lines represent typical perturbation magnitude. Green dashed lines represent the (pseudo) threshold between robust and non-robust dimensions. **Second row:** Blue lines represent the total covariance of each dimension with other dimensions and shaded regions represent sample standard deviation across the 45 binary class pairs. Green dashed lines represent the boundary between positive and negative total covariance.

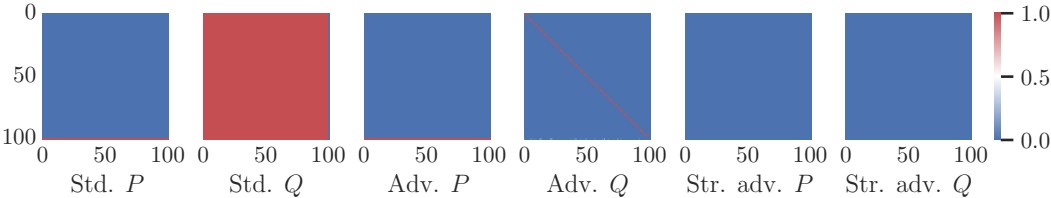

Figure A3: Parameter heatmaps induced by adversarial training (7) with $d = 100$ and $\lambda = 0.1$. For the standard, adversarial, and strong adversarial regimes, we used $\epsilon = 0$, $\frac{1+(d-1)(\lambda/2)}{d} = 0.06$, and $\frac{\lambda}{2} + \frac{3}{2}\frac{2-\lambda}{(d-1)\lambda^2+3} = 0.77$, respectively. We optimized (7) by stochastic gradient descent.

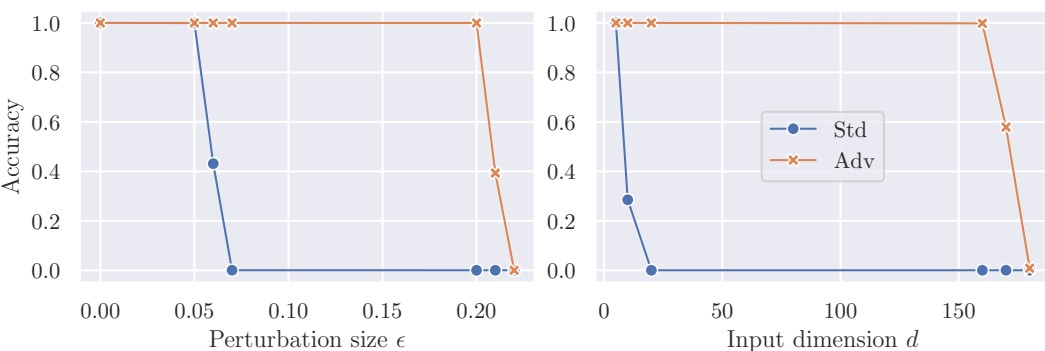

Figure A4: Accuracy (%) of standardly and adversarially pretrained single-layer transformers. Lines represent mean accuracy across batches and shaded regions represent unbiased standard deviation (notably small in magnitude). We used 1,000 batches, each containing 1,000 in-context demonstrations ($N = 1000$) and 1,000 query examples. Base configuration parameters were $d = 100$, $\lambda = 0.1$, and $\epsilon = 0.15$.

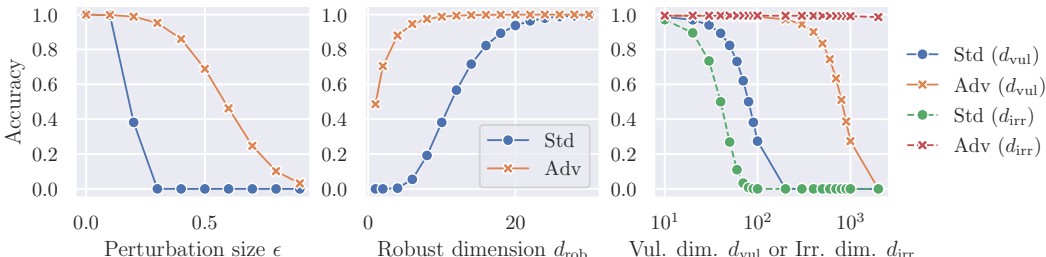

Figure A5: Accuracy (%) of standardly and adversarially pretrained single-layer transformers. Lines represent mean accuracy across batches and shaded regions represent unbiased standard deviation. We used 1,000 batches, each containing 1,000 in-context demonstrations ($N = 1000$) and 1,000 query examples. Base configuration parameters were $d_{\text{rob}} = 10$, $d_{\text{vul}} = 90$, $d_{\text{irr}} = 0$, $\alpha = 1.0$, $\beta = 0.1$, $\gamma = 0.1$, and $\epsilon = 0.2$.

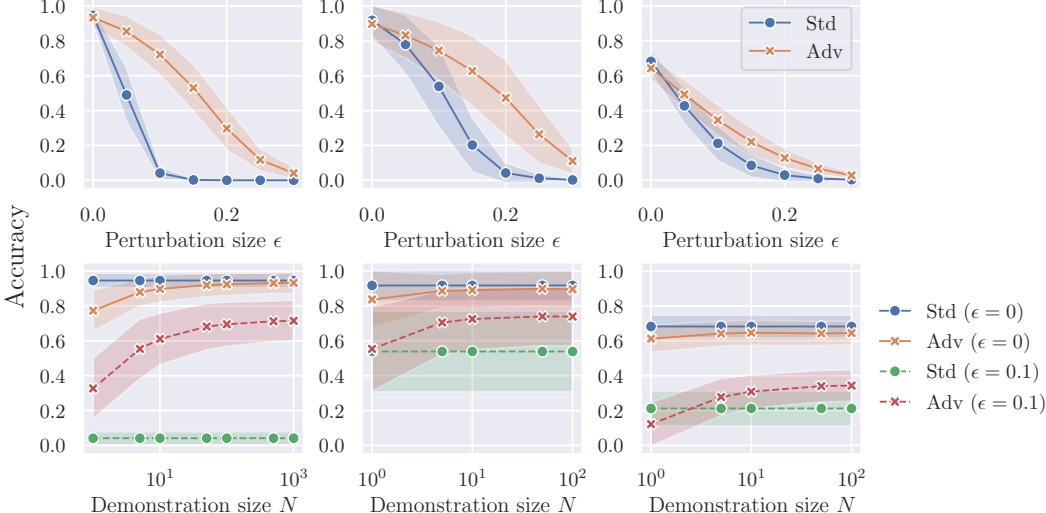

Figure A6: Accuracy (%) of standardly and adversarially pretrained single-layer transformers. Lines represent mean accuracy across 45 binary classification tasks (derived from all possible pairs of the ten classes) and shaded regions represent the unbiased standard deviation. The perturbation size was basically $\epsilon = 0.1$.

Since $\theta_1$ takes only 0 or 1, the optimal strategy is always $\theta_1 = 1$ when $yx^2\theta_2(x + \Delta)$ is positive and $\theta_1 = 0$ when negative. This transforms the problem to:

$$\max_{\theta_1,\theta_2 \in \{0,1\}} \min_\Delta \theta_1(yx^2\theta_2(x + \Delta)) = \max_{\theta_2 \in \{0,1\}} \max(0, \min_\Delta yx^2\theta_2(x + \Delta)). \qquad (A24)$$

Denoting all terms except $\theta_2$ as $h$:

$$\max_{\theta_2 \in \{0,1\}} \max(0, \min_\Delta yx^2\theta_2(x + \Delta)) = \max_{\theta_2 \in \{0,1\}} \max(0, \theta_2 h). \qquad (A25)$$

This provides intuition for the problem $\max_{b \in \{0,1\}} \sum_{i=1}^{d(d+1)} \max(0, \sum_{j=1}^{d+1} b_j h_{i,j})$ in Lemma 3.3.

$\square$

**Theorem 3.4** (Parameters learned via adversarial pretraining). The global minimizers of (7) are

$$\left(\text{1. Standard; } \epsilon = 0\right) \quad P = P^{\text{std}} := \begin{bmatrix} \mathbf{0}_{d,d+1} \\ \mathbf{1}_{d+1}^\top \end{bmatrix} \quad \text{and} \quad Q = Q^{\text{std}} := [\mathbf{1}_{d+1,d} \quad \mathbf{0}_{d+1}].$$

$$\left(\text{2. Adversarial; } \epsilon = \frac{1+(d-1)(\lambda/2)}{d}\right) \quad P = P^{\text{adv}} := \begin{bmatrix} \mathbf{0}_{d,d+1} \\ \mathbf{1}_{d+1}^\top \end{bmatrix} \quad \text{and} \quad Q = Q^{\text{adv}} := \begin{bmatrix} I_d & \mathbf{0}_d \\ \mathbf{0}_d^\top & 0 \end{bmatrix}.$$

$$\left(\text{3. Strongly adversarial; } \epsilon \geq \frac{\lambda}{2} + \frac{3}{2}\frac{2-\lambda}{(d-1)\lambda^2+3}\right) \quad P = \mathbf{0}_{d+1,d+1} \quad \text{and} \quad Q = \mathbf{0}_{d+1,d+1}.$$

*Proof.* This is the special case of the following theorem. $\square$

**Theorem E.1** (General case of Theorem 3.4). The global minimizer of (7) is as follows:

- If

$$0 \leq \epsilon \leq \frac{\lambda(\lambda(d-2)+4)}{2(\lambda(d-1)+2)}, \qquad (A26)$$

then $P = \begin{bmatrix} \mathbf{0}_{d,d+1} \\ \mathbf{1}_{d+1}^\top \end{bmatrix}$ and $Q = [\mathbf{1}_{d+1,d} \quad \mathbf{0}_{d+1}]$.

- If

$$\epsilon = \frac{1+(d-1)(\lambda/2)}{d}, \qquad (A27)$$

then $P = \begin{bmatrix} \mathbf{0}_{d,d+1} \\ \mathbf{1}_{d+1}^\top \end{bmatrix}$ and $Q = \begin{bmatrix} I_d & \mathbf{0}_d \\ \mathbf{0}_d^\top & 0 \end{bmatrix}$.

- If

$$\epsilon \geq \frac{\lambda}{2} + \frac{3}{2}\frac{2-\lambda}{(d-1)\lambda^2+3}, \qquad (A28)$$

then $P = \mathbf{0}_{d+1,d+1}$ and $Q = \mathbf{0}_{d+1,d+1}$.

*Proof.*

**Overview.** The loss function $\mathcal{L}(P, Q)$ is determined only by the last row of $P$ and the first $d$ columns of $Q$. Let

$$P := \begin{bmatrix} \mathbf{0}_{d,d+1} \\ b^\top \end{bmatrix}, \qquad Q := [A \quad \mathbf{0}_{d+1}], \qquad (A29)$$

where $b \in \mathbb{R}^{d+1}$ and $A := [a_1 \cdots a_d] \in \mathbb{R}^{(d+1) \times d}$. With $b$, $A$, and $G := Z_\Delta M Z_\Delta^\top / N$, the loss function $\mathcal{L}(P, Q)$ can be represented as:

$$\mathcal{L}(P, Q) := \mathbb{E}_{c,\{(x_n,y_n)\}_{n=1}^{N+1}} \left[ \max_{\|\Delta\|_\infty \leq \epsilon} -y_{N+1}[f(Z_\Delta; P, Q)]_{d+1,N+1} \right] \qquad (A30)$$

$$= \mathbb{E}_{c,\{(\boldsymbol{x}_n,y_n)\}_{n=1}^{N+1}} \left[ \max_{\|\boldsymbol{\Delta}\|_\infty \leq \epsilon} -y_{N+1} \left[ \boldsymbol{Z}_{\boldsymbol{\Delta}} + \frac{1}{N} \boldsymbol{P} \boldsymbol{Z}_{\boldsymbol{\Delta}} \boldsymbol{M} \boldsymbol{Z}_{\boldsymbol{\Delta}}^\top \boldsymbol{Q} \boldsymbol{Z}_{\boldsymbol{\Delta}} \right]_{d+1,N+1} \right] \quad (A31)$$

$$= \mathbb{E}_{c,\{(\boldsymbol{x}_n,y_n)\}_{n=1}^{N+1}} \left[ \max_{\|\boldsymbol{\Delta}\|_\infty \leq \epsilon} -y_{N+1} \boldsymbol{b}^\top \boldsymbol{G} \boldsymbol{A} (\boldsymbol{x}_{N+1} + \boldsymbol{\Delta}) \right]. \quad (A32)$$

Using $\boldsymbol{b}$ and $\boldsymbol{A}$, we redefine the loss function as $\mathcal{L}(\boldsymbol{b}, \boldsymbol{A}) := \mathcal{L}(\boldsymbol{P}, \boldsymbol{Q})$. Since $\boldsymbol{G}$ does not include $\boldsymbol{\Delta}$ and $\max_{\|\boldsymbol{\Delta}\|_\infty \leq \epsilon} \boldsymbol{w}^\top \boldsymbol{\Delta} = \epsilon \|\boldsymbol{w}\|_1$ for $\boldsymbol{w} \in \mathbb{R}^d$, the inner maximization can be solved as:

$$\mathcal{L}(\boldsymbol{b}, \boldsymbol{A}) = \mathbb{E}_{c,\{(\boldsymbol{x}_n,y_n)\}_{n=1}^{N+1}} \left[ -y_{N+1} \boldsymbol{b}^\top \boldsymbol{G} \boldsymbol{A} \boldsymbol{x}_{N+1} + \epsilon \|\boldsymbol{b}^\top \boldsymbol{G} \boldsymbol{A}\|_1 \right]. \quad (A33)$$

When $0 \leq \boldsymbol{b} \leq 1$ and $0 \leq \boldsymbol{A} \leq 1$, then $\|\boldsymbol{b}^\top \boldsymbol{G} \boldsymbol{A}\|_1 = \boldsymbol{b}^\top \boldsymbol{G} \boldsymbol{A} \boldsymbol{1}$ since all the elements of $\boldsymbol{G}$ are nonnegative. Thus,

$$\min_{0 \leq \boldsymbol{b} \leq 1, 0 \leq \boldsymbol{A} \leq 1} \mathcal{L}(\boldsymbol{b}, \boldsymbol{A})$$
$$= \min_{0 \leq \boldsymbol{b} \leq 1, 0 \leq \boldsymbol{A} \leq 1} \mathbb{E}_{c,\{(\boldsymbol{x}_n,y_n)\}_{n=1}^{N+1}} \left[ -y_{N+1} \boldsymbol{b}^\top \boldsymbol{G} \boldsymbol{A} \boldsymbol{x}_{N+1} + \epsilon \boldsymbol{b}^\top \boldsymbol{G} \boldsymbol{A} \boldsymbol{1} \right]. \quad (A34)$$

Let the $i$-th row of $\boldsymbol{G}$ be $\boldsymbol{g}_i^\top$. Rearranging the argument of the expectation as:

$$-y_{N+1} \boldsymbol{b}^\top \boldsymbol{G} \boldsymbol{A} \boldsymbol{x}_{N+1} + \epsilon \boldsymbol{b}^\top \boldsymbol{G} \boldsymbol{A} \boldsymbol{1} = -\sum_{j=1}^{d+1} \sum_{k=1}^{d} A_{j,k} \left( \sum_{i=1}^{d+1} b_i g_{i,j} (y_{N+1} x_{N+1,k} - \epsilon) \right). \quad (A35)$$

Thus, the objective function can be represented as:

$$\max_{0 \leq \boldsymbol{b} \leq 1, 0 \leq \boldsymbol{A} \leq 1} \sum_{j=1}^{d+1} \sum_{k=1}^{d} A_{j,k} \left( \sum_{i=1}^{d+1} b_i \mathbb{E}_{c,\{(\boldsymbol{x}_n,y_n)\}_{n=1}^{N+1}} [g_{i,j} (y_{N+1} x_{N+1,k} - \epsilon)] \right). \quad (A36)$$

Since the objective function is linear with respect to $\boldsymbol{b}$ and $\boldsymbol{A}$, respectively, the optimal solution exists on the boundary:

$$\max_{\boldsymbol{b} \in \{0,1\}^{d+1}, \boldsymbol{A} \in \{0,1\}^{(d+1) \times d}} \sum_{j=1}^{d+1} \sum_{k=1}^{d} A_{j,k} \left( \sum_{i=1}^{d+1} b_i \mathbb{E}_{c,\{(\boldsymbol{x}_n,y_n)\}_{n=1}^{N+1}} [g_{i,j} (y_{N+1} x_{N+1,k} - \epsilon)] \right). \quad (A37)$$

This is maximized by $A_{j,k} = 1$ if $\sum_{i=1}^{d+1} b_i \mathbb{E}_{c,\{(\boldsymbol{x}_n,y_n)\}_{n=1}^{N+1}} [g_{i,j} (y_{N+1} x_{N+1,k} - \epsilon)]) \geq 0$ and $0$ otherwise. Now,

$$\max_{\boldsymbol{b} \in \{0,1\}^{d+1}} \sum_{j=1}^{d+1} \sum_{k=1}^{d} \phi \left( \sum_{i=1}^{d+1} b_i \mathbb{E}_{c,\{(\boldsymbol{x}_n,y_n)\}_{n=1}^{N+1}} [g_{i,j} (y_{N+1} x_{N+1,k} - \epsilon)] \right), \quad (A38)$$

where $\phi(x) := \max(0, x)$. Calculating the expectation and optimizing $\boldsymbol{b}$, we obtain the solution.

**Calculation of the expectation.** First, we consider the expectation given $c$. Since $y_n x_{n,i} = 1$ if $i = c$ and $y_n x_{n,i} \sim U(0, \lambda)$ otherwise, the expectation of $y_n \boldsymbol{x}_n$ can be calculated as:

$$\mathbb{E}[y_n x_{n,i} \mid c] = \begin{cases} 1 & (i = c) \\ \frac{\lambda}{2} & (i \neq c) \end{cases}, \qquad \mathbb{E}[y_n \boldsymbol{x}_n^\top \mid c] = \begin{bmatrix} \frac{\lambda}{2} & \cdots & \frac{\lambda}{2} & \underbrace{1}_{c\text{-th}} & \frac{\lambda}{2} & \cdots & \frac{\lambda}{2} \end{bmatrix}. \quad (A39)$$

The expectation of $\boldsymbol{G}$ can be calculated as:

$$\mathbb{E}_{\{(\boldsymbol{x}_n,y_n)\}_{n=1}^{N}} [\boldsymbol{G} \mid c] = \frac{1}{N} \mathbb{E}_{\{(\boldsymbol{x}_n,y_n)\}_{n=1}^{N}} [\boldsymbol{Z}_{\boldsymbol{\Delta}} \boldsymbol{M} \boldsymbol{Z}_{\boldsymbol{\Delta}}^\top \mid c] \quad (A40)$$

$$= \frac{1}{N} \begin{bmatrix} \sum_{n=1}^{N} \mathbb{E}_{\boldsymbol{x}_n}[\boldsymbol{x}_n \boldsymbol{x}_n^\top \mid c] & \sum_{n=1}^{N} \mathbb{E}_{\boldsymbol{x}_n,y_n}[y_n \boldsymbol{x}_n \mid c] \\ \sum_{n=1}^{N} \mathbb{E}_{\boldsymbol{x}_n,y_n}[y_n \boldsymbol{x}_n^\top \mid c] & N \end{bmatrix} \quad (A41)$$

$$= \begin{bmatrix} \mathbb{E}_{\boldsymbol{x}_n}[\boldsymbol{x}_n \boldsymbol{x}_n^\top \mid c] & \mathbb{E}_{\boldsymbol{x}_n,y_n}[y_n \boldsymbol{x}_n \mid c] \\ \mathbb{E}_{\boldsymbol{x}_n,y_n}[y_n \boldsymbol{x}_n^\top \mid c] & 1 \end{bmatrix}. \quad (A42)$$

For $y_n = 1$ and $i, j \neq c$, $\mathbb{E}[x_{n,i}^2 \mid c] = \int_0^\lambda x^2/\lambda \, \mathrm{d}x = \lambda^2/3$ and $\mathbb{E}[x_{n,i}x_{n,j} \mid c] = \mathbb{E}[x_{n,i} \mid c]\mathbb{E}[x_{n,j} \mid c] = \lambda^2/4$. Thus,

$$
\mathbb{E}_{\{(\boldsymbol{x}_n, y_n)\}_{n=1}^N}[g_{i,j} \mid c] = \begin{cases}
1 & (i = c) \wedge (j = i, d+1) \\
\frac{\lambda}{2} & (i = c) \wedge (j \neq i, d+1) \\
\frac{\lambda^2}{3} & (i \in [d], i \neq c) \wedge (j = i) \\
\frac{\lambda}{2} & (i \in [d], i \neq c) \wedge (j = c, d+1) \\
\frac{\lambda^2}{4} & (i \in [d], i \neq c) \wedge (j \neq i, c, d+1) \\
1 & (i = d+1) \wedge (j = c, d+1) \\
\frac{\lambda}{2} & (i = d+1) \wedge (j \neq c, d+1)
\end{cases} . \tag{A43}
$$

Note that

$$
\mathbb{E}_{\{(\boldsymbol{x}_n, y_n)\}_{n=1}^N}[\boldsymbol{G} \mid c]
$$
$$
= \begin{bmatrix}
\lambda^2/3 & \lambda^2/4 & \lambda^2/4 & \cdots & \lambda^2/4 & \overbrace{\lambda/2}^{c\text{-th}} & \lambda^2/4 & \cdots & \lambda^2/4 & \lambda/2 \\
\lambda^2/4 & \lambda^2/3 & \lambda^2/4 & \cdots & \lambda^2/4 & \lambda/2 & \lambda^2/4 & \cdots & \lambda^2/4 & \lambda/2 \\
\vdots & & & & & & & & & \\
\lambda^2/4 & \lambda^2/4 & \lambda^2/4 & \cdots & \lambda^2/3 & \lambda/2 & \lambda^2/4 & \cdots & \lambda^2/4 & \lambda/2 \\
\lambda/2 & \lambda/2 & \lambda/2 & \cdots & \lambda/2 & 1 & \lambda/2 & \cdots & \lambda/2 & 1 \\
\lambda^2/4 & \lambda^2/4 & \lambda^2/4 & \cdots & \lambda^2/4 & \lambda/2 & \lambda^2/3 & \cdots & \lambda^2/4 & \lambda/2 \\
\vdots & & & & & & & & & \\
\lambda^2/4 & \lambda^2/4 & \lambda^2/4 & \cdots & \lambda^2/4 & \lambda/2 & \lambda^2/4 & \cdots & \lambda^2/3 & \lambda/2 \\
\lambda/2 & \lambda/2 & \lambda/2 & \cdots & \lambda/2 & 1 & \lambda/2 & \cdots & \lambda/2 & 1
\end{bmatrix} \Big\}c\text{-th}. \tag{A44}
$$

Let

$$
h_i(j; k; c) := \mathbb{E}_{\{(\boldsymbol{x}_n, y_n)\}_{n=1}^{N+1}}[g_{i,j}(y_{N+1}x_{N+1,k} - \epsilon) \mid c]. \tag{A45}
$$

Let $\epsilon_+ := 1 - \epsilon$ and $\epsilon_- := \lambda/2 - \epsilon$. By Eqs. (A39) and (A43),

$$
h_i(j; k; c) = \begin{cases}
\epsilon_+ & (i \in [d]) \wedge (j = i, d+1) \wedge (k = i) \wedge (c = i) \\
\epsilon_- & (i \in [d]) \wedge (j = i, d+1) \wedge (k \neq i) \wedge (c = i) \\
\frac{\lambda}{2}\epsilon_+ & (i \in [d]) \wedge (j \neq i, d+1) \wedge (k = i) \wedge (c = i) \\
\frac{\lambda}{2}\epsilon_- & (i \in [d]) \wedge (j \neq i, d+1) \wedge (k \neq i) \wedge (c = i) \\
\frac{\lambda^2}{3}\epsilon_- & (i \in [d]) \wedge (j = i) \wedge (k = i) \wedge (c \neq i) \\
\frac{\lambda}{2}\epsilon_- & (i \in [d]) \wedge (j = c, d+1) \wedge (k = i) \wedge (c \neq i) \\
\frac{\lambda^2}{4}\epsilon_- & (i \in [d]) \wedge (j \neq i, c, d+1) \wedge (k = i) \wedge (c \neq i) \\
\frac{\lambda^2}{3}\epsilon_+ & (i \in [d]) \wedge (j = i) \wedge (k = c) \wedge (c \neq i) \\
\frac{\lambda}{2}\epsilon_+ & (i \in [d]) \wedge (j = c, d+1) \wedge (k = c) \wedge (c \neq i) \\
\frac{\lambda^2}{4}\epsilon_+ & (i \in [d]) \wedge (j \neq i, c, d+1) \wedge (k = c) \wedge (c \neq i) \\
\frac{\lambda^2}{3}\epsilon_- & (i \in [d]) \wedge (j = i) \wedge (k \neq i, c) \wedge (c \neq i) \\
\frac{\lambda}{2}\epsilon_- & (i \in [d]) \wedge (j = c, d+1) \wedge (k \neq i, c) \wedge (c \neq i) \\
\frac{\lambda^2}{4}\epsilon_- & (i \in [d]) \wedge (j \neq i, c, d+1) \wedge (k \neq i, c) \wedge (c \neq i) \\
\epsilon_+ & (i = d+1) \wedge (j = c, d+1) \wedge (k = c) \\
\epsilon_- & (i = d+1) \wedge (j = c, d+1) \wedge (k \neq c) \\
\frac{\lambda}{2}\epsilon_+ & (i = d+1) \wedge (j \neq c, d+1) \wedge (k = c) \\
\frac{\lambda}{2}\epsilon_- & (i = d+1) \wedge (j \neq c, d+1) \wedge (k \neq c)
\end{cases} . \tag{A46}
$$

Then, we compute the expectation along $c$. Note that

$$
\mathbb{E}_{c, \{(\boldsymbol{x}_n, y_n)\}_{n=1}^{N+1}}[g_{i,j}(y_{N+1}x_{N+1,k} - \epsilon)] = \frac{1}{d}\sum_{c=1}^d h_i(j; k; c). \tag{A47}
$$

Let $H_{i,j,k} := \sum_{c=1}^d h_i(j; k; c)$. The summation of $h_i$ along $c$ can be calculated as:

For $(i \in [d]) \wedge (j = i) \wedge (k = i)$,

$$H_{i,j,k} = h_i(j = i; k = i; c = i) + \sum_{c \neq i}^{d} h_i(j = i; k = i; c \neq i) = \epsilon_+ + \frac{\lambda^2}{3}(d-1)\epsilon_- \tag{A48}$$

$$=: r_1. \tag{A49}$$

For $(i \in [d]) \wedge (j = i) \wedge (k \neq i)$,

$$H_{i,j,k} = h_i(j = i; k \neq i; c = i) + h_i(j = i; k = c; c \neq i) + \sum_{c \neq i,k}^{d} h(j = i; k \neq i, c; c \neq i) \tag{A50}$$

$$= \epsilon_- + \frac{\lambda^2}{3}\epsilon_+ + \frac{\lambda^2}{3}(d-2)\epsilon_- \tag{A51}$$

$$=: r_2. \tag{A52}$$

For $(i \in [d]) \wedge (j = d+1) \wedge (k = i)$,

$$H_{i,j,k} = h_i(j = d+1; k = i; c = i) + \sum_{c \neq i}^{d} h_i(j = d+1; k = i; c \neq i) \tag{A53}$$

$$= \epsilon_+ + \frac{\lambda}{2}(d-1)\epsilon_- \tag{A54}$$

$$=: r_3. \tag{A55}$$

For $(i \in [d]) \wedge (j = d+1) \wedge (k \neq i)$,

$$H_{i,j,k} = h_i(j = d+1; k \neq i; c = i) + h_i(j = d+1; k = c; c \neq i)$$

$$+ \sum_{c \neq i,k}^{d} h_i(j = d+1; k \neq i, c; c \neq i) \tag{A56}$$

$$= \epsilon_- + \frac{\lambda}{2}\epsilon_+ + \frac{\lambda}{2}(d-2)\epsilon_- \tag{A57}$$

$$=: r_4. \tag{A58}$$

For $(i \in [d]) \wedge (j \neq i, d+1) \wedge (k = i)$,

$$H_{i,j,k} = h_i(j \neq i, d+1; k = i; c = i) + h_i(j = c; k = i; c \neq i)$$

$$+ \sum_{c \neq i,j}^{d} h_i(j \neq i, c, d+1; k = i; c \neq i) \tag{A59}$$

$$= \frac{\lambda}{2}\epsilon_+ + \frac{\lambda}{2}\epsilon_- + \frac{\lambda^2}{4}(d-2)\epsilon_- \tag{A60}$$

$$=: r_5. \tag{A61}$$

For $(i \in [d]) \wedge (j \neq i, d+1) \wedge (k \neq i) \wedge (j = k)$,

$$H_{i,j,k} = h_i(j \neq i, d+1; k \neq i; c = i) + h_i(j = c; k = c; c \neq i)$$

$$+ \sum_{c \neq i,j,k}^{d} h_i(j \neq i, c, d+1; k \neq i, c; c \neq i) \tag{A62}$$

$$= \frac{\lambda}{2}\epsilon_- + \frac{\lambda}{2}\epsilon_+ + \frac{\lambda^2}{4}(d-2)\epsilon_- \tag{A63}$$

$$=: r_5. \tag{A64}$$

For $(i \in [d]) \wedge (j \neq i, d+1) \wedge (k \neq i) \wedge (j \neq k)$,

$$H_{i,j,k} = h_i(j \neq i, d+1; k \neq i; c = i) + h_i(j = c; k \neq i, c; c \neq i)$$

$$+ h_i(j \neq i, c, d+1; k = c; c \neq i)$$

$$+ \sum_{c \neq i,j,k}^{d} h_i(j \neq i, c, d+1; k \neq i, c; c \neq i) \tag{A65}$$

$$= \frac{\lambda}{2}\epsilon_- + \frac{\lambda}{2}\epsilon_- + \frac{\lambda^2}{4}\epsilon_+ + \frac{\lambda^2}{4}(d-3)\epsilon_- \tag{A66}$$

$$=: r_6. \tag{A67}$$

For $(i = d+1) \wedge (j = d+1)$,

$$H_{i,j,k} = h_i(j = d+1; k = c; c = k) + \sum_{c \neq k}^{d} h_i(j = d+1; k \neq c; c \neq k) \tag{A68}$$

$$= \epsilon_+ + (d-1)\epsilon_- \tag{A69}$$

$$=: r_7. \tag{A70}$$

For $(i = d+1) \wedge (j \neq d+1) \wedge (j = k)$,

$$H_{i,j,k} = h_i(j = c; k = c; c = k) + \sum_{c \neq k}^{d} h_i(j \neq d+1; k \neq c; c \neq k) \tag{A71}$$

$$= \epsilon_+ + \frac{\lambda}{2}(d-1)\epsilon_- \tag{A72}$$

$$=: r_3. \tag{A73}$$

For $(i = d+1) \wedge (j \neq d+1) \wedge (j \neq k)$,

$$H_{i,j,k} = h_i(j = c; k \neq c; c \neq k) + h_i(j \neq c; k = c; c = k)$$

$$+ \sum_{c \neq j,k}^{d} h_i(j \neq c, d+1; k \neq c; c \neq k) \tag{A74}$$

$$= \epsilon_- + \frac{\lambda}{2}\epsilon_+ + \frac{\lambda}{2}(d-2)\epsilon_- \tag{A75}$$

$$=: r_4. \tag{A76}$$

**Optimization of $A$ and $b$.** From Eq. (A38), we redefine the objective function as:

$$d \max_{\boldsymbol{b} \in \{0,1\}^{d+1}} \sum_{j=1}^{d+1} \sum_{k=1}^{d} \phi\left(\sum_{i=1}^{d+1} b_i \mathbb{E}_{c,\{(\boldsymbol{x}_n, y_n)\}_{n=1}^{N+1}}[g_{i,j}(y_{N+1} x_{N+1,k} - \epsilon)]\right)$$

$$= \max_{\boldsymbol{b} \in \{0,1\}^{d+1}} \sum_{j=1}^{d+1} \sum_{k=1}^{d} \phi\left(\sum_{i=1}^{d+1} b_i H_{i,j,k}\right). \tag{A77}$$

Recall that we set $A_{j,k} = 1$ if $\sum_{i=1}^{d+1} b_i H_{i,j,k} \geq 0$ and 0 otherwise. Let $[d]' := \{i \in [d] \mid b_i = 1\}$ and $d' := |[d]'|$. Now,

$$\sum_{j=1}^{d+1} \sum_{k=1}^{d} \phi\left(\sum_{i=1}^{d+1} b_i H_{i,j,k}\right) = \sum_{k=1}^{d} \phi\left(b_{d+1} H_{d+1,d+1,k} + \mathbb{1}[k \in [d]'] H_{k,d+1,k} + \sum_{i \in [d]', i \neq k} H_{i,d+1,k}\right)$$

$$+ \sum_{j=1}^{d} \phi\left(b_{d+1} H_{d+1,j,j} + \mathbb{1}[j \in [d]'] H_{j,j,j} + \sum_{i \in [d]', i \neq j} H_{i,j,j}\right)$$

$$+ \sum_{j=1}^{d} \sum_{k \neq j}^{d} \phi\left(b_{d+1} H_{d+1,j,k} + \mathbb{1}[j \in [d]'] H_{i,i,k}\right.$$

$$\left. + \mathbb{1}[k \in [d]'] H_{i,j,i} + \sum_{i \in [d]', i \neq j,k} H_{i,j,k}\right). \tag{A78}$$

By Eqs. (A55), (A58) and (A70),

$$\sum_{k=1}^{d} \phi\left( b_{d+1}H_{d+1,d+1,k} + \mathbb{1}[k \in [d]']H_{k,d+1,k} + \sum_{i \in [d]', i \neq k} H_{i,d+1,k} \right)$$

$$= \sum_{k=1}^{d} \phi\left( b_{d+1}r_7 + \mathbb{1}[k \in [d]']r_3 + \sum_{i \in [d]', i \neq k} r_4 \right) \tag{A79}$$

$$= d'\phi(\underbrace{b_{d+1}r_7 + r_3 + (d'-1)r_4}_{=:s_1(d',b_{d+1})}) + (d-d')\phi(\underbrace{b_{d+1}r_7 + d'r_4}_{=:s_2(d',b_{d+1})}). \tag{A80}$$

By Eqs. (A49), (A64) and (A73),

$$\sum_{j=1}^{d} \phi\left( b_{d+1}H_{d+1,j,j} + \mathbb{1}[j \in [d]']H_{j,j,j} + \sum_{i \in [d]', i \neq j} H_{i,j,j} \right)$$

$$= \sum_{j=1}^{d} \phi\left( b_{d+1}r_3 + \mathbb{1}[j \in [d]']r_1 + \sum_{i \in [d]', i \neq j} r_5 \right) \tag{A81}$$

$$= d'\phi(\underbrace{b_{d+1}r_3 + r_1 + (d'-1)r_5}_{=:s_3(d',b_{d+1})}) + (d-d')\phi(\underbrace{b_{d+1}r_3 + d'r_5}_{=:s_4(d',b_{d+1})}). \tag{A82}$$

By Eqs. (A52), (A61), (A67) and (A76),

$$\sum_{j=1}^{d}\sum_{k \neq j}^{d} \phi\left( b_{d+1}H_{d+1,j,k} + \mathbb{1}[j \in [d]']H_{i,i,k} + \mathbb{1}[k \in [d]']H_{i,j,i} + \sum_{i \in [d]', i \neq j,k} H_{i,j,k} \right)$$

$$= \sum_{j=1}^{d}\sum_{k \neq j}^{d} \phi\left( b_{d+1}r_4 + \mathbb{1}[j \in [d]']r_2 + \mathbb{1}[k \in [d]']r_5 + \sum_{i \in [d]', i \neq j,k} r_6 \right) \tag{A83}$$

$$= d'(d'-1)\phi(\underbrace{b_{d+1}r_4 + r_2 + r_5 + (d'-2)r_6}_{=:s_5(d',b_{d+1})}) + d'(d-d')\phi(\underbrace{b_{d+1}r_4 + r_2 + (d'-1)r_6}_{=:s_6(d',b_{d+1})})$$

$$+ d'(d-d')\phi(\underbrace{b_{d+1}r_4 + r_5 + (d'-1)r_6}_{=:s_7(d',b_{d+1})}) + (d-d')(d-d'-1)\phi(\underbrace{b_{d+1}r_4 + d'r_6}_{=:s_8(d',b_{d+1})}). \tag{A84}$$

Now,

$$\sum_{j=1}^{d+1}\sum_{k=1}^{d} \phi\left( \sum_{i=1}^{d+1} b_i H_{i,j,k} \right) = d'\phi(s_1(d',b_{d+1})) + (d-d')\phi(s_2(d',b_{d+1})) + d'\phi(s_3(d',b_{d+1}))$$

$$+ (d-d')\phi(s_4(d',b_{d+1})) + d'(d'-1)\phi(s_5(d',b_{d+1}))$$

$$+ d'(d-d')\phi(s_6(d',b_{d+1})) + d'(d-d')\phi(s_7(d',b_{d+1}))$$

$$+ (d-d')(d-d'-1)\phi(s_8(d',b_{d+1})) \tag{A85}$$

$$=: \mathrm{score}(d',b_{d+1}). \tag{A86}$$

We shall now summarize the discussion to Lemma E.2. The rest of the proof is left to Lemma E.3.

$\square$

**Optimization of transformed problem.**

**Lemma E.2.** Let $\phi(x) := \max(0,x)$, $d \in \mathbb{N}$, $0 < \lambda < 1$, $0 \leq \epsilon < 1$, $\epsilon_+ := 1-\epsilon$, and $\epsilon_- := \lambda/2 - \epsilon$. In addition, for $d' \in \{0, \ldots, d\}$ and $b_{d+1} \in \{0,1\}$,

$$r_1 := \epsilon_+ + \frac{\lambda^2}{3}(d-1)\epsilon_-, \tag{A87}$$

$$r_2 := \epsilon_- + \frac{\lambda^2}{3}\epsilon_+ + \frac{\lambda^2}{3}(d-2)\epsilon_-, \tag{A88}$$

$$r_3 := \epsilon_+ + \frac{\lambda}{2}(d-1)\epsilon_-, \tag{A89}$$

$$r_4 := \epsilon_- + \frac{\lambda}{2}\epsilon_+ + \frac{\lambda}{2}(d-2)\epsilon_-, \tag{A90}$$

$$r_5 := \frac{\lambda}{2}\epsilon_+ + \frac{\lambda}{2}\epsilon_- + \frac{\lambda^2}{4}(d-2)\epsilon_-, \tag{A91}$$

$$r_6 := \frac{\lambda}{2}\epsilon_- + \frac{\lambda}{2}\epsilon_- + \frac{\lambda^2}{4}\epsilon_+ + \frac{\lambda^2}{4}(d-3)\epsilon_-, \tag{A92}$$

$$r_7 := \epsilon_+ + (d-1)\epsilon_-, \tag{A93}$$

$$s_1(d', b_{d+1}) := b_{d+1}r_7 + r_3 + (d'-1)r_4, \tag{A94}$$

$$s_2(d', b_{d+1}) := b_{d+1}r_7 + d'r_4, \tag{A95}$$

$$s_3(d', b_{d+1}) := b_{d+1}r_3 + r_1 + (d'-1)r_5, \tag{A96}$$

$$s_4(d', b_{d+1}) := b_{d+1}r_3 + d'r_5, \tag{A97}$$

$$s_5(d', b_{d+1}) := b_{d+1}r_4 + r_2 + r_5 + (d'-2)r_6, \tag{A98}$$

$$s_6(d', b_{d+1}) := b_{d+1}r_4 + r_2 + (d'-1)r_6, \tag{A99}$$

$$s_7(d', b_{d+1}) := b_{d+1}r_4 + r_5 + (d'-1)r_6, \tag{A100}$$

$$s_8(d', b_{d+1}) := b_{d+1}r_4 + d'r_6, \tag{A101}$$

$$\begin{aligned}
\text{score}(d', b_{d+1}) := \ & d'\phi(s_1(d', b_{d+1})) + (d-d')\phi(s_2(d', b_{d+1})) + d'\phi(s_3(d', b_{d+1})) \\
& + (d-d')\phi(s_4(d', b_{d+1})) + d'(d'-1)\phi(s_5(d', b_{d+1})) \\
& + d'(d-d')\phi(s_6(d', b_{d+1})) + d'(d-d')\phi(s_7(d', b_{d+1})) \\
& + (d-d')(d-d'-1)\phi(s_8(d', b_{d+1})).
\end{aligned} \tag{A102}$$

Considering the following optimization problem:

$$\max_{d'\in\{0,\dots,d\},b_{d+1}\in\{0,1\}} \text{score}(d', b_{d+1}). \tag{A103}$$

Then, setting $\boldsymbol{P}, \boldsymbol{Q} \in \mathbb{R}^{(d+1)\times(d+1)}$ to

$$\boldsymbol{P} = \begin{bmatrix} \boldsymbol{0}_{d,d+1} \\ \boldsymbol{b}^\top \end{bmatrix}, \ \ \boldsymbol{Q} = [\boldsymbol{A} \ \ \boldsymbol{0}_{d+1}], \ \ \boldsymbol{b}^\top = [\underbrace{1 \ \ 1 \ \ \cdots \ \ 1}_{d'} \ \ \underbrace{0 \ \ 0 \ \ \cdots \ \ 0}_{d-d'} \ \ b_{d+1}], \tag{A104}$$

$$\begin{aligned}
&A_{jk} \\
&= \begin{cases}
\mathbb{1}[b_{d+1}r_7 + \mathbb{1}[k \le d']r_3 + (d' - \mathbb{1}[k \le d'])r_4 \ge 0] \\
\quad (j = d+1) \\
\mathbb{1}[b_{d+1}r_3 + \mathbb{1}[j \le d']r_1 + (d' - \mathbb{1}[j \le d'])r_5 \ge 0] \\
\quad (j \ne d+1) \wedge (j = k) \\
\mathbb{1}[b_{d+1}r_4 + \mathbb{1}[j \le d']r_2 + \mathbb{1}[k \le d']r_5 + (d' - \mathbb{1}[j \le d'] - \mathbb{1}[k \le d'])r_6 \ge 0] \\
\quad (j \ne d+1) \wedge (j \ne k)
\end{cases},
\end{aligned} \tag{A105}$$

the global maximizer of (A103) is the global minimizer of (7).

*Proof.* See the above discussion. $\square$

**Lemma E.3.** The global maximizer of (A103) is as follows:

(a) If

$$0 \leq \epsilon \leq \frac{\lambda(\lambda(d-2)+4)}{2(\lambda(d-1)+2)}, \tag{A106}$$

then $d' = d$ and $b_{d+1} = 1$. This corresponds to $\boldsymbol{b} = \mathbf{1}_{d+1}$ and $\boldsymbol{A} = \mathbf{1}_{d+1,d}$.

(b) If

$$\epsilon = \frac{\lambda(d-1)+2}{2d}, \tag{A107}$$

then $d' = d$ and $b_{d+1} = 1$. This corresponds to $\boldsymbol{b} = \mathbf{1}_{d+1}$ and $\boldsymbol{A} = [\boldsymbol{I}_d \ \boldsymbol{0}_d]^\top$.

(c) If

$$\epsilon \geq \frac{\lambda}{2} + \frac{3}{2} \frac{2-\lambda}{\lambda^2(d-1)+3}, \tag{A108}$$

then $d' = 0$ and $b_{d+1} = 0$. This corresponds to $\boldsymbol{b} = \mathbf{1}_{d+1}$ and $\boldsymbol{A} = \boldsymbol{0}_{d+1,d}$.

*Proof.* For notational simplicity, we abbreviate terms including variables such as $x_1, x_2, \ldots$ (e.g., $x_1^2 + 3x_2 + \cdots$) using the notation $\Theta(x_1, x_2, \ldots)$. In particular, when the expression is strictly nonnegative (e.g., $x_1^2 + x_2^2$) or nonpositive, we use $\Theta_+(x_1, x_2, \ldots)$ or $\Theta_-(x_1, x_2, \ldots)$, respectively. These terms are not essential to the analysis and too long. They can be derived by simple basic arithmetic operations. These concrete values can be showed by our python codes.

We define $\epsilon_1, \ldots, \epsilon_7$ as

$$r_1 = 0 \iff \epsilon = \frac{\lambda}{2} + \frac{3}{2} \frac{2-\lambda}{\lambda^2(d-1)+3} =: \epsilon_1, \tag{A109}$$

$$r_2 = 0 \iff \epsilon = \frac{\lambda(\lambda^2(d-2)+2\lambda+3)}{2(\lambda^2(d-1)+3)} =: \epsilon_2, \tag{A110}$$

$$r_3 = 0 \iff \epsilon = \frac{\lambda^2(d-1)+4}{2(\lambda(d-1)+2)} =: \epsilon_3, \tag{A111}$$

$$r_4 = 0 \iff \epsilon = \frac{\lambda(\lambda(d-2)+4)}{2(\lambda(d-1)+2)} =: \epsilon_4, \tag{A112}$$

$$r_5 = 0 \iff \epsilon = \frac{\lambda^2(d-2)+2\lambda+4}{2(\lambda(d-2)+4)} =: \epsilon_5, \tag{A113}$$

$$r_6 = 0 \iff \epsilon = \frac{\lambda(\lambda(d-3)+6)}{2(\lambda(d-2)+4)} =: \epsilon_6, \tag{A114}$$

$$r_7 = 0 \iff \epsilon = \frac{\lambda(d-1)+2}{2d} =: \epsilon_7, \tag{A115}$$

$$s_5(d,1) = 0 \iff \epsilon = \frac{\lambda}{2} \frac{3d^2\lambda^2 - 8d\lambda^2 + 24d\lambda + 4\lambda^2 - 34\lambda + 48}{3d^2\lambda^2 - 5d\lambda^2 + 18d\lambda + 2\lambda^2 - 18\lambda + 24} =: \epsilon_{s_5}. \tag{A116}$$

Since

$$\epsilon_1 - \epsilon_3 = \frac{\lambda(d-1)(2-\lambda)(3-2\lambda)}{2(\lambda(d-1)+2)(\lambda^2(d-1)+3)} \geq 0, \tag{A117}$$

$$\epsilon_3 - \epsilon_5 = \frac{(2-\lambda)^2}{(\lambda(d-2)+4)(\lambda(d-1)+2)} \geq 0, \tag{A118}$$

$$\epsilon_5 - \epsilon_7 = \frac{(d-2)(2-\lambda)^2}{2d(\lambda(d-2)+4)} \geq 0, \tag{A119}$$

$$\epsilon_7 - \epsilon_{s_5} = \frac{(2-\lambda)(-3d\lambda^2 + 6d\lambda + 2\lambda^2 - 18\lambda + 24)}{2d(3d^2\lambda^2 - 5d\lambda^2 + 18d\lambda + 2\lambda^2 - 18\lambda + 24)} \geq 0, \tag{A120}$$

$$\epsilon_{s_5} - \epsilon_4 = \frac{\lambda^2(2-\lambda)}{(\lambda(d-1)+2)(3d^2\lambda^2 - 5d\lambda^2 + 18d\lambda + 2\lambda^2 - 18\lambda + 24)} \geq 0, \tag{A121}$$

$$\epsilon_4 - \epsilon_6 = \frac{\lambda(2-\lambda)^2}{2(\lambda(d-2)+4)(\lambda(d-1)+2)} \geq 0, \tag{A122}$$

$$\epsilon_6 - \epsilon_2 = \frac{\lambda(3-\lambda)(2-\lambda)(1-\lambda)}{2(\lambda(d-2)+4)(\lambda^2(d-1)+3)} \geq 0, \tag{A123}$$

for $d \geq 2$, they are ordered as

$$\epsilon_2 \leq \epsilon_6 \leq \epsilon_4 \leq \epsilon_{s_5} \leq \epsilon_7 \leq \epsilon_5 \leq \epsilon_3 \leq \epsilon_1. \tag{A124}$$

In score, $b_{d+1}$ appears as $b_{d+1}r_3$, $b_{d+1}r_4$, or $b_{d+1}r_7$, each with a positive coefficient in $d$ and $d'$. Thus, if $r_3, r_4, r_7 \leq 0$, then $b_{d+1}$ should be zero. If $r_3, r_4, r_7 \geq 0$, then $b_{d+1}$ should be one. Considering Ineq. (A124), for $d \geq 2$, the optimal $b_{d+1}$ is one if $\epsilon \leq \epsilon_4$ and zero if $\epsilon \geq \epsilon_3$.

**One-Dimensional Case.** If $d = 1$,

$$\begin{aligned}
&\text{score}(d', b_{d+1}) \\
&= \mathbb{1}[d' = 0](\phi(b_{d+1}r_7) + \phi(b_{d+1}r_3)) + \mathbb{1}[d' = 1](\phi(b_{d+1}r_7 + r_3) + \phi(b_{d+1}r_3 + r_1)) \quad \text{(A125)} \\
&= \mathbb{1}[d' = 0](\phi(b_{d+1}\epsilon_+) + \phi(b_{d+1}\epsilon_+)) \\
&\quad + \mathbb{1}[d' = 1](\phi(b_{d+1}\epsilon_+ + \epsilon_+) + \phi(b_{d+1}\epsilon_+ + \epsilon_+)). \tag{A126}
\end{aligned}$$

As $\epsilon_+$ is always positive for $0 \leq \epsilon < 1$, $d' = d = 1$ and $b_{d+1} = 1$ are the optimal. This aligns with the following case analysis.

**Weak Adversarial (Case 1).** Assume $d \geq 2$ and $0 \leq \epsilon \leq \epsilon_6$. As $\epsilon \leq \epsilon_6 \leq \epsilon_4$, $b_{d+1} = 1$ is the optimal. By Ineq. (A124), $r_1, r_3, r_4, r_5, r_6, r_7 \geq 0$. The sign of $r_2$ depends on $\epsilon$. Thus, $s_1(d', 1), s_2(d', 1), s_3(d', 1), s_4(d', 1), s_7(d', 1), s_8(d', 1) \geq 0$ for $0 \leq d' \leq d$. In addition, for $d' \geq 2$,

$$s_5(d', 1) \geq r_4 + r_2 \tag{A127}$$

$$= \frac{\lambda^3}{6}(d-2) + \frac{\lambda^2}{12}(3d-2) + \frac{3\lambda}{2} - \frac{\epsilon}{6}(2\lambda^2(d-1) + 3\lambda(d-1) + 12) \tag{A128}$$

$$\geq \frac{\lambda^2(2-\lambda)(5-2\lambda)}{12(\lambda(d-2)+4)} \qquad (\because \epsilon \leq \epsilon_6) \tag{A129}$$

$$\geq 0. \tag{A130}$$

Thus, $d'(d'-1)s_5(d', 1)$ is nonnegative for $0 \leq d' \leq d$. Similarly, by $s_6(d', 1) \geq r_4 + r_2 \geq 0$ for $d' \geq 1$, $d'(d'-1)s_6(d', 1)$ is nonnegative for $0 \leq d' \leq d$. Thus,

$$\begin{aligned}
\text{score}(d', 1) :=& d's_1(d', 1) + (d-d')s_2(d', 1) + d's_3(d', 1) + (d-d')s_4(d', 1) \\
&+ d'(d'-1)s_5(d', 1) + d'(d-d')s_6(d', 1) + d'(d-d')s_7(d', 1) \\
&+ (d-d')(d-d'-1)s_8(d', 1) \tag{A131} \\
=& dr_7 + d'r_3 + d'(d-1)r_4 + dr_3 + d'r_1 + d'(d-1)r_5 \\
&+ dr_4 + d'r_2 + d'r_5 + d'(d-1)(d-2)r_6. \tag{A132}
\end{aligned}$$

This monotonically increases in $d'$. Therefore, $d' = d$ is the optimal. By Lemma E.2, $\boldsymbol{b} = \boldsymbol{1}_{d+1}$. In addition, from $s_1(d, 1), s_3(d, 1), s_5(d, 1) \geq 0$, $\boldsymbol{A} = \boldsymbol{1}_{d+1,d}$.

**Weak Adversarial (Case 2).** Assume $d \geq 2$ and $\epsilon_6 \leq \epsilon \leq \epsilon_4$. As $\epsilon \leq \epsilon_4$, $b_{d+1} = 1$ is the optimal. By Ineq. (A124), $r_1, r_3, r_4, r_5, r_7 \geq 0$ and $r_2, r_6 \leq 0$. Thus, $s_1(d', 1), s_2(d', 1), s_3(d', 1), s_4(d', 1) \geq 0$. In addition,

$$s_5(d', 1) \geq s_5(d, 1) \geq \frac{\lambda^2(2-\lambda)}{12(\lambda(d-1)+2)} \geq 0 \qquad (\because \epsilon \leq \epsilon_4), \tag{A133}$$

$$s_7(d', 1) \geq s_7(d, 1) \geq \frac{\lambda(2-\lambda)^3}{8(\lambda(d-1)+2)} \geq 0 \qquad (\because \epsilon \leq \epsilon_4). \tag{A134}$$

Due to the following inequality, $s_8(d', 1)$ is always larger than $s_6(d', 1)$:

$$s_8(d', 1) - s_6(d', 1) = -\frac{\lambda^3}{24}(d+1) + \frac{5\lambda^2}{12} - \frac{\lambda}{2} + \frac{\epsilon}{12}(\lambda^2(d+2) + 12(1-\lambda)) \tag{A135}$$

$$\geq \frac{\lambda(3-\lambda)(2-\lambda)(1-\lambda)}{6(\lambda(d-2)+4)} \qquad (\because \epsilon \geq \epsilon_6) \tag{A136}$$

$$\geq 0. \tag{A137}$$

If $s_6(d',1), s_8(d',1) \geq 0$,

$$\frac{\mathrm{d}\,\mathrm{score}(d',1)}{\mathrm{d}d'} = \frac{(2+\lambda(d-1)-2d\epsilon)(\lambda^2(3d^2-5d+2)+18\lambda(d-1)+24)}{24} \geq 0. \tag{A138}$$

We used

$$2 + \lambda(d-1) - 2d\epsilon \geq \frac{(2-\lambda)^2}{\lambda(d-1)+2} \geq 0 \qquad (\because \epsilon \leq \epsilon_4). \tag{A139}$$

If $s_6(d',1) \leq 0, s_8(d',1) \geq 0$,

$$\frac{\mathrm{d}\,\mathrm{score}(d',1)}{\mathrm{d}d'} = \Theta(d,d',\lambda) - \frac{\epsilon}{12}\left\{3d\lambda^2((d-d')^2 + 2d'^2) + 6\lambda(2-\lambda)\left\{\left(d - \frac{1}{2}d'\right)^2 + \frac{11}{4}d'^2\right\}\right.$$

$$\left. + 8dd'\lambda^2 + d'(4\lambda^2 - 36\lambda + 48)\right\} \tag{A140}$$

$$\geq \Theta(d,\lambda) - \frac{\lambda(2-\lambda)}{24(\lambda(d-1)+2)}d'(9d'\lambda(2-\lambda) + 6\lambda^2(d+1) - 4\lambda(3d+7) + 24)$$

$$(\because \epsilon \leq \epsilon_4) \tag{A141}$$

$$\geq \frac{(2-\lambda)(d\lambda^3 + d\lambda(12 - 7\lambda) - \lambda^3 + 11\lambda^2 - 30\lambda + 24)}{12(\lambda(d-1)+2)} \tag{A142}$$

$$\geq 0. \tag{A143}$$

We used for $0 \leq d' \leq d$,

$$d'(9d'\lambda(2-\lambda) + 6\lambda^2(d+1) - 4\lambda(3d+7) + 24)$$

$$\leq d\lambda(3d\lambda(2-\lambda) + 6\lambda^2 - 28\lambda + 24). \tag{A144}$$

If $s_6(d',1) \leq 0, s_8(d',1) \leq 0$,

$$\frac{\mathrm{d}\,\mathrm{score}(d',1)}{\mathrm{d}d'}$$

$$= \Theta(d,d',\lambda) - \frac{\epsilon}{12}\{3d^2\lambda(\lambda+4) + 6d(-\lambda^2 - \lambda + 2) + 6\lambda + 12(d-1)$$

$$+ 2d'(3d^2\lambda^2 + 8d\lambda(-\lambda+1) + 4(2\lambda^2 + (d-6)\lambda + 3))\} \tag{A145}$$

$$\geq \Theta(d,\lambda) - \frac{\lambda(2-\lambda)}{12(\lambda(d-1)+2)}d'(-3d\lambda^2 + 6d\lambda + 6\lambda^2 - 20\lambda + 12) \qquad (\because \epsilon \leq \epsilon_4) \tag{A146}$$

$$\geq \frac{(2-\lambda)(-d\lambda^3 - 8d\lambda^2 + 24d\lambda - 2\lambda^3 + 22\lambda^2 - 60\lambda + 48)}{24(\lambda(d-1)+2)} \qquad (\because d' \leq d) \tag{A147}$$

$$\geq 0. \tag{A148}$$

From the above discussion, for any case, $(s_6, s_8 \geq 0)$, $(s_6 \leq 0 \text{ and } s_8 \geq 0)$, or $(s_6, s_8 \leq 0)$, the derivative of $\mathrm{score}(d',1)$ with respect to $d'$ is nonnegative. Thus, $d' = d$ is the optimal. By Lemma E.2, $\boldsymbol{b} = \mathbf{1}_{d+1}$. In addition, from $s_1(d,1), s_3(d,1), s_5(d,1) \geq 0$, $\boldsymbol{A} = \mathbf{1}_{d+1,d}$.

**Adversarial.** Assume $d \geq 2$ and $\epsilon = \epsilon_7$. By Ineq. (A124), $r_1, r_3, r_5 \geq 0$, $r_7 = 0$, and $r_2, r_4, r_6 \leq 0$. Thus, $s_3(d', b_{d+1}), s_4(d', b_{d+1}) \geq 0$ and $s_2(d', b_{d+1}), s_6(d', b_{d+1}), s_8(d', b_{d+1}) \leq 0$. Now,

$$s_1(d',1) = s_1(d',0) \geq \frac{(d-d')(2-\lambda)^2}{4d} \geq 0 \qquad (\because \epsilon = \epsilon_7). \tag{A149}$$

Thus,

$$\mathrm{score}(d', b_{d+1}) = d's_1(d',0) + d's_3(d', b_{d+1}) + (d-d')s_4(d', b_{d+1})$$

$$+ d'(d'-1)\phi(s_5(d', b_{d+1})) + d'(d-d')\phi(s_7(d', b_{d+1})) \tag{A150}$$

$$= d's_1(d',0) + d'r_1 + (d-1)d'r_5 + db_{d+1}r_3$$

$$+ d'(d' - 1)\phi(b_{d+1}r_4 + r_2 + r_5 + (d' - 2)r_6)$$
$$+ d'(d - d')\phi(b_{d+1}r_4 + r_5 + (d' - 1)r_6). \tag{A151}$$

Since $r_4$ is nonpositive, this indicates that score changes by $dr_3 + d'(d - 1)r_4$ at least by switching $b_{d+1}$ to one from zero. Moreover,

$$dr_3 + d'(d - 1)r_4 \geq \frac{(d - 1)(d - d')(2 - \lambda)^2}{4d} \geq 0 \qquad (\because \epsilon = \epsilon_7). \tag{A152}$$

Therefore, $b_{d+1} = 1$ is the optimal. From Ineq. (A124) and $\epsilon = \epsilon_7$, $s_7(d', b_{d+1}) - s_5(d', b_{d+1}) \geq 0$. If $s_5(d', 1), s_7(d', 1) \geq 0$,

$$\frac{\mathrm{d\,score}(d', 1)}{\mathrm{d}d'} = \Theta(d, d', \lambda) - \Theta_+(d, d', \lambda)\epsilon \tag{A153}$$

$$= \Theta(d, \lambda) - \Theta_+(d, \lambda)d' \qquad (\because \epsilon = \epsilon_7) \tag{A154}$$

$$\geq 0 \qquad (\because d' \leq d_{s_5}), \tag{A155}$$

where

$$s_5(d', 1) \geq 0 \iff d' \leq \frac{3d\lambda^2 - 6d\lambda + 2\lambda^2 - 18\lambda + 24}{6\lambda(\lambda - 2)} =: d_{s_5}. \tag{A156}$$

When $s_5(d', 1) \leq 0, s_7(d', 1) \geq 0$, then $\frac{\mathrm{d\,score}(d', 1)}{\mathrm{d}d'} \geq 0$ similarly holds. If $s_5(d', 1), s_7(d', 1) \leq 0$, $\frac{\mathrm{d\,score}(d', 1)}{\mathrm{d}d'} \geq 0$ for $d' \leq d - 1$. Comparing $\mathrm{score}(d', 1)$ with $d' = d - 1$ and $d' = d$, we obtain $\mathrm{score}(d, 1) \geq \mathrm{score}(d - 1, 1)$. In summary, $d' = d$ is the optimal. By Lemma E.2, $b = 1_{d+1}$. In addition, from $s_3(d, 1) \geq 0$, $s_1(d, 1) = 0$, and $s_5(d, 1) < 0$, $A = [I_d \ 0_d]^\top$.

**Strong Adversarial.** Assume $d \geq 2$ and $\epsilon \geq \epsilon_1$. By Ineq. (A124), $r_1, \ldots, r_7$ are nonpositive. Thus, $s_1(d', b_{d+1}), \ldots, s_8(d', b_{d+1})$ are nonpositive. Therefore, $d' = 0$ and $b_{d+1} = 0$ are the optimal. By Lemma E.2, $b = 0_{d+1}$ and $A = 0_{d+1,d}$. $\qquad\qquad\square$

## F  PROOF OF THEOREMS 3.5 AND 3.6 (ROBUSTNESS)

For notational convenience, we occasionally describe representations and equations under the assumption that $\mathcal{S}_{\mathrm{rob}} := \{1, \ldots, d_{\mathrm{rob}}\}$, $\mathcal{S}_{\mathrm{vul}} := \{d_{\mathrm{rob}} + 1, \ldots, d_{\mathrm{rob}} + d_{\mathrm{vul}}\}$, and $\mathcal{S}_{\mathrm{irr}} := \{d_{\mathrm{rob}} + d_{\mathrm{vul}} + 1, \ldots, d_{\mathrm{rob}} + d_{\mathrm{vul}} + d_{\mathrm{irr}}\}$. This assumption is made without loss of generality.

We use *uniform* big-O and -Theta notation. Denote $f(x) = \mathcal{O}(g(x))$ if there exists a positive constant $C > 0$ such that $|f(x)| \leq C|g(x)|$ for *every* $x$ in the domain. Denote $f(x) = \Theta(g(x))$ if there exist $C_1, C_2 > 0$ such that $C_1|g(x)| \leq |f(x)| \leq C_2|g(x)|$ for *every* $x$ in the domain.

For notational simplicity, we abbreviate the following matrix:

$$\begin{bmatrix} C_1\alpha \\ C_2\alpha \\ \vdots \\ C_{d_{\mathrm{rob}}}\alpha \\ C_{d_{\mathrm{rob}}+1}\beta \\ \vdots \\ C_{d_{\mathrm{rob}}+d_{\mathrm{vul}}}\beta \\ C_{d_{\mathrm{rob}}+d_{\mathrm{vul}}+1}\gamma \\ \vdots \\ C_{d_{\mathrm{rob}}+d_{\mathrm{vul}}+d_{\mathrm{irr}}}\gamma \end{bmatrix} \qquad \text{as} \qquad \begin{bmatrix} C_i\alpha \\ C_i\beta \\ C_i\gamma \end{bmatrix}. \tag{A157}$$

**Theorem 3.5** (Standard pretraining case). There exist a constant $C > 0$ and a strictly positive function $g(d_{\mathrm{rob}}, d_{\mathrm{vul}}, d_{\mathrm{irr}}, \alpha, \beta, \gamma)$ such that

$$\mathbb{E}_{\{(x_n, y_n)\}_{n=1}^{N+1} \overset{\mathrm{i.i.d.}}{\sim} \mathcal{D}^{\mathrm{te}}} \left[ \min_{\|\Delta\|_\infty \leq \epsilon} y_{N+1}[f(Z_\Delta; P^{\mathrm{std}}, Q^{\mathrm{std}})]_{d+1, N+1} \right]$$

$$\leq \quad g(d_{\mathrm{rob}}, d_{\mathrm{vul}}, d_{\mathrm{irr}}, \alpha, \beta, \gamma) \quad \Big\{ \quad \underbrace{C(d_{\mathrm{rob}}\alpha + d_{\mathrm{vul}}\beta)}_{\text{Prediction for original data}} \quad - \quad \underbrace{(d_{\mathrm{rob}} + d_{\mathrm{vul}} + d_{\mathrm{irr}})\epsilon}_{\text{Adversarial effect}} \quad \Big\}. \quad (8)$$

*Proof.* Since $\boldsymbol{b} = \boldsymbol{1}_{d+1}$, $\boldsymbol{A} = \boldsymbol{1}_{d+1,d}$, and $\boldsymbol{Z_\Delta} \boldsymbol{M} \boldsymbol{Z_\Delta}^\top$ is positive semidefinite, every entry in $\boldsymbol{b}^\top \boldsymbol{Z_\Delta} \boldsymbol{M} \boldsymbol{Z_\Delta}^\top \boldsymbol{A}$ is nonnegative. Thus, we can solve the inner minimization as

$$\min_{\|\boldsymbol{\Delta}\|_\infty \leq \epsilon} y_{N+1}[\boldsymbol{f}(\boldsymbol{Z_\Delta}; \boldsymbol{P}, \boldsymbol{Q})]_{d+1, N+1} = \min_{\|\boldsymbol{\Delta}\|_\infty \leq \epsilon} \frac{1}{N} \boldsymbol{b}^\top \boldsymbol{Z_\Delta} \boldsymbol{M} \boldsymbol{Z_\Delta}^\top \boldsymbol{A} y_{N+1}(\boldsymbol{x}_{N+1} + \boldsymbol{\Delta}) \quad (A158)$$

$$= \frac{1}{N} \boldsymbol{b}^\top \boldsymbol{Z_\Delta} \boldsymbol{M} \boldsymbol{Z_\Delta}^\top \boldsymbol{A} (y_{N+1}\boldsymbol{x}_{N+1} - \epsilon \boldsymbol{1}_d). \quad (A159)$$

Using $(\boldsymbol{x}, y) \sim \mathcal{D}^{\mathrm{te}}$,

$$\mathbb{E}\left[\frac{1}{N} \boldsymbol{Z_\Delta} \boldsymbol{M} \boldsymbol{Z_\Delta}^\top\right]$$

$$= \begin{bmatrix} \mathbb{E}[\boldsymbol{x}\boldsymbol{x}^\top] & \mathbb{E}[y\boldsymbol{x}] \\ \mathbb{E}[y\boldsymbol{x}^\top] & 1 \end{bmatrix} \quad (A160)$$

$$= \begin{bmatrix} \mathbb{E}[y\boldsymbol{x}]\mathbb{E}[y\boldsymbol{x}^\top] & \mathbb{E}[y\boldsymbol{x}] \\ \mathbb{E}[y\boldsymbol{x}^\top] & 1 \end{bmatrix} + \begin{bmatrix} \mathbb{E}[(y\boldsymbol{x} - \mathbb{E}[y\boldsymbol{x}])(y\boldsymbol{x} - \mathbb{E}[y\boldsymbol{x}])^\top] & \boldsymbol{0}_d \\ \boldsymbol{0}_d^\top & 0 \end{bmatrix}. \quad (A161)$$

Since the second term is positive semidefinite,

$$\mathbb{E}\left[\frac{1}{N} \boldsymbol{1}_{d+1}^\top \boldsymbol{Z_\Delta} \boldsymbol{M} \boldsymbol{Z_\Delta}^\top \boldsymbol{1}_{d+1}\right]$$

$$= \boldsymbol{1}_{d+1}^\top \left( \begin{bmatrix} \mathbb{E}[y\boldsymbol{x}]\mathbb{E}[y\boldsymbol{x}^\top] & \mathbb{E}[y\boldsymbol{x}] \\ \mathbb{E}[y\boldsymbol{x}^\top] & 1 \end{bmatrix} + \begin{bmatrix} \mathbb{E}[(y\boldsymbol{x} - \mathbb{E}[y\boldsymbol{x}])(y\boldsymbol{x} - \mathbb{E}[y\boldsymbol{x}])^\top] & \boldsymbol{0}_d \\ \boldsymbol{0}_d^\top & 0 \end{bmatrix} \right) \boldsymbol{1}_{d+1} \quad (A162)$$

$$\geq \boldsymbol{1}_{d+1}^\top \begin{bmatrix} \mathbb{E}[y\boldsymbol{x}^\top]\mathbb{E}[y\boldsymbol{x}] & \mathbb{E}[y\boldsymbol{x}] \\ \mathbb{E}[y\boldsymbol{x}^\top] & 1 \end{bmatrix} \boldsymbol{1}_{d+1}. \quad (A163)$$

Since every entry of $\mathbb{E}[y\boldsymbol{x}^\top]\mathbb{E}[y\boldsymbol{x}]$ and $\mathbb{E}[y\boldsymbol{x}]$ is nonnegative,

$$\mathbb{E}\left[\frac{1}{N} \boldsymbol{1}_{d+1}^\top \boldsymbol{Z_\Delta} \boldsymbol{M} \boldsymbol{Z_\Delta}^\top \boldsymbol{1}_{d+1}\right] \geq \boldsymbol{1}_{d+1}^\top \begin{bmatrix} \mathbb{E}[y\boldsymbol{x}^\top]\mathbb{E}[y\boldsymbol{x}] & \mathbb{E}[y\boldsymbol{x}] \\ \mathbb{E}[y\boldsymbol{x}^\top] & 1 \end{bmatrix} \boldsymbol{1}_{d+1} \geq 1. \quad (A164)$$

Representing $\mathbb{E}[\boldsymbol{b}^\top \boldsymbol{Z_\Delta} \boldsymbol{M} \boldsymbol{Z_\Delta}^\top \boldsymbol{A}/N] = [g(d_{\mathrm{rob}}, d_{\mathrm{vul}}, d_{\mathrm{irr}}, \alpha, \beta, \gamma) \quad \cdots \quad g(d_{\mathrm{rob}}, d_{\mathrm{vul}}, d_{\mathrm{irr}}, \alpha, \beta, \gamma)]$ using some positive function $g(d_{\mathrm{rob}}, d_{\mathrm{vul}}, d_{\mathrm{irr}}, \alpha, \beta, \gamma) > 0$, there exists a positive constant $C > 0$ such that

$$\mathbb{E}\left[\frac{1}{N} \boldsymbol{b}^\top \boldsymbol{Z_\Delta} \boldsymbol{M} \boldsymbol{Z_\Delta}^\top \boldsymbol{A}(y_{N+1}\boldsymbol{x}_{N+1} - \epsilon \boldsymbol{1}_d)\right]$$

$$= \begin{bmatrix} g(d_{\mathrm{rob}}, d_{\mathrm{vul}}, d_{\mathrm{irr}}, \alpha, \beta, \gamma) \\ \vdots \\ g(d_{\mathrm{rob}}, d_{\mathrm{vul}}, d_{\mathrm{irr}}, \alpha, \beta, \gamma) \end{bmatrix}^\top (\mathbb{E}[y_{N+1}\boldsymbol{x}_{N+1}] - \epsilon \boldsymbol{1}_d) \quad (A165)$$

$$= g(d_{\mathrm{rob}}, d_{\mathrm{vul}}, d_{\mathrm{irr}}, \alpha, \beta, \gamma)(\Theta(d_{\mathrm{rob}}\alpha + d_{\mathrm{vul}}\beta) - d\epsilon) \quad (A166)$$

$$\leq g(d_{\mathrm{rob}}, d_{\mathrm{vul}}, d_{\mathrm{irr}}, \alpha, \beta, \gamma)(C(d_{\mathrm{rob}}\alpha + d_{\mathrm{vul}}\beta) - (d_{\mathrm{rob}} + d_{\mathrm{vul}} + d_{\mathrm{irr}})\epsilon). \quad (A167)$$

$\square$

**Theorem 3.6** (Adversarial pretraining case). Suppose that $q_{\mathrm{rob}}$ and $q_{\mathrm{vul}}$ defined in Assumption 3.2 are sufficiently small. There exist constants $C_1, C_2 > 0$ such that

$$\mathbb{E}_{\{(\boldsymbol{x}_n, y_n)\}_{n=1}^{N+1} \overset{\mathrm{i.i.d.}}{\sim} \mathcal{D}^{\mathrm{te}}}\left[\min_{\|\boldsymbol{\Delta}\|_\infty \leq \epsilon} y_{N+1}[\boldsymbol{f}(\boldsymbol{Z_\Delta}; \boldsymbol{P}^{\mathrm{adv}}, \boldsymbol{Q}^{\mathrm{adv}})]_{d+1, N+1}\right]$$

$$\geq \underbrace{C_1(d_{\mathrm{rob}}\alpha + d_{\mathrm{vul}}\beta + 1)(d_{\mathrm{rob}}\alpha^2 + d_{\mathrm{vul}}\beta^2)}_{\text{Prediction for original data}}$$

$$- C_2 \left\{ (d_{\text{rob}}\alpha + d_{\text{vul}}\beta + 1)\left(d_{\text{rob}}\alpha + d_{\text{vul}}\beta + \frac{d_{\text{irr}}\gamma}{\sqrt{N}}\right) + d_{\text{irr}}\left(\sqrt{\frac{d_{\text{irr}}}{N}} + 1\right)\gamma^2 \right\}\epsilon . \quad (9)$$

$$\underbrace{\phantom{- C_2 \left\{ (d_{\text{rob}}\alpha + d_{\text{vul}}\beta + 1)\left(d_{\text{rob}}\alpha + d_{\text{vul}}\beta + \frac{d_{\text{irr}}\gamma}{\sqrt{N}}\right) + d_{\text{irr}}\left(\sqrt{\frac{d_{\text{irr}}}{N}} + 1\right)\gamma^2 \right\}\epsilon}}_{\text{Adversarial effect}}$$

*Proof.* This is the special case of the following theorem. $\qquad\square$

**Theorem F.1** (General case of Theorem 3.6). There exist constants $C, C', C'' > 0$ such that

$$\mathbb{E}_{\{(\boldsymbol{x}_n, y_n)\}_{n=1}^{N+1} \overset{\text{i.i.d.}}{\sim} \mathcal{D}^{\text{te}}}\left[\min_{\|\boldsymbol{\Delta}\|_\infty \leq \epsilon} y_{N+1}[f(\boldsymbol{Z}_{\boldsymbol{\Delta}}; \boldsymbol{P}^{\text{adv}}, \boldsymbol{Q}^{\text{adv}})]_{d+1, N+1}\right]$$

$$\geq C(d_{\text{rob}}\alpha + d_{\text{vul}}\beta)\left\{(1 - cq_{\text{rob}})d_{\text{rob}}\alpha^2 + (1 - cq_{\text{vul}})d_{\text{vul}}\beta^2\right\} + C'(d_{\text{rob}}\alpha^2 + d_{\text{vul}}\beta^2)$$

$$- C'''\left\{(d_{\text{rob}}\alpha + d_{\text{vul}}\beta + 1)\left(d_{\text{rob}}\alpha + d_{\text{vul}}\beta + \frac{d_{\text{irr}}\gamma}{\sqrt{N}}\right) + d_{\text{irr}}\left(\sqrt{\frac{d_{\text{irr}}}{N}} + 1\right)\gamma^2\right\}\epsilon, \quad (A168)$$

where

$$c := \frac{(\max_{i \in \mathcal{S}_{\text{rob}} \cup \mathcal{S}_{\text{vul}}} C_i)(\max_{i \in \mathcal{S}_{\text{rob}} \cup \mathcal{S}_{\text{vul}}} C_{i,2})}{\min_{i \in \mathcal{S}_{\text{rob}} \cup \mathcal{S}_{\text{vul}}} C_i^3}. \quad (A169)$$

In particular, if there exists a constant $C''' > 0$ such that $1 - cq_{\text{rob}} \geq C'''$ and $1 - cq_{\text{vul}} \geq C'''$, then there exist constants $C_1, C_2 > 0$ such that Ineq. (9) holds.

*Proof.* Similarly to Eq. (A33), we can solve the minimization as

$$\min_{\|\boldsymbol{\Delta}\|_\infty \leq \epsilon} y_{N+1}[\boldsymbol{f}(\boldsymbol{Z}_{\boldsymbol{\Delta}}; \boldsymbol{P}, \boldsymbol{Q})]_{d+1, N+1}$$

$$= \min_{\|\boldsymbol{\Delta}\|_\infty \leq \epsilon} \frac{1}{N}\boldsymbol{b}^\top \boldsymbol{Z}_{\boldsymbol{\Delta}} \boldsymbol{M} \boldsymbol{Z}_{\boldsymbol{\Delta}}^\top \boldsymbol{A} y_{N+1}(\boldsymbol{x}_{N+1} + \boldsymbol{\Delta}) \quad (A170)$$

$$= \frac{1}{N}\boldsymbol{b}^\top \boldsymbol{Z}_{\boldsymbol{\Delta}} \boldsymbol{M} \boldsymbol{Z}_{\boldsymbol{\Delta}}^\top \boldsymbol{A} y_{N+1}\boldsymbol{x}_{N+1} - \epsilon\left\|\frac{1}{N}\boldsymbol{b}^\top \boldsymbol{Z}_{\boldsymbol{\Delta}} \boldsymbol{M} \boldsymbol{Z}_{\boldsymbol{\Delta}}^\top \boldsymbol{A}\right\|_1. \quad (A171)$$

By Eq. (A161), we can rearrange the first term as

$$\mathbb{E}\left[\frac{1}{N}\boldsymbol{b}^\top \boldsymbol{Z}_{\boldsymbol{\Delta}} \boldsymbol{M} \boldsymbol{Z}_{\boldsymbol{\Delta}}^\top \boldsymbol{A} y_{N+1}\boldsymbol{x}_{N+1}\right]$$

$$= \boldsymbol{1}_{d+1}^\top \begin{bmatrix} \mathbb{E}[y\boldsymbol{x}]\mathbb{E}[y\boldsymbol{x}^\top] \\ \mathbb{E}[y\boldsymbol{x}^\top] \end{bmatrix}\mathbb{E}[y_{N+1}\boldsymbol{x}_{N+1}] + \boldsymbol{1}_d^\top\mathbb{E}[(y\boldsymbol{x} - \mathbb{E}[y\boldsymbol{x}])(y\boldsymbol{x} - \mathbb{E}[y\boldsymbol{x}])^\top]\mathbb{E}[y_{N+1}\boldsymbol{x}_{N+1}]. \quad (A172)$$

The first term of Eq. (A172) can be rearranged as

$$\boldsymbol{1}_{d+1}^\top \begin{bmatrix} \mathbb{E}[y\boldsymbol{x}]\mathbb{E}[y\boldsymbol{x}^\top] \\ \mathbb{E}[y\boldsymbol{x}^\top] \end{bmatrix}\mathbb{E}[y_{N+1}\boldsymbol{x}_{N+1}]$$

$$= \boldsymbol{1}_{d+1}^\top \begin{bmatrix} C_i C_j \alpha^2 & C_i C_j \alpha\beta & \boldsymbol{0} \\ C_i C_j \alpha\beta & C_i C_j \beta^2 & \boldsymbol{0} \\ \boldsymbol{0} & \boldsymbol{0} & C_i^2 \gamma^2 \boldsymbol{I} \\ C_i\alpha & C_i\beta & \boldsymbol{0} \end{bmatrix}\begin{bmatrix} C_i\alpha \\ C_i\beta \\ \boldsymbol{0} \end{bmatrix} \quad (A173)$$

$$= \left(\sum_{i \in \mathcal{S}_{\text{rob}}} C_i\alpha + \sum_{i \in \mathcal{S}_{\text{vul}}} C_i\beta + 1\right)\left(\sum_{i \in \mathcal{S}_{\text{rob}}} C_i^2\alpha^2 + \sum_{i \in \mathcal{S}_{\text{vul}}} C_i^2\beta^2\right) \quad (A174)$$

$$= \left(\min_{i \in \mathcal{S}_{\text{rob}} \cup \mathcal{S}_{\text{vul}}} C_i^3\right)(d_{\text{rob}}\alpha + d_{\text{vul}}\beta)(d_{\text{rob}}\alpha^2 + d_{\text{vul}}\beta^2) + \sum_{i \in \mathcal{S}_{\text{rob}}} C_i^2\alpha^2 + \sum_{i \in \mathcal{S}_{\text{vul}}} C_i^2\beta^2. \quad (A175)$$

Consider the second term of Eq. (A172). Now,

$$|\mathbb{E}[(yx_i - \mathbb{E}[yx_i])(yx_j - \mathbb{E}[yx_j])]|$$

$$
\leq \begin{cases} \sqrt{C_{i,2}}\sqrt{C_{j,2}}\alpha^2 & (i,j \in \mathcal{S}_{\mathrm{rob}}) \\ \sqrt{C_{i,2}}\sqrt{C_{j,2}}\beta^2 & (i,j \in \mathcal{S}_{\mathrm{vul}}) \\ \sqrt{C_{i,2}}\sqrt{C_{j,2}}\alpha\beta & (i \in \mathcal{S}_{\mathrm{rob}} \land j \in \mathcal{S}_{\mathrm{vul}}) \lor (i \in \mathcal{S}_{\mathrm{vul}} \land j \in \mathcal{S}_{\mathrm{rob}}) \end{cases}. \tag{A176}
$$

Let

$$
\mathcal{S} := \left\{ i \in \mathcal{S}_{\mathrm{rob}} \cup \mathcal{S}_{\mathrm{vul}} \mid \sum_{j \in \mathcal{S}_{\mathrm{rob}} \cup \mathcal{S}_{\mathrm{vul}}} \mathbb{E}[(yx_i - \mathbb{E}[yx_i])(yx_j - \mathbb{E}[yx_j])] < 0 \right\}. \tag{A177}
$$

The second term of Eq. (A172) can be computed as

$$
\mathbf{1}_d^\top \mathbb{E}[(y\boldsymbol{x} - \mathbb{E}[y\boldsymbol{x}])(y\boldsymbol{x} - \mathbb{E}[y\boldsymbol{x}])^\top] \mathbb{E}[y_{N+1}\boldsymbol{x}_{N+1}]
$$

$$
\geq - \begin{bmatrix} \left. \begin{array}{c} \sqrt{C_{i,2}}\alpha\left(\sum_{j \in \mathcal{S}_{\mathrm{rob}}} \sqrt{C_{j,2}}\alpha + \sum_{j \in \mathcal{S}_{\mathrm{vul}}} \sqrt{C_{j,2}}\beta\right) \\ \vdots \\ \sqrt{C_{i,2}}\alpha\left(\sum_{j \in \mathcal{S}_{\mathrm{rob}}} \sqrt{C_{j,2}}\alpha + \sum_{j \in \mathcal{S}_{\mathrm{vul}}} \sqrt{C_{j,2}}\beta\right) \end{array} \right\} \leq q_{\mathrm{rob}}d_{\mathrm{rob}} \\ \mathbf{0} \\ \left. \begin{array}{c} \sqrt{C_{i,2}}\beta\left(\sum_{j \in \mathcal{S}_{\mathrm{rob}}} \sqrt{C_{j,2}}\alpha + \sum_{j \in \mathcal{S}_{\mathrm{vul}}} \sqrt{C_{j,2}}\beta\right) \\ \vdots \\ \sqrt{C_{i,2}}\beta\left(\sum_{j \in \mathcal{S}_{\mathrm{rob}}} \sqrt{C_{j,2}}\alpha + \sum_{j \in \mathcal{S}_{\mathrm{vul}}} \sqrt{C_{j,2}}\beta\right) \end{array} \right\} \leq q_{\mathrm{vul}}d_{\mathrm{vul}} \\ \mathbf{0} \end{bmatrix}^\top \begin{bmatrix} C_i\alpha \\ C_i\beta \\ \mathbf{0} \end{bmatrix} \tag{A178}
$$

$$
= -\left(\sum_{i \in \mathcal{S}_{\mathrm{rob}}} \sqrt{C_{i,2}}\alpha + \sum_{i \in \mathcal{S}_{\mathrm{vul}}} \sqrt{C_{i,2}}\beta\right)
$$

$$
\times \left(\sum_{i \in \mathcal{S}_{\mathrm{rob}} \cap \mathcal{S}} C_i \sqrt{C_{i,2}}\alpha^2 + \sum_{i \in \mathcal{S}_{\mathrm{vul}} \cap \mathcal{S}} C_i \sqrt{C_{i,2}}\beta^2\right) \tag{A179}
$$

$$
\geq -\left(\max_{i \in \mathcal{S}_{\mathrm{rob}} \cup \mathcal{S}_{\mathrm{vul}}} \sqrt{C_{i,2}}\right)\left(\max_{i \in (\mathcal{S}_{\mathrm{rob}} \cup \mathcal{S}_{\mathrm{vul}}) \cap \mathcal{S}} C_i \sqrt{C_{i,2}}\right)
$$

$$
\times (d_{\mathrm{rob}}\alpha + d_{\mathrm{vul}}\beta)(q_{\mathrm{rob}}d_{\mathrm{rob}}\alpha^2 + q_{\mathrm{vul}}d_{\mathrm{vul}}\beta^2) \tag{A180}
$$

$$
\geq -\left(\max_{i \in \mathcal{S}_{\mathrm{rob}} \cup \mathcal{S}_{\mathrm{vul}}} C_i\right)\left(\max_{i \in \mathcal{S}_{\mathrm{rob}} \cup \mathcal{S}_{\mathrm{vul}}} C_{i,2}\right)(d_{\mathrm{rob}}\alpha + d_{\mathrm{vul}}\beta)(q_{\mathrm{rob}}d_{\mathrm{rob}}\alpha^2 + q_{\mathrm{vul}}d_{\mathrm{vul}}\beta^2). \tag{A181}
$$

By Lemma F.2, we can compute the second term as

$$
\mathbb{E}\left[\left\|\frac{1}{N}\boldsymbol{b}^\top \boldsymbol{Z_\Delta} \boldsymbol{M} \boldsymbol{Z_\Delta}^\top \boldsymbol{A}\right\|_1\right]
$$

$$
= \mathcal{O}\left((d_{\mathrm{rob}}\alpha + d_{\mathrm{vul}}\beta + 1)\left(d_{\mathrm{rob}}\alpha + d_{\mathrm{vul}}\beta + \frac{d_{\mathrm{irr}}\gamma}{\sqrt{N}}\right) + d_{\mathrm{irr}}\left(\sqrt{\frac{d_{\mathrm{irr}}}{N}} + 1\right)\gamma^2\right). \tag{A182}
$$

Finally,

$$
\mathbb{E}\left[\frac{1}{N}\boldsymbol{b}^\top \boldsymbol{Z_\Delta} \boldsymbol{M} \boldsymbol{Z_\Delta}^\top \boldsymbol{A} y_{N+1}\boldsymbol{x}_{N+1}\right] - \epsilon\mathbb{E}\left[\left\|\frac{1}{N}\boldsymbol{b}^\top \boldsymbol{Z_\Delta} \boldsymbol{M} \boldsymbol{Z_\Delta}^\top \boldsymbol{A}\right\|_1\right]
$$

$$
\geq \left(\min_{i \in \mathcal{S}_{\mathrm{rob}} \cup \mathcal{S}_{\mathrm{vul}}} C_i^3\right)(d_{\mathrm{rob}}\alpha + d_{\mathrm{vul}}\beta)(d_{\mathrm{rob}}\alpha^2 + d_{\mathrm{vul}}\beta^2) + \sum_{i \in \mathcal{S}_{\mathrm{rob}}} C_i^2\alpha^2 + \sum_{i \in \mathcal{S}_{\mathrm{vul}}} C_i^2\beta^2
$$

$$
- \left(\max_{i \in \mathcal{S}_{\mathrm{rob}} \cup \mathcal{S}_{\mathrm{vul}}} C_i\right)\left(\max_{i \in \mathcal{S}_{\mathrm{rob}} \cup \mathcal{S}_{\mathrm{vul}}} C_{i,2}\right)(d_{\mathrm{rob}}\alpha + d_{\mathrm{vul}}\beta)(q_{\mathrm{rob}}d_{\mathrm{rob}}\alpha^2 + q_{\mathrm{vul}}d_{\mathrm{vul}}\beta^2)
$$

$$
+ \mathcal{O}\left((d_{\mathrm{rob}}\alpha + d_{\mathrm{vul}}\beta + 1)\left(d_{\mathrm{rob}}\alpha + d_{\mathrm{vul}}\beta + \frac{d_{\mathrm{irr}}\gamma}{\sqrt{N}}\right) + d_{\mathrm{irr}}\left(\sqrt{\frac{d_{\mathrm{irr}}}{N}} + 1\right)\gamma^2\right). \tag{A183}
$$

$\square$

**Lemma F.2.** If $(\boldsymbol{x}_1, y_1), \ldots, (\boldsymbol{x}_N, y_N)$ are i.i.d. and follow $\mathcal{D}^{\text{te}}$, then

$$\mathbb{E}\left[\left\|\frac{1}{N}\boldsymbol{b}^\top \boldsymbol{Z}_\Delta \boldsymbol{M} \boldsymbol{Z}_\Delta^\top \boldsymbol{A}\right\|_1\right]$$

$$= \mathcal{O}\left((d_{\text{rob}}\alpha + d_{\text{vul}}\beta + 1)\left(d_{\text{rob}}\alpha + d_{\text{vul}}\beta + \frac{d_{\text{irr}}\gamma}{\sqrt{N}}\right) + d_{\text{irr}}\left(\sqrt{\frac{d_{\text{irr}}}{N}} + 1\right)\gamma^2\right), \quad \text{(A184)}$$

where $\boldsymbol{b} = \boldsymbol{1}_{d+1}$ and $\boldsymbol{A}^\top := [\boldsymbol{I}_d \quad \boldsymbol{0}_d]$.

*Proof.* We can rearrange the given expectation as

$$\mathbb{E}\left[\left\|\frac{1}{N}\boldsymbol{b}^\top \boldsymbol{Z}_\Delta \boldsymbol{M} \boldsymbol{Z}_\Delta^\top \boldsymbol{A}\right\|_1\right] = \mathbb{E}\left[\left\|\frac{1}{N}\boldsymbol{1}_{d+1}^\top \begin{bmatrix} \sum_{n=1}^N \boldsymbol{x}_n \boldsymbol{x}_n^\top & \sum_{n=1}^N y_n \boldsymbol{x}_n \\ \sum_{n=1}^N y_n \boldsymbol{x}_n^\top & N \end{bmatrix} \begin{bmatrix} \boldsymbol{I}_d \\ \boldsymbol{0}_d^\top \end{bmatrix}\right\|_1\right] \quad \text{(A185)}$$

$$= \mathbb{E}\left[\left\|\frac{1}{N}\boldsymbol{1}_{d+1}^\top \begin{bmatrix} \sum_{n=1}^N \boldsymbol{x}_n \boldsymbol{x}_n^\top \\ \sum_{n=1}^N y_n \boldsymbol{x}_n^\top \end{bmatrix}\right\|_1\right] \quad \text{(A186)}$$

$$= \sum_{i=1}^d \mathbb{E}\left[\left|\frac{1}{N}\sum_{n=1}^N \left(y_n + \sum_{j=1}^d x_{n,j}\right) x_{n,i}\right|\right]. \quad \text{(A187)}$$

By the Lyapunov inequality, for $N+1$ i.i.d. random variables $X, X_1, \ldots, X_N$,

$$\mathbb{E}\left[\left|\frac{1}{N}\sum_{n=1}^N X_n\right|\right] \leq \sqrt{\mathbb{E}\left[\left(\frac{1}{N}\sum_{n=1}^N X_n\right)^2\right]} = \sqrt{\frac{1}{N}\mathbb{E}[X^2] + \frac{N-1}{N}\mathbb{E}[X]^2}. \quad \text{(A188)}$$

Thus, using $(\boldsymbol{x}, y) \sim \mathcal{D}^{\text{te}}$,

$$\sum_{i=1}^d \mathbb{E}\left[\left|\frac{1}{N}\sum_{n=1}^N \left(y_n + \sum_{j=1}^d x_{n,j}\right) x_{n,i}\right|\right]$$

$$\leq \sum_{i=1}^d \sqrt{\frac{1}{N}\mathbb{E}\left[\left(y + \sum_{j=1}^d x_j\right)^2 x_i^2\right] + \frac{N-1}{N}\mathbb{E}\left[\left(y + \sum_{j=1}^d x_j\right) x_i\right]^2}. \quad \text{(A189)}$$

From Lemma F.3, we can compute the second term of using

$$\mathbb{E}\left[\left(y + \sum_{j=1}^d x_j\right) x_i\right] = \mathbb{E}[yx_i] + \sum_{j=1}^d \mathbb{E}[x_j x_i] \quad \text{(A190)}$$

$$= \begin{cases} \mathcal{O}(\alpha(d_{\text{rob}}\alpha + d_{\text{vul}}\beta + 1)) & (i \in \mathcal{S}_{\text{rob}}) \\ \mathcal{O}(\beta(d_{\text{rob}}\alpha + d_{\text{vul}}\beta + 1)) & (i \in \mathcal{S}_{\text{vul}}) \\ \mathcal{O}(\gamma^2) & (i \in \mathcal{S}_{\text{irr}}) \end{cases} . \quad \text{(A191)}$$

From Lemma F.3, we can compute the first term of using

$$\mathbb{E}\left[\left(y + \sum_{j=1}^d x_j\right)^2 x_i^2\right] = \mathbb{E}[x_i^2] + 2\sum_{j=1}^d \mathbb{E}[yx_j x_i^2] + \sum_{j,k=1}^d \mathbb{E}[x_j x_k x_i^2] \quad \text{(A192)}$$

$$= \begin{cases} \mathcal{O}(\alpha^2\{(d_{\text{rob}}\alpha + d_{\text{vul}}\beta + 1)^2 + d_{\text{irr}}\gamma^2\}) & (i \in \mathcal{S}_{\text{rob}}) \\ \mathcal{O}(\beta^2\{(d_{\text{rob}}\alpha + d_{\text{vul}}\beta + 1)^2 + d_{\text{irr}}\gamma^2\}) & (i \in \mathcal{S}_{\text{vul}}) \\ \mathcal{O}(\gamma^2\{(d_{\text{rob}}\alpha + d_{\text{vul}}\beta + 1)^2 + d_{\text{irr}}\gamma^2\}) & (i \in \mathcal{S}_{\text{irr}}) \end{cases} . \quad \text{(A193)}$$

Thus,

$$\sum_{i=1}^{d} \sqrt{\frac{1}{N}\mathbb{E}\left[\left(y + \sum_{j=1}^{d} x_j\right)^2 x_i^2\right] + \frac{N-1}{N}\mathbb{E}\left[\left(y + \sum_{j=1}^{d} x_j\right) x_i\right]^2}$$

$$= \mathcal{O}\Bigg( d_{\text{rob}}\left(\alpha(d_{\text{rob}}\alpha + d_{\text{vul}}\beta + 1) + \sqrt{\frac{d_{\text{irr}}}{N}}\alpha\gamma\right)$$

$$+ d_{\text{vul}}\left(\beta(d_{\text{rob}}\alpha + d_{\text{vul}}\beta + 1) + \sqrt{\frac{d_{\text{irr}}}{N}}\beta\gamma\right)$$

$$+ d_{\text{irr}}\left(\gamma^2 + \frac{\gamma}{\sqrt{N}}\left((d_{\text{rob}}\alpha + d_{\text{vul}}\beta + 1) + \sqrt{d_{\text{irr}}}\gamma\right)\right)\Bigg) \tag{A194}$$

$$= \mathcal{O}\Bigg( (d_{\text{rob}}\alpha + d_{\text{vul}}\beta + 1)\left(d_{\text{rob}}\alpha + d_{\text{vul}}\beta + \frac{d_{\text{irr}}\gamma}{\sqrt{N}}\right) + d_{\text{irr}}\left(\sqrt{\frac{d_{\text{irr}}}{N}} + 1\right)\gamma^2\Bigg). \tag{A195}$$

$\square$

**Lemma F.3.** If $(\boldsymbol{x}, y) \sim \mathcal{D}^{\text{te}}$, then

(a)

$$\mathbb{E}[x_j x_i] = \begin{cases} \mathcal{O}(\alpha^2) & (i, j \in \mathcal{S}_{\text{rob}}) \\ \mathcal{O}(\beta^2) & (i, j \in \mathcal{S}_{\text{vul}}) \\ \mathcal{O}(\gamma^2) & (i = j) \wedge (i, j \in \mathcal{S}_{\text{irr}}) \\ \mathcal{O}(\alpha\beta) & (i \in \mathcal{S}_{\text{rob}} \wedge j \in \mathcal{S}_{\text{vul}}) \vee (i \in \mathcal{S}_{\text{vul}} \wedge j \in \mathcal{S}_{\text{rob}}) \\ 0 & (i \neq j) \wedge (i \in \mathcal{S}_{\text{irr}} \vee j \in \mathcal{S}_{\text{irr}}) \end{cases} . \tag{A196}$$

(b)

$$\mathbb{E}[y x_j x_i^2] = \begin{cases} \mathcal{O}(\alpha^3) & (i, j \in \mathcal{S}_{\text{rob}}) \\ \mathcal{O}(\beta^3) & (i, j \in \mathcal{S}_{\text{vul}}) \\ \mathcal{O}(\alpha^2\beta) & (i \in \mathcal{S}_{\text{rob}} \wedge j \in \mathcal{S}_{\text{vul}}) \\ \mathcal{O}(\alpha\beta^2) & (i \in \mathcal{S}_{\text{vul}} \wedge j \in \mathcal{S}_{\text{rob}}) \\ \mathcal{O}(\alpha\gamma^2) & (i \in \mathcal{S}_{\text{irr}} \wedge j \in \mathcal{S}_{\text{rob}}) \\ \mathcal{O}(\beta\gamma^2) & (i \in \mathcal{S}_{\text{irr}} \wedge j \in \mathcal{S}_{\text{vul}}) \\ 0 & (j \in \mathcal{S}_{\text{irr}}) \end{cases} . \tag{A197}$$

(c)

$$\mathbb{E}[x_j x_k x_i^2]$$
$$= \begin{cases} \mathcal{O}(\alpha^4) & (i, j, k \in \mathcal{S}_{\text{rob}}) \\ \mathcal{O}(\beta^4) & (i, j, k \in \mathcal{S}_{\text{vul}}) \\ \mathcal{O}(\gamma^4) & (j = k) \wedge (i, j, k \in \mathcal{S}_{\text{irr}}) \\ \mathcal{O}(\alpha^3\beta) & (i \in \mathcal{S}_{\text{rob}}) \wedge \{(j \in \mathcal{S}_{\text{rob}} \wedge k \in \mathcal{S}_{\text{vul}}) \vee (j \in \mathcal{S}_{\text{vul}} \wedge k \in \mathcal{S}_{\text{rob}})\} \\ \mathcal{O}(\alpha\beta^3) & (i \in \mathcal{S}_{\text{vul}}) \wedge \{(j \in \mathcal{S}_{\text{rob}} \wedge k \in \mathcal{S}_{\text{vul}}) \vee (j \in \mathcal{S}_{\text{vul}} \wedge k \in \mathcal{S}_{\text{rob}})\} \\ \mathcal{O}(\alpha^2\beta^2) & (i \in \mathcal{S}_{\text{rob}} \wedge j, k \in \mathcal{S}_{\text{vul}}) \vee (i \in \mathcal{S}_{\text{vul}} \wedge j, k \in \mathcal{S}_{\text{rob}}) \\ \mathcal{O}(\alpha^2\gamma^2) & (i \in \mathcal{S}_{\text{irr}} \wedge j, k \in \mathcal{S}_{\text{rob}}) \vee (j = k \wedge j, k \in d_{\text{irr}} \wedge i \in \mathcal{S}_{\text{rob}}) \\ \mathcal{O}(\beta^2\gamma^2) & (i \in \mathcal{S}_{\text{irr}} \wedge j, k \in \mathcal{S}_{\text{vul}}) \vee (j = k \wedge j, k \in d_{\text{irr}} \wedge i \in \mathcal{S}_{\text{vul}}) \\ \mathcal{O}(\alpha\beta\gamma^2) & (i \in \mathcal{S}_{\text{irr}}) \wedge \{(j \in \mathcal{S}_{\text{rob}} \wedge k \in \mathcal{S}_{\text{vul}}) \vee (j \in \mathcal{S}_{\text{vul}} \wedge k \in \mathcal{S}_{\text{rob}})\} \\ 0 & (j \neq k) \wedge (j \in \mathcal{S}_{\text{irr}} \vee k \in \mathcal{S}_{\text{irr}}) \end{cases} . \tag{A198}$$

*Proof.* We first note that

$$\mathbb{E}[x_i^2] = \mathbb{E}[(yx_i)^2] = \mathbb{E}[(yx_i - \mathbb{E}[yx_i])^2] + \mathbb{E}[yx_i]^2 = \begin{cases} \mathcal{O}(\alpha^2) & (i \in \mathcal{S}_{\text{rob}}) \\ \mathcal{O}(\beta^2) & (i \in \mathcal{S}_{\text{vul}}) \\ \mathcal{O}(\gamma^2) & (i \in \mathcal{S}_{\text{irr}}) \end{cases}, \qquad \text{(A199)}$$

$$\mathbb{E}[yx_i^3] = \mathbb{E}[(yx_i)^3] \qquad\qquad\qquad\qquad\qquad\qquad\qquad\qquad\qquad \text{(A200)}$$

$$= \mathbb{E}[(yx_i - \mathbb{E}[yx_i])^3] + 3\mathbb{E}[(yx_i)^2]\mathbb{E}[yx_i] - 2\mathbb{E}[yx_i]^3 \qquad \text{(A201)}$$

$$= \begin{cases} \mathcal{O}(\alpha^3) & (i \in \mathcal{S}_{\text{rob}}) \\ \mathcal{O}(\beta^3) & (i \in \mathcal{S}_{\text{vul}}) \\ 0 & (i \in \mathcal{S}_{\text{irr}}) \end{cases}, \qquad\qquad\qquad\qquad \text{(A202)}$$

$$\mathbb{E}[x_i^4] = \mathbb{E}[(yx_i - \mathbb{E}[yx_i])^4] + 4\mathbb{E}[yx_i^3]\mathbb{E}[yx_i] - 6\mathbb{E}[x_i^2]\mathbb{E}[yx_i]^2 + 3\mathbb{E}[yx_i]^4 \qquad \text{(A203)}$$

$$= \begin{cases} \mathcal{O}(\alpha^4) & (i \in \mathcal{S}_{\text{rob}}) \\ \mathcal{O}(\beta^4) & (i \in \mathcal{S}_{\text{vul}}) \\ \mathcal{O}(\gamma^4) & (i \in \mathcal{S}_{\text{irr}}) \end{cases}. \qquad\qquad\qquad\qquad \text{(A204)}$$

(a) For $(i \neq j) \wedge (i \in \mathcal{S}_{\text{irr}} \vee j \in \mathcal{S}_{\text{irr}})$, $\mathbb{E}[x_j x_i] = \mathbb{E}[x_j]\mathbb{E}[x_i] = 0$. Using the Cauthy-Schwarz inequality,

$$\mathbb{E}[x_j x_i] \leq \sqrt{\mathbb{E}[x_j^2]}\sqrt{\mathbb{E}[x_i^2]} \qquad\qquad\qquad\qquad\qquad \text{(A205)}$$

$$= \begin{cases} \mathcal{O}(\alpha^2) & (i, j \in \mathcal{S}_{\text{rob}}) \\ \mathcal{O}(\beta^2) & (i, j \in \mathcal{S}_{\text{vul}}) \\ \mathcal{O}(\gamma^2) & (i, j \in \mathcal{S}_{\text{irr}}) \wedge (i = j) \\ \mathcal{O}(\alpha\beta) & (i \in \mathcal{S}_{\text{rob}} \wedge j \in \mathcal{S}_{\text{vul}}) \vee (i \in \mathcal{S}_{\text{vul}} \wedge j \in \mathcal{S}_{\text{rob}}) \end{cases}. \qquad \text{(A206)}$$

(b) For $j \in \mathcal{S}_{\text{irr}}, j = i$, $\mathbb{E}[yx_j x_i^2] = \mathbb{E}[y]\mathbb{E}[x_i^3] = 0$. For $j \in \mathcal{S}_{\text{irr}}, j \neq i$, $\mathbb{E}[yx_j x_i^2] = \mathbb{E}[x_j]\mathbb{E}[yx_i^2] = 0$. Using the Cauthy-Schwarz inequality,

$$\mathbb{E}[yx_j x_i^2] \leq \sqrt{\mathbb{E}[x_j^2]}\sqrt{\mathbb{E}[x_i^4]} = \begin{cases} \mathcal{O}(\alpha^3) & (i, j \in \mathcal{S}_{\text{rob}}) \\ \mathcal{O}(\beta^3) & (i, j \in \mathcal{S}_{\text{vul}}) \\ \mathcal{O}(\alpha^2\beta) & (i \in \mathcal{S}_{\text{rob}} \wedge j \in \mathcal{S}_{\text{vul}}) \\ \mathcal{O}(\alpha\beta^2) & (i \in \mathcal{S}_{\text{vul}} \wedge j \in \mathcal{S}_{\text{rob}}) \\ \mathcal{O}(\alpha\gamma^2) & (i \in \mathcal{S}_{\text{irr}} \wedge j \in \mathcal{S}_{\text{rob}}) \\ \mathcal{O}(\beta\gamma^2) & (i \in \mathcal{S}_{\text{irr}} \wedge j \in \mathcal{S}_{\text{vul}}) \end{cases}. \qquad \text{(A207)}$$

(c) For $(j \neq k) \wedge (j \in \mathcal{S}_{\text{irr}} \vee k \in \mathcal{S}_{\text{irr}})$, $\mathbb{E}[x_j x_k x_i^2] = 0$. For $j = k$, using the Cauthy-Schwarz inequality,

$$\mathbb{E}[x_j x_k x_i^2] \leq \sqrt{\mathbb{E}[x_j^4]}\sqrt{\mathbb{E}[x_i^4]} = \begin{cases} \mathcal{O}(\gamma^4) & (j = k) \wedge (i, j, k \in \mathcal{S}_{\text{irr}}) \\ \mathcal{O}(\alpha^2\gamma^2) & (j = k) \wedge (j, k \in d_{\text{irr}} \wedge i \in \mathcal{S}_{\text{rob}}) \\ \mathcal{O}(\beta^2\gamma^2) & (j = k) \wedge (j, k \in d_{\text{irr}} \wedge i \in \mathcal{S}_{\text{vul}}) \end{cases}. \qquad \text{(A208)}$$

Using the Cauthy-Schwarz inequality,

$$\mathbb{E}[x_j x_k x_i^2]$$
$$\leq \sqrt{\mathbb{E}[x_j^2]}\sqrt{\mathbb{E}[x_k^2]}\sqrt{\mathbb{E}[x_i^4]} \qquad\qquad\qquad\qquad\qquad\qquad \text{(A209)}$$

$$= \begin{cases} \mathcal{O}(\alpha^4) & (i, j, k \in \mathcal{S}_{\text{rob}}) \\ \mathcal{O}(\beta^4) & (i, j, k \in \mathcal{S}_{\text{vul}}) \\ \mathcal{O}(\alpha^3\beta) & (i \in \mathcal{S}_{\text{rob}}) \wedge \{(j \in \mathcal{S}_{\text{rob}} \wedge k \in \mathcal{S}_{\text{vul}}) \vee (j \in \mathcal{S}_{\text{vul}} \wedge k \in \mathcal{S}_{\text{rob}})\} \\ \mathcal{O}(\alpha\beta^3) & (i \in \mathcal{S}_{\text{vul}}) \wedge \{(j \in \mathcal{S}_{\text{rob}} \wedge k \in \mathcal{S}_{\text{vul}}) \vee (j \in \mathcal{S}_{\text{vul}} \wedge k \in \mathcal{S}_{\text{rob}})\} \\ \mathcal{O}(\alpha^2\beta^2) & (i \in \mathcal{S}_{\text{rob}} \wedge j, k \in \mathcal{S}_{\text{vul}}) \vee (i \in \mathcal{S}_{\text{vul}} \wedge j, k \in \mathcal{S}_{\text{rob}}) \\ \mathcal{O}(\alpha^2\gamma^2) & (i \in \mathcal{S}_{\text{irr}} \wedge j, k \in \mathcal{S}_{\text{rob}}) \\ \mathcal{O}(\beta^2\gamma^2) & (i \in \mathcal{S}_{\text{irr}} \wedge j, k \in \mathcal{S}_{\text{vul}}) \\ \mathcal{O}(\alpha\beta\gamma^2) & (i \in \mathcal{S}_{\text{irr}}) \wedge \{(j \in \mathcal{S}_{\text{rob}} \wedge k \in \mathcal{S}_{\text{vul}}) \vee (j \in \mathcal{S}_{\text{vul}} \wedge k \in \mathcal{S}_{\text{rob}})\} \end{cases}. \qquad \text{(A210)}$$

$\square$

## G  PROOF OF THEOREM 3.7 (TRADE-OFF)

**Theorem 3.7** (Accuracy–robustness trade-off). Assume $d_{\text{rob}} = 1$, $d_{\text{vul}} = d - 1$, and $d_{\text{irr}} = 0$. In addition to Assumption 3.2, for $(\boldsymbol{x}, y) \sim \mathcal{D}^{\text{te}}$, suppose that $yx_i$ takes $\alpha$ with probability $p > 0.5$ and $-\alpha$ with probability $1 - p$ for $i \in \mathcal{S}_{\text{rob}}$. Moreover, $yx_i$ takes $\beta$ with probability one for $i \in \mathcal{S}_{\text{vul}}$. Define $\tilde{f}(\boldsymbol{P}, \boldsymbol{Q}) := \mathbb{E}_{\{(\boldsymbol{x}_n, y_n)\}_{n=1}^N \overset{\text{i.i.d.}}{\sim} \mathcal{D}^{\text{te}}}[y_{N+1}[f(\boldsymbol{Z_0}; \boldsymbol{P}, \boldsymbol{Q})]_{d+1, N+1}]$. Then, there exist strictly positive functions $g_1(d, \alpha, \beta)$ and $g_2(d, \alpha, \beta)$ such that

$$\tilde{f}(\boldsymbol{P}^{\text{std}}, \boldsymbol{Q}^{\text{std}}) = \begin{cases} g_1(d, \alpha, \beta)(\alpha + (d-1)\beta) & (\text{w.p.} \quad p) \\ g_1(d, \alpha, \beta)(-\alpha + (d-1)\beta) & (\text{w.p.} \quad 1-p) \end{cases}, \tag{10}$$

$$\tilde{f}(\boldsymbol{P}^{\text{adv}}, \boldsymbol{Q}^{\text{adv}}) \leq g_2(d, \alpha, \beta)\{-(2p-1)\alpha^2 + (d-1)\beta^2\} \quad (\text{w.p.} \quad 1-p). \tag{11}$$

*Proof.* Using $\boldsymbol{b}$ and $\boldsymbol{A}$ defined in Appendix E, we can rearrange $\tilde{f}(\boldsymbol{P}, \boldsymbol{Q})$ as

$$\tilde{f}(\boldsymbol{P}, \boldsymbol{Q}) := \mathbb{E}_{\{(\boldsymbol{x}_n, y_n)\}_{n=1}^N}[y_{N+1}[f(\boldsymbol{Z_0}; \boldsymbol{P}, \boldsymbol{Q})]_{d+1, N+1}] \tag{A211}$$

$$= \frac{1}{N}\boldsymbol{b}^\top \mathbb{E}_{\{(\boldsymbol{x}_n, y_n)\}_{n=1}^N}[\boldsymbol{Z_0} \boldsymbol{M} \boldsymbol{Z_0}^\top]\boldsymbol{A} y_{N+1}\boldsymbol{x}_{N+1}. \tag{A212}$$

**Standard Transformer.** Similarly to the proof of Theorem 3.5, using some positive function $g(d, \alpha, \beta) > 0$, we can represent $\mathbb{E}[\boldsymbol{b}^\top \boldsymbol{Z_0} \boldsymbol{M} \boldsymbol{Z_0}^\top \boldsymbol{A}/N] = [g(d, \alpha, \beta) \quad \cdots \quad g(d, \alpha, \beta)]$. Thus,

$$\frac{1}{N}\boldsymbol{b}\mathbb{E}_{\{(\boldsymbol{x}_n, y_n)\}_{n=1}^N}[\boldsymbol{Z_0} \boldsymbol{M} \boldsymbol{Z_0}^\top]\boldsymbol{A} y_{N+1}\boldsymbol{x}_{N+1} = \begin{bmatrix} g(d, \alpha, \beta) \\ \vdots \\ g(d, \alpha, \beta) \end{bmatrix}^\top y_{N+1}\boldsymbol{x}_{N+1} \tag{A213}$$

$$= g(d, \alpha, \beta)y_{N+1}\sum_{i=1}^d x_{N+1, i} \tag{A214}$$

$$= \begin{cases} \alpha + (d-1)\beta & (\text{w.p.} \quad p) \\ -\alpha + (d-1)\beta & (\text{w.p.} \quad 1-p) \end{cases}. \tag{A215}$$

**Adversarially Trained Transformer.** Now,

$$\frac{1}{N}\mathbb{E}_{\{(\boldsymbol{x}_n, y_n)\}_{n=1}^N}[\boldsymbol{Z_0} \boldsymbol{M} \boldsymbol{Z_0}^\top]$$

$$= \begin{bmatrix} \mathbb{E}[(y\boldsymbol{x})(y\boldsymbol{x}^\top)] & \mathbb{E}[y\boldsymbol{x}] \\ \mathbb{E}[y\boldsymbol{x}^\top] & 1 \end{bmatrix} \tag{A216}$$

$$= \begin{bmatrix} \alpha^2 & (2p-1)\alpha\beta & \cdots & (2p-1)\alpha\beta & (2p-1)\alpha \\ (2p-1)\alpha\beta & \beta^2 & \cdots & \beta^2 & \beta \\ (2p-1)\alpha\beta & \beta^2 & \cdots & \beta^2 & \beta \\ \vdots & & & & \\ (2p-1)\alpha\beta & \beta^2 & \cdots & \beta^2 & \beta \\ (2p-1)\alpha & \beta & \cdots & \beta & 1 \end{bmatrix}. \tag{A217}$$

Thus,

$$\frac{1}{N}\boldsymbol{b}^\top \mathbb{E}_{\{(\boldsymbol{x}_n, y_n)\}_{n=1}^N}[\boldsymbol{Z_0} \boldsymbol{M} \boldsymbol{Z_0}^\top]\boldsymbol{A} = \begin{bmatrix} \alpha\{\alpha + (d-1)(2p-1)\beta + (2p-1)\} \\ \beta\{(2p-1)\alpha + (d-1)\beta + 1\} \\ \vdots \\ \beta\{(2p-1)\alpha + (d-1)\beta + 1\} \end{bmatrix}^\top. \tag{A218}$$

Therefore,

$$\frac{1}{N}\boldsymbol{b}^\top \mathbb{E}_{\{(\boldsymbol{x}_n,y_n)\}_{n=1}^N}[\boldsymbol{Z_0 M Z_0^\top}]\boldsymbol{A}y_{N+1}\boldsymbol{x}_{N+1}$$

$$= \begin{bmatrix} \alpha\{\alpha+(d-1)(2p-1)\beta+(2p-1)\} \\ \beta\{(2p-1)\alpha+(d-1)\beta+1\} \\ \vdots \\ \beta\{(2p-1)\alpha+(d-1)\beta+1\} \end{bmatrix}^\top \begin{bmatrix} y_{N+1}x_{N+1,1} \\ \beta \\ \vdots \\ \beta \end{bmatrix} \tag{A219}$$

$$= \begin{cases} \alpha^2\{\alpha+(d-1)(2p-1)\beta+(2p-1)\} \\ \quad +(d-1)\beta^2\{(2p-1)\alpha+(d-1)\beta+1\} & (\text{w.p.}\quad p) \\ -\alpha^2\{\alpha+(d-1)(2p-1)\beta+(2p-1)\} \\ \quad +(d-1)\beta^2\{(2p-1)\alpha+(d-1)\beta+1\} & (\text{w.p.}\quad 1-p) \end{cases}. \tag{A220}$$

In particular,

$$-\alpha^2\{\alpha+(d-1)(2p-1)\beta+(2p-1)\}+(d-1)\beta^2\{(2p-1)\alpha+(d-1)\beta+1\}$$
$$= \{(2p-1)\alpha+(d-1)\beta+1\}(-C\alpha^2+(d-1)\beta^2), \tag{A221}$$

where

$$C = \frac{\alpha+(d-1)(2p-1)\beta+(2p-1)}{(2p-1)\alpha+(d-1)\beta+1} > \frac{(2p-1)^2\alpha+(d-1)(2p-1)\beta+(2p-1)}{(2p-1)\alpha+(d-1)\beta+1} \tag{A222}$$

$$= 2p-1. \tag{A223}$$

$$\square$$

## H  PROOF OF THEOREM H.1 (NEED FOR LARGER SAMPLE SIZE)

**Theorem H.1** (Need for Larger Sample Size). Assume the same assumptions in Theorem 3.7. Then,

$$\mathbb{E}_{\boldsymbol{x}_{N+1},y_{N+1}}[y_{N+1}[f(\boldsymbol{Z_0};\boldsymbol{P}^{\text{std}},\boldsymbol{Q}^{\text{std}})]_{d+1,N+1}] > 0 \qquad (\text{w.p. at least } 1-e^{-pN}). \tag{A224}$$

In addition, suppose that there exists a constant $0 < C < 1$ such that $(d-1)\beta+1 < C\alpha$. Moreover, assume that $N$ is an even number. Then, as $p \to \frac{1}{2}$ with $p > \frac{1}{2}$, for $4 \le N \le \frac{2}{C}$,

$$\mathbb{E}_{\boldsymbol{x}_{N+1},y_{N+1}}[y_{N+1}[f(\boldsymbol{Z_0};\boldsymbol{P}^{\text{adv}},\boldsymbol{Q}^{\text{adv}})]_{d+1,N+1}] > 0$$

$$\left(\text{w.p. at most } 1 - \frac{0.483}{\sqrt{N}} < 1 - e^{-pN}\right). \tag{A225}$$

*Proof.* Using $\boldsymbol{b}$ and $\boldsymbol{A}$ defined in Appendix E, we can calculate

$$\mathbb{E}_{\boldsymbol{x}_{N+1},y_{N+1}}[y_{N+1}[f(\boldsymbol{Z_0};\boldsymbol{P},\boldsymbol{Q})]_{d+1,N+1}] = \frac{1}{N}\boldsymbol{b}^\top \boldsymbol{Z_0 M Z_0^\top}\boldsymbol{A}\mathbb{E}[y_{N+1}\boldsymbol{x}_{N+1}]. \tag{A226}$$

Now,

$$\frac{1}{N}\boldsymbol{Z_0 M Z_0^\top}$$

$$= \begin{bmatrix} \alpha^2 & \frac{\beta}{N}\sum_{n=1}^N y_n x_{n,1} & \cdots & \frac{\beta}{N}\sum_{n=1}^N y_n x_{n,1} & \frac{1}{N}\sum_{n=1}^N y_n x_{n,1} \\ \frac{\beta}{N}\sum_{n=1}^N y_n x_{n,1} & \beta^2 & \cdots & \beta^2 & \beta \\ \frac{\beta}{N}\sum_{n=1}^N y_n x_{n,1} & \beta^2 & \cdots & \beta^2 & \beta \\ \vdots & & & & \\ \frac{\beta}{N}\sum_{n=1}^N y_n x_{n,1} & \beta^2 & \cdots & \beta^2 & \beta \\ \frac{1}{N}\sum_{n=1}^N y_n x_{n,1} & \beta & \cdots & \beta & 1 \end{bmatrix}. \tag{A227}$$

**Standard Transformer.** From the configuration of $\boldsymbol{b}$ and $\boldsymbol{A}$, all the entries of $\boldsymbol{b}^\top \boldsymbol{Z_0 M Z_0^\top}\boldsymbol{A}$ are the same. Since all the entries of $\mathbb{E}[y_{N+1}\boldsymbol{x}_{N+1}]$ are positive, with some positive function $g(d,\alpha,\beta) > 0$,

$$\frac{1}{N}\boldsymbol{b}^\top \boldsymbol{Z_0 M Z_0^\top}\boldsymbol{A}\mathbb{E}[y_{N+1}\boldsymbol{x}_{N+1}] = g(d,\alpha,\beta)\frac{1}{N}\boldsymbol{1}_{d+1}^\top \boldsymbol{Z_0 M Z_0^\top}\boldsymbol{1}_{d+1}. \tag{A228}$$

Now,

$$\frac{1}{N}\mathbf{1}_{d+1}^\top \boldsymbol{Z_0}\boldsymbol{M}\boldsymbol{Z_0}^\top \mathbf{1}_{d+1}$$

$$= (d-1)^2\beta^2 + 2(d-1)\beta + 1 + \alpha^2 + \frac{2}{N}\sum_{n=1}^N y_n x_{n,1} + 2(d-1)\frac{\beta}{N}\sum_{n=1}^N y_n x_{n,1} \qquad \text{(A229)}$$

$$= \{(d-1)\beta + 1\}^2 + \alpha^2 + \frac{2\{(d-1)\beta + 1\}}{N}\sum_{n=1}^N y_n x_{n,1} \qquad \text{(A230)}$$

$$= [\{(d-1)\beta + 1\} - \alpha]^2 + \frac{2\{(d-1)\beta + 1\}}{N}\sum_{n=1}^N (\alpha + y_n x_{n,1}) \qquad \text{(A231)}$$

$$> 0 \qquad (\text{w.p. at least } 1 - (1-p)^N > 1 - e^{-pN}). \qquad \text{(A232)}$$

**Adversarially Trained Transformer.** Note that $\mathbb{E}[y_{N+1}\boldsymbol{x}_{N+1}] = [(2p-1)\alpha \ \beta \ \cdots \ \beta]$. Thus,

$$\frac{1}{N}\mathbf{1}_{d+1}^\top \boldsymbol{Z_0}\boldsymbol{M}\boldsymbol{Z_0}^\top \boldsymbol{I}_d \mathbb{E}[y_{N+1}\boldsymbol{x}_{N+1}]$$

$$= (2p-1)\alpha\left(\alpha^2 + (d-1)\frac{\beta}{N}\sum_{n=1}^N y_n x_{n,1} + \frac{1}{N}\sum_{n=1}^N y_n x_{n,1}\right)$$

$$+ (d-1)\beta\left(\frac{\beta}{N}\sum_{n=1}^N y_n x_{n,1} + (d-1)\beta^2 + \beta\right) \qquad \text{(A233)}$$

$$= [(2p-1)\alpha^3 + (d-1)\beta^2\{(d-1)\beta + 1\}]$$

$$+ [(2p-1)\alpha\{(d-1)\beta + 1\} + (d-1)\beta^2]\frac{1}{N}\sum_{n=1}^N y_n x_{n,1}. \qquad \text{(A234)}$$

This indicates $\mathbb{E}_{\boldsymbol{x}_{N+1},y_{N+1}}[y_{N+1}[f(\boldsymbol{Z_0};\boldsymbol{P}^{\text{adv}},\boldsymbol{Q}^{\text{adv}})]_{d+1,N+1}] > 0$ only if

$$\frac{1}{N}\sum_{n=1}^N y_n x_{n,1} > -\frac{(2p-1)\alpha^3 + (d-1)\beta^2\{(d-1)\beta + 1\}}{(2p-1)\alpha\{(d-1)\beta + 1\} + (d-1)\beta^2}. \qquad \text{(A235)}$$

Representing $y_n x_{n,1} = \alpha(2X_n - 1)$ with $X_n$ taking 1 with probability $p$ and 0 with probability $1 - p$,

$$\frac{1}{N}\sum_{n=1}^N \alpha(2X_n - 1) > -\frac{(2p-1)\alpha^3 + (d-1)\beta^2\{(d-1)\beta + 1\}}{(2p-1)\alpha\{(d-1)\beta + 1\} + (d-1)\beta^2}$$

$$\iff \sum_{n=1}^N X_n > \frac{N}{2}\left(1 - \frac{1}{\alpha}\frac{(2p-1)\alpha^3 + (d-1)\beta^2\{(d-1)\beta + 1\}}{(2p-1)\alpha\{(d-1)\beta + 1\} + (d-1)\beta^2}\right). \qquad \text{(A236)}$$

Let $Y \sim B(N,p)$, where $B(N,p)$ is the Binomial distribution. Consider the following probability:

$$\mathbb{P}_{Y\sim B(N,p)}\left[Y > \frac{N}{2}\left(1 - \frac{1}{\alpha}\frac{(2p-1)\alpha^3 + (d-1)\beta^2\{(d-1)\beta + 1\}}{(2p-1)\alpha\{(d-1)\beta + 1\} + (d-1)\beta^2}\right)\right]. \qquad \text{(A237)}$$

When $p \to 1/2$,

$$\mathbb{P}_{Y\sim B(N,p)}\left[Y > \frac{N}{2}\left(1 - \frac{1}{\alpha}\frac{(2p-1)\alpha^3 + (d-1)\beta^2\{(d-1)\beta + 1\}}{(2p-1)\alpha\{(d-1)\beta + 1\} + (d-1)\beta^2}\right)\right]$$

$$\to \mathbb{P}_{Y\sim B(N,1/2)}\left[Y > \frac{N}{2}\left(1 - \frac{(d-1)\beta + 1}{\alpha}\right)\right] \qquad \text{(A238)}$$

$$\le \mathbb{P}_{Y\sim B(N,1/2)}\left[Y > \frac{N}{2}(1 - C)\right] \qquad \text{(A239)}$$

$$\leq \mathbb{P}_{Y \sim B(N,1/2)}\left[Y > \frac{N}{2} - 1\right]. \tag{A240}$$

From Ash (1990), for an integer $0 < k < N/2$,

$$\mathbb{P}_{Y \sim B(N,1/2)}[Y \leq k] \geq \frac{1}{\sqrt{8N\frac{k}{N}(1 - \frac{k}{N})}} \exp\left(-ND\left(\frac{k}{N} /\!/ \frac{1}{2}\right)\right), \tag{A241}$$

where $D$ is the Kullback–Leibler divergence. Substituting $k = \frac{N}{2} - 1$,

$$\mathbb{P}_{Y \sim B(N,1/2)}\left[Y \leq \frac{N}{2} - 1\right]$$

$$\geq \frac{1}{\sqrt{8N(\frac{1}{2} - \frac{1}{N})\{1 - (\frac{1}{2} - \frac{1}{N})\}}} \exp\left(-ND\left(\frac{1}{2} - \frac{1}{N} /\!/ \frac{1}{2}\right)\right) \tag{A242}$$

$$= \frac{1}{\sqrt{2(1 - \frac{4}{N^2})}} \frac{1}{\sqrt{N}} \exp\left(-ND\left(\frac{1}{2} - \frac{1}{N} /\!/ \frac{1}{2}\right)\right). \tag{A243}$$

Note that

$$D\left(\frac{1}{2} - \frac{1}{N} /\!/ \frac{1}{2}\right) = \frac{1}{2}\left\{\left(1 - \frac{2}{N}\right)\ln\left(1 - \frac{2}{N}\right) + \left(1 + \frac{2}{N}\right)\ln\left(1 + \frac{2}{N}\right)\right\}. \tag{A244}$$

For $N \geq 4$,

$$\frac{1}{\sqrt{2(1 - \frac{4}{N^2})}} \exp\left(-ND\left(\frac{1}{2} - \frac{1}{N} /\!/ \frac{1}{2}\right)\right) > 0.483. \tag{A245}$$

In summary,

$$\mathbb{P}_{Y \sim B(N,1/2)}\left[Y > \frac{N}{2} - 1\right] = 1 - \mathbb{P}_{Y \sim B(N,1/2)}\left[Y \leq \frac{N}{2} - 1\right] \leq 1 - \frac{0.483}{\sqrt{N}}. \tag{A246}$$

$\square$

