# OpenReview forum: "Adversarially Pretrained Transformers May Be Universally Robust In-Context Learners"
_ICLR.cc/2026/Conference — ICLR 2026 Poster_

### Official Review · Reviewer_oCaH · 2025-10-31

**Soundness:** 3
**Presentation:** 4
**Contribution:** 3
**Rating:** 8
**Confidence:** 3

**Summary:**

This paper considers the adversarial robustness of pre-trained transformers using in-context learning. They derive a set of theoretical results for a single-layer "linear" transformer and show that adversarially trained transformers can be universally robust in-context learners. Their results improve the mathematical validity of previously empirically and intuitively understood concepts in adversarial robustness, such as the number of redundant, robust, and non-robust features, the effect of the strength of the adversarial perturbation, and the robustness-accuracy tradeoff. Finally, they briefly validate some of the theorems with simple experiments.

**Strengths:**

- The theoretical results are significant in improving the mathematical understanding of adversarial robustness under norm-bounded perturbations.
- The paper is organized and written well, and while the theory is dense, most of the main text is presented and discussed with clarity and intuition.

**Weaknesses:**

- The experiments are brief, and Table 2 is a somewhat expected result, regardless of the contributions of the paper. However, the brevity of the experiments is made up for by the theoretical results.
- The proofs are far too long and tedious. I understand that proving the theorems in a more succinct manner is non-trivial, but this still makes verifying the proofs rather difficult. Regardless, at a cursory glance, the proofs seem correct and seem to follow sound logic.

**Questions:**

- Lemma 3.3: This lemma seems integral to the rest of the derivations. However, the intuition behind the lemma and its connection to the following theorems are slightly difficult to grasp. Could the authors elaborate on this further and add the additional elaborations to the text?

- Line 199: “Taylor’s law (Taylor, 1961), is observed in a wide range of natural datasets and distributions.” To improve the flow of reading, it would be good to mention that examples of this are provided further below in the text.

- Section 3.2: “Warm-up” may be a misleading term, as it usually pertains to warm-up during training of a neural network. Perhaps it’s better to use another term or simply call the section “linear classifiers and oracle.”

- Section 3.2: Although a reasonable assumption, the authors should mention that $w \in [0,1]^d$ and provide a brief justification.

- Line 288: “non-linearity and non-convexity in the trainable parameters P and Q”: parameters cannot be linear or non-linear, convex or non-convex; it is a property of functions or sets. Please elaborate or correct this notation.

- Line 289: “High non-linearity” is a vague term; perhaps it is better to bundle all mentioned non-linearities as simply “non-linearity” as a boolean attribute.

- Thrm 3.4: Shouldn’t the second case be “$0<\epsilon \leq$” instead of “$\epsilon =$”? Additionally, there is a discontinuity between the thresholds in cases 2 & 3. Could the authors elaborate on why this is and what happens in the left-out values of $\epsilon$?

While I currently suggested "Accept", should these minor considerations be addressed, I believe the paper is worthy of a "Strong accept" score.

---

> ### Author Response · Authors · 2025-11-20
>
> We sincerely appreciate the reviewer's careful reading of our paper. We acknowledge the weaknesses pointed out by the reviewer and address the concerns as follows:
>
> ---
>
> > Lemma 3.3: Could the authors elaborate on this further and add the additional elaborations to the text?
>
> We add the following proof sketch:
>
> Let us simplify the transformer definition (2) $f(Z_\Delta;P,Q) := (1/N) P Z_\Delta M Z^\top_\Delta Q Z_\Delta$ as $f(x;\theta_1,\theta_2) := \theta_1 x^2 \theta_2 (x+\Delta)$, where $\theta_1, \theta_2 \in \\{0, 1\\}$. Then the optimization problem (7) becomes:
>
> $$
> \min_{\theta_1, \theta_2 \in \\{0, 1\\}} \max_{\Delta} - y f(x + \Delta; \theta_1, \theta_2) = \min_{\theta_1, \theta_2 \in \\{0, 1\\}} \max_{\Delta} - y ( \theta_1 x^2 \theta_2 (x+\Delta)).
> $$
>
> This can be transformed to:
>
> $$
> \min_{\theta_1, \theta_2 \in \\{0, 1\\}} \max_{\Delta} - y ( \theta_1 x^2 \theta_2 (x+\Delta))
> = \max_{\theta_1, \theta_2 \in \\{0, 1\\}} \min_{\Delta} \theta_1 (y x^2 \theta_2 (x+\Delta)).
> $$
>
> Since $\theta_1$ takes only 0 or 1, the optimal strategy is always $\theta_1 = 1$ when $y x^2 \theta_2 (x+\Delta)$ is positive and $\theta_1 = 0$ when negative. This transforms the problem to:
>
> $$
> \max_{\theta_1, \theta_2 \in \\{0, 1\\}} \min_{\Delta} \theta_1 (y x^2 \theta_2 (x+\Delta))
> = \max_{\theta_2 \in \\{0, 1\\}} \max( 0, \min_{\Delta} y x^2 \theta_2 (x+\Delta)).
> $$
>
> Denoting all terms except $\theta_2$ as $h$:
>
> $$
> \max_{\theta_2 \in \\{0, 1\\}} \max( 0, \min_{\Delta} y x^2 \theta_2 (x+\Delta))
> = \max_{\theta_2 \in \\{0, 1\\}} \max( 0, \theta_2 h).
> $$
>
> This provides intuition for the problem $\max\_{b \in \\{0, 1\\}} \sum^{d(d+1)}\_{i=1} \max(0, \sum^{d+1}\_{j=1} b_j h_{i,j})$ in Lemma 3.3.
>
> > Line 199: “Taylor’s law (Taylor, 1961), is observed in a wide range of natural datasets and distributions.” To improve the flow of reading, it would be good to mention that examples of this are provided further below in the text.
>
> We will add the following examples: In images after typical preprocessing, pixel intensities and contrast exhibit exponential probability distribution tails [1, 2], and exponential distributions satisfy Taylor's law. Additionally, across diverse text datasets, word frequency follows this law [3].
>
> [1] D. L Ruderman et al. The statistics of natural images. Network: Computation in Neural Systems. 1994.
> [2] A. Srivastava et al. On advances in statistical modeling of natural images. Journal of Mathematical Imaging and Vision. 2003.
> [3] M. Gerlach et al. Scaling laws and fluctuations in the statistics of word frequencies. New Journal of Physics. 2014.
>
> > Section 3.2: “Warm-up” may be a misleading term, as it usually pertains to warm-up during training of a neural network. Perhaps it’s better to use another term or simply call the section “linear classifiers and oracle.”
>
> We will rename this following the reviewer's suggestion.
>
> > Section 3.2: Although a reasonable assumption, the authors should mention that $w \in [0, 1]^d$ and provide a brief justification.
>
> We will add the following explanation: We impose the constraint on $w$ to prevent the problem from becoming ill-posed. We chose $[0, 1]^d$ rather than $[-1, 1]^d$ to simplify the theoretical derivation; using $[-1, 1]^d$ would merely increase the number of cases to consider without providing additional insights.
>
> > Line 288: “non-linearity and non-convexity in the trainable parameters P and Q”: parameters cannot be linear or non-linear, convex or non-convex; it is a property of functions or sets. Please elaborate or correct this notation.
>
> We will revise to: "the non-linearity and non-convexity of the model with respect to the trainable parameters P and Q."
>
> > Line 289: “High non-linearity” is a vague term; perhaps it is better to bundle all mentioned non-linearities as simply “non-linearity” as a boolean attribute.
>
> We will remove "High."
>
> > Thrm 3.4: Shouldn’t the second case be "$0 < \epsilon \leq$" instead of $\epsilon =$? Additionally, there is a discontinuity between the thresholds in cases 2 & 3. Could the authors elaborate on why this is and what happens in the left-out values of $\epsilon$?
>
> The dynamics of the trained paramters with respect to $\epsilon$ can be found in Theorem D.1. In summary,
>
> - If $0 \leq \epsilon \leq \frac{\lambda(\lambda(d-2)+4)}{2(\lambda(d-1)+2)}$, training converges to the standard regime.
> - If $\epsilon = \frac{1+(d-1)(\lambda/2)}{d}$, it converges to the adversarial regime.
> - If $\epsilon \geq \frac{\lambda}{2} + \frac{3}{2} \frac{2-\lambda}{(d-1)\lambda^2 + 3}$, it converges to the strongly adversarial regime.
>
> As the reviewer pointed out, there are discontinuities in $\epsilon$. This is for computational tractability. For unspecified $\epsilon$, adversarial pretraining (i.e., the minimization problem (7)) becomes nearly intractable.

---

> ### Comment · Reviewer_oCaH · 2025-11-25
>
> I would like to thank the authors for the clarifications. Most of my concerns have been addressed, and I do sincerely hope the authors incorporate these changes and clarifications into the manuscript.
>
> With that said, currently, other reviewers have not raised any major concerns in my eyes. I believe this paper makes a reasonable theoretical contribution to adversarial robustness and that more sound theoretical results should be encouraged within the community. Thus, I will be happy to raise my score, should no significant developments occur.

---

### Official Review · Reviewer_rGBV · 2025-11-01

**Soundness:** 2
**Presentation:** 3
**Contribution:** 2
**Rating:** 6
**Confidence:** 3

**Summary:**

In this paper, the authors theoretically demonstrate that single-layer linear transformers, after adversarial pre-training across multiple classification tasks, can robustly adapt to previously unseen classification tasks via in-context learning without requiring any additional training. The authors also conduct small-scale empirical experiments to provide preliminary evidence supporting their theoretical claims.

**Strengths:**

- The paper is clearly written and easy to follow.
- The topic is promising. If robustness can indeed be efficiently achieved during pre-training and transferred to downstream tasks, it would be highly meaningful.
- The paper seems to provide solid and convincing theoretical analysis.
- The authors also offer empirical results that provide a certain level of support for the proposed approach.

**Weaknesses:**

- The paper is limited to single-layer linear transformers, and it remains unclear whether the theoretical results can generalize to more realistic multi-layer non-linear transformers, which form the basis of modern foundation models.
- The empirical evaluation is also restricted to single-layer linear transformers. I suggest conducting experiments on multi-layer non-linear transformers and on more complex datasets to better demonstrate the generalizability of the conclusions (not necessarily large-scale, but at least small-scale experiments on nonlinear multi-layer transformers would strengthen the claim.)
- It would be helpful if the authors could discuss the practicality of achieving robustness during pre-training and transferring it to downstream tasks, compared to performing robust fine-tuning at the adaptation stage (for example [1]).

[1] AutoLoRa: An Automated Robust Fine-Tuning Framework. ICLR 24

**Questions:**

Please see the weaknesses.

---

> ### Author Response · Authors · 2025-11-18
>
> We sincerely appreciate the reviewer's constructive feedback.
>
> **Summary.** We emphasize that our theoretical assumptions are comparable in strength to those employed in existing studies. Regarding the limitations of our experiments, we respectfully ask the reviewer to reconsider that our experimental results are sufficient to validate the theoretical findings in this paper, and that our primary objective is to provide a theoretical foundation for universally robust foundation models.
>
> ---
>
> > The paper is limited to single-layer linear transformers, and it remains unclear whether the theoretical results can generalize to more realistic multi-layer non-linear transformers, which form the basis of modern foundation models.
>
> We acknowledge that the model simplicity is a limitation of our work. However, we would like to emphasize that single-layer linear transformers are the standard choice in theoretical research on in-context learning, even for simpler settings without adversaries (cf. studies listed in Appendix A). Theoretical analysis of in-context learning is still an emerging field, and the mathematical frameworks to analyze multi-layer non-linear transformers remain largely undeveloped. Given this context, we consider that our theoretical contributions on single-layer linear models represent meaningful progress.
>
> > The empirical evaluation is also restricted to single-layer linear transformers. I suggest conducting experiments on multi-layer non-linear transformers and on more complex datasets to better demonstrate the generalizability of the conclusions (not necessarily large-scale, but at least small-scale experiments on nonlinear multi-layer transformers would strengthen the claim.)
>
> We acknowledge the limited experimental evaluation. However, we would like to emphasize that our main contribution is theoretical. In theory-oriented work, it is not always necessary to empirically validate every possible scenario or potential problem instance. Instead, experiments are meant to verify theoretical results. In this respect, the experiments we conducted are appropriate for verifying our theory. We therefore kindly ask the reviewer to reconsider our work in terms of its intended scope and theoretical significance. Our results provide the first rigorous theoretical foundation for universally robust foundation models, offering an important advancement in this area.
>
> Besides, it should be noted that extending experiments to (large) multi-layer non-linear transformers presents difficulties in constructing appropriate pretraining datasets. Our analysis demonstrates that the pretraining distribution (Assumption 3.1) enables single-layer linear transformers to adapt to various test distributions (Assumption 3.2, MNIST, Fashion-MNIST, and CIFAR-10). However, appropriate pretraining datasets for multi-layer non-linear transformers remains unclear.
>
> Training large transformers (like LLMs) illustrates this challenge well. Intuitively, training them on our pretraining distribution (Assumption 3.1) would likely lead to overfitting, preventing the development of their in-context learning ability. However, this does not prove that such transformers lack in-context learning capability. Assessing the applicability of in-context learning requires an appropriate pretraining dataset, but how to design such a dataset for multi-layer non-linear transformers is not obvious, aside from using very large-scale training datasets.
>
> Nevertheless, if this concern significantly impacts the reviewer's assessment, we would welcome specific suggestions for experimental settings (datasets and network architectures, particularly, the number of layers) that would most effectively address the reviewer's concerns. We are committed to pursuing additional experiments based on the reviewer's guidance.
>
> > It would be helpful if the authors could discuss the practicality of achieving robustness during pre-training and transferring it to downstream tasks, compared to performing robust fine-tuning at the adaptation stage (for example [1]).
> >
> > [1] AutoLoRa: An Automated Robust Fine-Tuning Framework. ICLR 24
>
> The comparison between ours (adversarial pretraining + in-context learning) and adversarial pretraining + standard finetuning can be found in Section B.4. Our analysis shows that unlike in-context learning, standard finetuning faces a challenge: the selection of trainable parameters. When this selection fails, finetuned models lose the robustness gained during pretraining.
>
> Similarly, for standard pretraining followed by adversarial finetuning, we can demonstrate that this approach fails to achieve robustness when parameter selection is incorrect. This highlights a key advantage of in-context learning-based approach: it sidesteps the parameter selection problem, maintaining robust performance.

---

### Official Review · Reviewer_2dSm · 2025-11-04

**Soundness:** 3
**Presentation:** 4
**Contribution:** 3
**Rating:** 4
**Confidence:** 4

**Summary:**

The authors demonstrate that adversarially pretraining on a mix of classification tasks can make a (single-layer, linear) transformer act as a “universally robust” in-context learner: once pretrained, it can adapt to new classification tasks from clean demos and stay robust to attacks on the query, without any downstream adversarial training. The theory says robustness comes from the model learning to attend to task-shared robust features.

**Strengths:**

1. the paper is in general well written with clear notation and explanations
2. The author provides a formal analysis that proves task-transferable robustness under a matched threat model.
3. The author provides a good experiental setup with clear sample creation scheme.

**Weaknesses:**

1. The proofs are for single-layer linear transformers with matched adversaries; the paper text, as described, markets this as universally robust foundation models. For it to be an universally robust model, I would expect an transferability for different perturbation class and distribution shift setup. The whole story depends on the downstream adversary matching (or being close to) the one used in pretraining.

2. Another key theoretical baseline missing is that the author did not compare the no-pretraining case. Can the author demonstrate an separation or difference in sample complexity between adversarial training + ICL over direct linear regression on the test distribution. And is there an way to provide both an upper and lower bound on the error. E.g. LR > Pretrain + ICL> Adv + ICL > Robust LR ?

3. The explanation for the connection between theorem 3.5 and 3.6 is confusing. I do not observe an strong separation and see why adversarial pretraining leads to an strictly smaller robust error.

4. The design of the training dataset is quite arbitrary and to me it directly encodes the robust feature and thus it is able provide generalize to the test dataset which contains more weaker signals. To demonstrate the universality, I think the author should compare the training and test dataset with general distributions.

In addition, the theory already says transfer happens when the new task uses the same robust features as the pretraining mix. That’s a strong precondition. The author should discuss the scenario where the training data covers an diverse set of robustness features. Otherwise, this distribution coverage assumption is too strong. Similar thing should be analyzed in the experient as well.

5. As the author mentions, theorem 3.7 requires small N regime, the author should consider include an sample complexity bound that can be directly compared against standard ICL case as in [1].


[1] Ruiqi Zhang, Spencer Frei, and Peter L. Bartlett. Trained transformers learn linear models in-context. JMLR, 25(49):1–55, 2024b.

**Questions:**

1. The author assumes clean demonstrations for the new task. How sensitive is the in-context robustness to label noise in the demonstrations?
2. How much of the benefit comes from task diversity in adversarial pretraining vs. adversarial pretraining itself? In other words, if I adversarially pretrain on fewer tasks, does the cross-task robust ICL degrade smoothly or abruptly?
3. It is known that non-robust feature can also transfer. Can the author provide an in-depth analysis of the learned transformer weight to clarify which component are robustly learned parameters and which is not.

---

> ### Author Response · Authors · 2025-11-18
> **Official Comment by Authors (1/3)**
>
> We sincerely appreciate the reviewer's thoughtful comments.
>
> **Summary.** Regarding the strength of our data assumptions, we consider they provide a reasonable approximation of natural objects and are comparable in strength to those commonly made in the existing theoretical literature. For other concerns, based on the reviewer's comments, we will add several clarifications and explanations to avoid potential misunderstandings.
>
> ---
>
> > 1. The proofs are for single-layer linear transformers with matched adversaries; the paper text, as described, markets this as universally robust foundation models. For it to be an universally robust model, I would expect an transferability for different perturbation class and distribution shift setup. The whole story depends on the downstream adversary matching (or being close to) the one used in pretraining.
>
> We define "universal robustness" as the capability to handle data distributions not encountered during training (e.g., lines 14 and 65). Given that there is still no unified interpretation of "universal robustness" in the field (to the best of our knowledge), we consider this usage is acceptable. However, to avoid reader misunderstanding, we will explicitly mention this in the limitation section (i.e., Section 5).
>
> > 2. Another key theoretical baseline missing is that the author did not compare the no-pretraining case. Can the author demonstrate an separation or difference in sample complexity between adversarial training + ICL over direct linear regression on the test distribution. And is there an way to provide both an upper and lower bound on the error. E.g. LR > Pretrain + ICL> Adv + ICL > Robust LR ?
>
> We will add the following clarification to the Appendix.
>
> **Adversarial training + ICL**
>
> First, we should clarify that a model trained on a single task, unlike one pretrained on multiple tasks (i.e., ours), lacks in-context learning capability. That is, an approach like "adversarial training (not adversarial pretraining) + ICL" is not feasible.
>
> Specifically, the parameters of a model trained on a single task differ significantly from those on multiple tasks (shown in Theorem 3.4). Such models cannot provide correct answers for new tasks via ICL even in standard settings (without adversaries). The interesting property of transformers is that when trained on multiple tasks rather than a single one—as with large language models—they develop distinctly different parameters that enable in-context learning capability.
>
> **Task-specific adversarial training vs ICL**
>
> The reviewer's another concern might relate to performance comparisons between task-specific models and those using ICL (our approach). We did not include this comparison because the former does not offer *universal* robustness that is the focus of our work.
>
> As the no-free-lunch theorem indicates, task-specific approaches achieve higher performance. Specifically, for robust accuracy:
>
> (1) Task-specific adversarially trained model $\geq$ (2) Adversarially pretrained model with ICL $\gg$ (3) Task-specific normally trained model $\approx$ (4) Normally pretrained model with ICL $\approx$ 0%
>
> More precisely, model (2) has the limitation of robustness, $d_\mathrm{vul} \leq (\alpha / \beta)^2 d_\mathrm{rob}$ (line 388). However, if training and test distributions match, model (1) does not. The sample complexity follows (1) $\leq$ (2) as well.
>
> However, we emphasize that approach (1) requires users to perform adversarial training for each individual task and cannot generalize to test distributions other than the one it was trained on. This makes it unsuitable for our research focus on universally robust foundation models that can generalize across a wide range of tasks without task-specific adversarial training. We will clarify this point in the Appendix.
>
> > 3. The explanation for the connection between theorem 3.5 and 3.6 is confusing. I do not observe an strong separation and see why adversarial pretraining leads to an strictly smaller robust error.
>
> The difference between these theorems stems from the difference in model parameters. Theorem 3.5 uses $P^\mathrm{std}$ and $Q^\mathrm{std}$, while Theorem 3.6 uses $P^\mathrm{adv}$ and $Q^\mathrm{adv}$. These symbols are defined in Theorem 3.4.
>
> In our setting, single-layer linear transformers develop parameters like $P^\mathrm{std}$ and $Q^\mathrm{std}$ when pretrained without adversaries (i.e., $\epsilon = 0$), and develop $P^\mathrm{adv}$ and $Q^\mathrm{adv}$ when pretrained with sufficiently strong adversaries (e.g., $\epsilon = \frac{1+(d-1)(\lambda/2)}{d}$ in our case). This parameter difference leads to the distinct ICL behaviors of both models.

---

> ### Author Response · Authors · 2025-11-18
> **Official Comment by Authors (2/3)**
>
> > 4. The design of ...
>
> (First, note that limited *training* datasets are not problematic but rather beneficial. The value of universal robustness lies in its ability to generalize from limited training distributions to a wide range of test distributions. Indeed, achieving effective pretraining with smaller training datasets would be desirable.)
>
> Our training and test dataset design is not arbitrary. It is grounded in well-established empirical observations in this area (i.e., the presence of robust and non-robust features) and aligns with the theoretical framework used in Tsipras et al. (2019), providing a solid theoretical basis for studying universally robust foundation models. We suspect that the reviewer’s concern for data assumptions may stem from the following questions.
>
> 1. How different are the training and test distributions?
> 2. Is adaptation achieved simply because training and test distributions share the same robust features?
> 3. Is the test distribution truly realistic?
>
> **Regarding (1):** The class of the test distribution is strictly broader than that of the training distribution (cf. line 208). In particular, the test distribution is based on weaker (more general) distributional assumptions and independence properties of the features, and therefore cannot be regarded merely as a mixture of the training distributions. The training distribution considers only uniform distributions, while the test distribution does not require a specific distributional form unless Assumption 3.2(2) holds. The training distribution assumes that all features are independent, while the test distribution does not. Unlike the training distribution, the test distribution allows for much richer patterns of robust, non-robust, and irrelevant features.
>
> **Regarding (2):** The answer to this question depends on how we define "the same robust features." If we mean identical occurrence patterns (e.g., robust features appearing only in the first dimension), then the answer is No. The test distributions include many patterns not present in the training distributions. However, if we mean that patterns in the test distribution can be composed from combinations of training distribution patterns—setting aside the differences discussed in (1)—then the answer is Yes. Neverthelss, we emphasize that this setup is reasonable. Natural datasets share the same *low-level* robust features.
>
> In general, robust features refer to the set of high-magnitude dimensions that cannot be destroyed by attacks, such as shapes and edges. In natural datasets, while object shapes (high-level robust features) differ across datasets, edges composing those shapes (low-level robust features) are shared. For example, cat ears, dog mouths, and human eyes have different shapes, but they all share common vertical, horizontal, and diagonal edges. We consider that a wide rage of natural objects shares "the same robust features."
>
> This assumption is supported in [1]. They demonstrated that providing only low-level information like edges during pretraining enables neural networks to achieve high adaptability to real-world datasets.
>
> [1] H. Kataoka et al. Pre-training without natural images. ACCV20.
>
> In our case, the training distribution only demonstrates that each dimension can potentially be a low-level robust feature, while the test distribution represents meaningful patterns of these features, i.e., high-level robust features. We expect the model to connect low-level robust features (learned in pretraining) to task-specific high-level robust features (via ICL in inference).
>
> Our experimental results also suuport this view. As shown in Section 4, single-layer linear transformers adversarially pretrained on our training distributions successfully adapt to MNIST, Fashion-MNIST, and CIFAR-10 although these four datasets do not share *high-level* features. This result suggests that our training distribution offers *low-level* features that are shared in them.
>
> **Regarding (3):** We consider that our test distribution captures the essential characteristics of real-world datasets—features can be categorized as robust or non-robust—as empirically observed in the literature, even though it does not fully model all aspects of such datasets (which is inherently difficult in theory). Please also refer to lines 187-232 for the difference between our assumptions and real-world datasets. Particularlly, the listed examples show diverse applicapability of our test distributions.
>
> Furthermore, it should be noted that theoretical ICL research commonly employs easy dataset assumptions (e.g., linear datasets), even for simpler problems without adversaries (cf. the research listed in Appendix A). We consider that our theoretical assumptions are comparable in strength to those in existing literature. Our work provides the foundational theoretical framework for this emerging area.

---

> ### Author Response · Authors · 2025-11-18
> **Official Comment by Authors (3/3)**
>
> > 5. As the author mentions, theorem 3.7 requires small N regime, the author should consider include an sample complexity bound that can be directly compared against standard ICL case as in [1].
> >
> >    [1] Ruiqi Zhang, Spencer Frei, and Peter L. Bartlett. Trained transformers learn linear models in-context. JMLR, 25(49):1–55, 2024b.
>
> Direct comparison is challenging due to numerous differences in problem settings, including data distributions, tasks (they consider regression while we consider classification), and error metrics. If the reviewer could specify particular problem settings and theorems from [1] for the comparision, we would be happy to provide detailed analysis.
>
> > 1. The author assumes clean demonstrations for the new task. How sensitive is the in-context robustness to label noise in the demonstrations?
>
> For random label noise, with sufficiently large N, the impact is limited. Similar to standard training without ICL, the effect depends on the probability of label noise and sample size. Roughly, we conjecture that the impact would be comparable to that in noisy linear classification.
>
> > 2. How much of the benefit comes from task diversity in adversarial pretraining vs. adversarial pretraining itself? In other words, if I adversarially pretrain on fewer tasks, does the cross-task robust ICL degrade smoothly or abruptly?
>
> In our setting, ICL capability degrades smoothly as the number of training tasks decreases.
>
> > 3. It is known that non-robust feature can also transfer. Can the author provide an in-depth analysis of the learned transformer weight to clarify which component are robustly learned parameters and which is not.
>
> The parameter $Q^\mathrm{adv}$ from adversarial pretraining is sparser than $Q^\mathrm{std}$ from standard pretraining. Interestingly, this sparsity in parameters from adversarial training aligns with previous results in neural networks (e.g., [2]). Parameter sparsity might be related to robustness.
>
> [2] P. Chalasani, et al. Concise Explanations of Neural Networks using Adversarial Training. ICML20.

---

### Official Review · Reviewer_SXX6 · 2025-11-08

**Soundness:** 3
**Presentation:** 3
**Contribution:** 2
**Rating:** 6
**Confidence:** 3

**Summary:**

The paper provides a theoretical investigation of the robustness of single-layer linear transformers to adversarial perturbations to query tokens when performing in-context classification.

**Strengths:**

- Generally well written paper.
- The core insight of the paper (theorem 3.6) -- that universally robust linear transformer (for very particular class of classification tasks) are realizable is moderately surprising and interesting imo.
- Other main results of the paper (accuracy-robustness tradeoff, need for larger in-context datasets) are well presented though not that surprising.
- The theoretical results are interpretable and provide some intuition about accuracy-robustness trade off.
- The paper empirically validates its theoretical results.
- Limitations of the paper are well-acknowledged.

**Weaknesses:**

Most of the weaknesses of the paper relate to the (narrow) assumptions authors make to make the theoretical results tractable. While typical of theory papers, these are nevertheless weaknesses as they limit the relevance of these results to practical contexts.
- The paper only studies single-layer linear transformers. This is a major weakness of the paper; as all the results pertain to this narrow class of models, I am not certain whether there is something special about this class of models, or can we expect some variation of these results to hold for other types of transformers as well. Given the relative difficulty of theoretical investigation for other types of tansformers, including some sort of empirical investigatons of other transformer classes would be helpful.
- The paper only considers a single type of perturbation (L-infty norm).
- The setup used in the paper is somewhat artificial.
- Theorem G.1 is not empirically validated (I guess this is because these transformers don't length generalize?)

Minor:
- In lines 393-394, I was initially confused by the current phrasing and would suggest rephrasing to "For example, when α = 160/255 and β = 8/255, the standard model becomes vulnerable at dvul ≳ 20drob, whereas the adversarially pretrained model **becomes vulnerable at dvul ≳ 400drob**".

**Questions:**

- Do authors have any comments or thoughts about whether universally robust multi-layer linear transformers, or softmax transformers, are realizable or not? I would in particular appreciate any insights that authors may have on the interplay between robustness and depth of the transformer.
-

---

> ### Author Response · Authors · 2025-11-18
>
> We sincerely appreciate the reviewer's thoughtful comments.
>
> **Summary.** We emphasize that our theoretical assumptions are comparable in strength to those employed in existing studies. Regarding the limitations of our experiments, we respectfully ask the reviewer to reconsider that our experimental results are sufficient to validate the theoretical findings in this paper, and that our primary objective is to provide a theoretical foundation for universally robust foundation models.
>
> ---
>
> > Most of the weaknesses of the paper relate to the (narrow) assumptions ...
>
> We acknowledge the weaknesses identified by the reviewer:
>
> - Architecture: single-layer linear transformers
> - Perturbation type: L-inf norm
> - Data distributions: Assumptions 3.1 and 3.2
>
> However, we emphasize that these assumptions are comparable in strength to those commonly made in the theoretical literature. For example, current in-context learning research is predominantly confined to single-layer linear transformers (cf. studies listed in Appendix A). Similarly, most in-context learning literature restricts analysis to simple datasets such as linear data. Given that our problem setting is considerably more complex due to the presence of adversaries, we consider that Assumptions 3.1 and 3.2 are reasonable and do not significantly diminish our contributions.
>
> > including some sort of empirical investigatons of other transformer classes would be helpful.
>
> We acknowledge the limited experimental evaluation. However, we would like to emphasize that our main contribution is theoretical. In theory-oriented work, it is not always necessary to empirically validate every possible scenario or potential problem instance. Instead, experiments are meant to verify theoretical results. In this respect, the experiments we conducted are appropriate for verifying our theory. We therefore kindly ask the reviewer to reconsider our work in terms of its intended scope and theoretical significance. Our results provide the first rigorous theoretical foundation for universally robust foundation models, offering an important advancement in this area.
>
> Besides, it should be noted that extending experiments to (large) multi-layer non-linear transformers presents difficulties in constructing appropriate pretraining datasets. Our analysis demonstrates that the pretraining distribution (Assumption 3.1) enables single-layer linear transformers to adapt to various test distributions (Assumption 3.2, MNIST, Fashion-MNIST, and CIFAR-10). However, appropriate pretraining datasets for multi-layer non-linear transformers remains unclear.
>
> Training large transformers (like LLMs) illustrates this challenge well. Intuitively, training them on our pretraining distribution (Assumption 3.1) would likely lead to overfitting, preventing the development of their in-context learning ability. However, this does not prove that such transformers lack in-context learning capability. Assessing the applicability of in-context learning requires an appropriate pretraining dataset, but how to design such a dataset for multi-layer non-linear transformers is not obvious, aside from using very large-scale training datasets.
>
> Nevertheless, if this concern significantly impacts the reviewer's assessment, we would welcome specific suggestions for experimental settings (datasets and network architectures, particularly, the number of layers) that would most effectively address the reviewer's concerns. We are committed to pursuing additional experiments based on the reviewer's guidance.
>
> > Theorem G.1 is not empirically validated
>
> The experimental validation for Theorem G.1 can be found in Appendix C.3 (Figure A6 and line 1121). In summary, standard single-layer linear transformers maintain high classification accuracy in small demonstration regimes, whereas adversarially trained models show degraded performance.
>
> > In lines 393-394, I was initially confused by the current phrasing ...
>
> We agree that the original phrasing is potentially confusing. We will adopt the reviewer's suggested revision or at least rephrase the representation.
>
> > Do authors have any comments or thoughts about whether universally robust multi-layer linear transformers, or softmax transformers, are realizable or not?
>
> We consider that multi-layer non-linear transformers can serve as universally robust foundation models. The increased expressiveness from single-layer to multi-layer architectures may overcome the robustness limitations of single-layer linear transformers, such as $d_\mathrm{vul} \leq (\alpha/\beta)^2 d_\mathrm{rob}$ (cf. line 388). Whether the robustness requires the non-linearity of attention is an open question. While softmax attention dominates in non-adversarial settings, theoretical and empirical evidence suggesting that adversarial robustness favors model linearity (e.g., [1]) leaves room for potential advantages of linear attention.
>
> [1] C. Qin et al. Adversarial Robustness through Local Linearization. NeurIPS19.

---

> > ### Comment · Reviewer_SXX6 · 2025-11-20
> >
> > Thanks for the response. I don't think additional experiments are necessary. However, as Reviewer 2dSm pointed out, the claims about universal robustness are a bit overstated. I would suggest making the tile more precise (the abstract is fine).

---

> > > ### Author Response · Authors · 2025-11-21
> > >
> > > We sincerely appreciate the reviewer’s response.
> > >
> > > We are open to adopting a more appropriate term in place of "universally robust" if one emerges through discussion with the reviewers. If Reviewer SXX6 has any suggestions, we would be happy to consider them (though there is no obligation to respond to this message).
> > >
> > > We would like to note that we consider that "universally robust" is a reasonable description of the model's behavior: despite being trained on a limited distribution, it generalizes robustly across a broad range of distributions. Although we acknowledge that our study examines specific architectures and l-inf perturbations, we consider that the phrase "may be" in the title helps avoid overstating our claims. Nevertheless, we remain fully open to revising the terminology based on the reviewer's guidance.

---

### Author Response · Authors · 2025-12-02
**To the Newly Assigned AC: Summary of Our Study and Discussion**

We extend our deepest appreciation to the newly assigned AC for their efforts on behalf of the community. We are also grateful to the ICLR PCs and committee for seeking the best resolution to this matter. Finally, we sincerely thank the four reviewers who engaged thoughtfully with our paper.

This message summarizes our paper's contributions and the discussions with the reviewers. The score changes and response status are as follows:

- **Reviewer oCaH**: 8 → 10 (noted: "I will be happy to raise my score, should no significant developments occur")
- **Reviewer SXX6**: 6 → 6 (score maintained)
- **Reviewer rGBV**: 6 (no response)
- **Reviewer 2dSm**: 4 (no response)

Note that the responses from Reviewers oCaH and SXX6 were received clearly before this incident became widely known (11/28).

**Contribution**: To address the high computational cost of adversarial training, we propose the novel concept of universally robust foundation models—models that can robustly adapt to diverse downstream tasks with only lightweight tuning. Specifically, we present the first theoretical analysis demonstrating that single-layer linear transformers, after adversarial pretraining across various classification tasks, can robustly generalize to unseen classification tasks through in-context learning from clean demonstrations (i.e., without requiring any additional adversarial training or examples). Beyond this main results (Theorems 3.5 and 3.6), we provide in-depth analysis of various properties of these trained transformers, including learned parameters (Theorem 3.4), attention to robust features (Theorems 3.5 and 3.6), and robustness trade-offs (Theorems 3.7 and G.1). This study initiates discussion on the utility of universally robust foundation models. While their pretraining is expensive, the investment proves worthwhile as downstream tasks can enjoy free adversarial robustness.

**Discussion**: Regardless of scores, all reviewers praised our paper's concept and theoretical contributions. Specifically, they highlighted as a common strength our proposal of universally robust foundation models and the clearly presented theoretical results supporting such approaches. The main weaknesses and our responses are as follows:

*Architectural Assumption (SXX6 and rGBV)*: As acknowledged in Section 5, our theoretical results are constrained to single-layer linear transformers, which reviewers noted differs from commonly used multi-layer non-linear transformers. While these criticisms are valid, we emphasize that single-layer linear transformers are the standard choice in theoretical research on in-context learning, even for simpler settings without adversaries. Given that mathematical frameworks for analyzing multi-layer non-linear transformers are largely undeveloped, our contributions on single-layer linear models represent meaningful progress.

*Data Assumption (2dSm and potentially SXX6)*: As acknowledged in Section 5, our theoretical results are limited to data distributions satisfying Assumptions 3.1 and 3.2. Reviewers raised concerns about how well these reflect real-world data characteristics. While our assumptions do not fully model all aspects of real-world datasets (inherently difficult in theory), we emphasize that our dataset design is grounded in well-established empirical observations (the presence of robust and non-robust features) and aligns with the theoretical framework of Tsipras et al. (2019). Furthermore, theoretical research on in-context learning commonly employs simplified dataset assumptions (e.g., linear datasets), even for problems without adversaries. We consider our theoretical assumptions comparable in strength to existing literature.

*Limited Experiments (oCaH, SXX6, rGBV, and potentially 2dSm)*: Reviewers noted that our experiments do not include large-scale training and testing. However, we emphasize that the main contribution of our paper is theoretical. In theory-oriented work, we argue that it is not always necessary to empirically validate every possible scenario or potential problem instance. Instead, experiments serve to verify the theoretical results. In this respect, we would like to stress that we conducted comprehensive experiments to verify our theoretical findings. We therefore kindly ask the AC and PCs to evaluate our work within its intended scope and theoretical significance. We also note that experimental extension to multi-layer transformers is non-trivial (please refer to our responses to SXX6 and rGBV).

**Summary**: Universally robust foundation models offer a promising approach that enables anyone, regardless of financial capacity, to easily obtain robust models. We provide the first theoretical support for this concept, establishing important guidance for future theoretical and empirical research. While our paper shares certain limitations common to many theoretical studies, we believe our theoretical results are highly valuable to the community.

---

### Meta-Review · Area_Chair_H3wr · 2026-01-05

**Summary:**

Reviewers mainly pointed out:
- Limitation to single-layer *linear* transformers, L-infty norm perturbations, and the assumption of clean demonstrations
- Limited evaluations to assess the benefits of adversarial pre-training vs simple baselines (2dSm)
- Overstated claims about universal robustness (2dSm and SXX6)

**Reviewer Concerns:**

The authors mainly argue this is a theoretical result using available modeling tools.  This may not appeal to all members of the community, e.g. as seen in the ratings by reviewers and oCaH and 2dSm.  That said, there is definitely value in the theoretical development and proofs contributed in this submission.

**Reviewer Scores:**

Reviewer ratings were generally favorable starting at 8/6/6/4.  Following the discussion, I would expect a final score above 6.

---

### Decision · Program_Chairs · 2026-01-26

Accept (Poster)